# Direct Distributional Optimization for Provable Alignment of Diffusion Models

**Ryotaro Kawata**[1,2,*]**, Kazusato Oko**[2,3,†]**, Atsushi Nitanda**[4,5,‡]**, Taiji Suzuki**[1,2,§]

[1]Department of Mathematical Informatics, University of Tokyo, Japan
[2]Center for Advanced Intelligence Project, RIKEN, Japan
[3]Department of ECCS, UC Berkeley
[4]CFAR and IHPC, Agency for Science, Technology and Research (A⋆STAR), Singapore
[5]College of Computing and Data Science, Nanyang Technological University, Singapore
[*]`kawata-ryotaro725@g.ecc.u-tokyo.ac.jp`
[†]`oko@berkeley.edu`
[‡]`atsushi_nitanda@cfar.a-star.edu.sg`
[§]`taiji@mist.u-tokyo.ac.jp`

## Abstract

We introduce a novel alignment method for diffusion models from distribution optimization perspectives while providing rigorous convergence guarantees. We first formulate the problem as a generic regularized loss minimization over probability distributions and directly optimize the distribution using the Dual Averaging method. Next, we enable sampling from the learned distribution by approximating its score function via Doob's $h$-transform technique. The proposed framework is supported by rigorous convergence guarantees and an end-to-end bound on the sampling error, which imply that when the original distribution's score is known accurately, the complexity of sampling from shifted distributions is independent of isoperimetric conditions. This framework is broadly applicable to general distribution optimization problems, including alignment tasks in Reinforcement Learning with Human Feedback (RLHF), Direct Preference Optimization (DPO), and Kahneman-Tversky Optimization (KTO). We empirically validate its performance on synthetic and image datasets using the DPO objective.

## 1 Introduction

Diffusion models (Sohl-Dickstein et al., 2015; Ho et al., 2020; Song et al., 2021) have recently emerged as powerful tools for learning complex distributions and performing efficient sampling. Within the framework of foundation models, a common approach involves pre-training on large-scale datasets, followed by adapting the model to downstream tasks or aligning it with human preferences (Ouyang et al., 2022). This alignment is typically formalized as nonlinear distribution optimization, with a regularization term that encourages proximity to the pre-trained distribution. Examples of such alignment methods include Reinforcement Learning with Human Feedback (RLHF) (Ziegler et al., 2020), Direct Preference Optimization (DPO) (Rafailov et al., 2023; Wallace et al., 2024), and Kahneman-Tversky Optimization (KTO) (Ethayarajh et al., 2024; Li et al., 2024a).

Specifically, there methods solve the minimization problems of a regularized functional $F(q) + \beta D_{\mathrm{KL}}(q \| p_{\mathrm{ref}})$ over $q$ in the probability space $\mathcal{P}$, where $p_{\mathrm{ref}}$ is the probability density corresponding to the pretrained diffusion model. However, this type of distributional optimization problem over the density $q$ is challenging because the output density governed by the reference model cannot be evaluated and neither is the aligned model $q$. They are accessible only through samples generated from their corresponding generative models. Existing distributional optimization methods such as *mean-field Langevin dynamics* (Mei et al., 2018) and *particle dual averaging* (Nitanda et al., 2021) resolved this problem by adapting a Langevin type sampling procedure to calculate a functional derivative of the objective without explicitly evaluating the densities. However, the distributions $q$ and $p_{\mathrm{ref}}$ are highly complex and multimodal from which it is extremely hard to generate data by a standard MCMC type methods including the Langevin dynamics. This difficulty can be mathemati-

cally characterized by isoperimetric conditions, such as logarithmic Sobolev inequality (LSI) (Bakry et al., 2014), that has usually exponential dependency on the data dimension $d$ for multimodal data yielding the curse of dimensionality. Unfortunately, the existing distribution optimization methods mentioned above are sensitive to the LSI constant, so they suffer from severely slow convergence, failing to align diffusion models.

That is to say, alignment of diffusion models has two challenges: (i) inaccessibility of the output densities and (ii) muitimodality of the densities. This naturally leads to a fundamental question:

*Can we develop an alignment algorithm for diffusion models from distribution optimization perspectives, while ensuring rigorous convergence guarantees without isoperimetric conditions?*

We address this question by developing a diffusion-model based distribution optimization method and providing rigorous convergence and sampling error guarantees, and demonstrate its applicability to several tasks involved with diffusion model alignment. Our method represents the aligned model by a diffusion model that can be described by merely adding a correction term to the score function of the original reference model. During optimizing the model, we don't rely on any MCMC sampler but only use samples generated by the original reference model (and the aligned diffusion model). This characteristics is helpful to resolve the issue of isoperimetric condition.

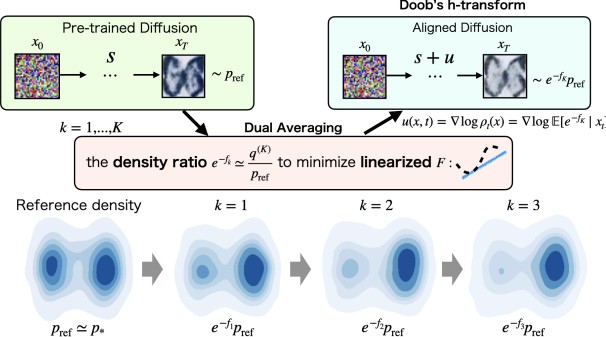

Figure 1: Overview of the proposed method integrating Dual Averaging and Doob's h-transform.

## 1.1 OUR CONTRIBUTIONS

To tackle the two challenges mentioned above: absence of (i) sampling guarantee and (ii) isoperimetry, we propose a general framework that integrates dual averaging (DA) method (Nesterov, 2009) and diffusion model (Sohl-Dickstein et al., 2015; Ho et al., 2020; Song et al., 2021). The DA method is an iterative algorithm that constructs the Gibbs distribution converging to the optimal distribution (i.e., alignment). A key advantage of this DA scheme is its ability to bypass isoperimetric conditions with the help of high sampling efficiency of the reference diffusion model, enabling isoperimetry-free sampling from an aligned distribution. Specifically, an aligned diffusion process that approximately generates the Gibbs distribution obtained by DA method can be constructed through the Doob's $h$-transform (Rogers & Williams, 2000) and density ratio estimation with respect to a reference distribution using neural networks (see Figure 1 for illustration).

We summarize our contribution below.

- We establish a model alignment method to align diffusion models, with convergence guarantees for both convex and nonconvex objectives, based on distribution optimization theory and Doob's $h$-transform. Notable distinctions from other mean-field optimization methods are that our method works without isoperimetry conditions such as LSI and allows for sampling from multimodal distributions.

- We also analyze the sampling error due to the approximation of the drift estimators with neural networks and the discretization of the process, and evaluate how these errors affect the final sampling accuracy.

- Our general framework encompasses several major alignment problems such as RLHF, DPO, and KTO, establishing a provable alignment method for these scenarios. We demonstrate the applicability of our framework to these settings and empirically validate its performance

on both synthetic and image datasets, aiming at data augmentation for a specific mode of distribution, using DPO objective.

We also emphasize that our method has the potential to be applied to general distribution optimization problems beyond alignment tasks such as density ratio estimation under the covariate shift setting (Sugiyama et al., 2008; Tsuboi et al., 2009) and climate change tracking (Ling et al., 2024).

## 1.2 RELATED WORK.

**Mean-field optimization.** PDA method (Nitanda et al., 2021; Nishikawa et al., 2022), an extension of DA method (Nesterov, 2009) to the distribution optimization setting, was the first method that proves the quantitative convergence for minimizing entropy regularized convex functional. Subsequently, P-SDCA method (Oko et al., 2022), inspired by the SDCA method (Shalev-Shwartz & Zhang, 2013), achieved the linear convergence rate. Mean-field Langevin dynamics (Mei et al., 2018) is the most standard particle-based distribution optimization method, derived as the mean-field limit of the noisy gradient descent, and its convergence rate has been well studied by Mei et al. (2018); Hu et al. (2021); Nitanda et al. (2022); Chizat (2022); Suzuki et al. (2023); Nitanda (2024). Additionally, several mean-field optimization methods such mean-field Fisher-Rao gradient flow (Liu et al., 2023b) and entropic fictitious play (Chen et al., 2023a; Nitanda et al., 2023) have been proposed with provable convergence guarantees. We note that the convergence rates of these methods were established under isoperimetric conditions such as log-Sobolev and Poincaré inequalities, which ensure concentration of the probability mass. IKLPD method (Yao et al., 2024) shares similarities with our method, as it employs the normalizing flow to solve intermediate subproblems in the distributional optimization procedure, and its convergence does not depend on isoperimetric conditions. However, the applicability of IKLPD to alignment tasks remains uncertain since handling the proximity to the reference distribution is non-trivial.

**Fine-tuning of diffusion models.** Recently, alignment of diffusion models have been investigated, inspired by LLM fine-tune methods, such as RLHF (Ziegler et al., 2020), DPO (Rafailov et al., 2023), and KTO (Ethayarajh et al., 2024). Applying them to diffusion models entails additional difficulty since the output density $p_{\mathrm{ref}}$ of the diffusion model is not available, and hence several techniques have been developed to circumvent the explicit calculation of $p_{\mathrm{ref}}$. For instance, Fan et al. (2023); Black et al. (2024); Clark et al. (2024) invented the maximization algorithm of the reward in each diffusion time step. Uehara et al. (2024b) used Doob's $h$-transform to compute the correction term that can be automatically derived from the density ratio between the generated and the reference distributions. Instead of optimizing original DPO and KTO objectives, Wallace et al. (2024) considered the evidence lower bound (ELBO) and Li et al. (2024a) defined a new objective function to replicate KTO. Marion et al. (2024) also studied fine-tuning of diffusion models as distributional optimization within the RLHF framework and conducted convergence analysis for the one-dimensional Gaussian distribution.

## 2 PROBLEM SETTING

**Distributional Optimization.** Let $\mathcal{P}$ be the space of probability density functions with respect to the Lebesgue measure on $(\mathbb{R}^d, \mathcal{B}(\mathbb{R}^d))$. Let $F : \mathcal{P} \to \mathbb{R}$ be a functional and $p_{\mathrm{ref}} \in \mathcal{P}$ be the reference density. In this work, we consider the regularized loss minimization problem over $\mathcal{P}$:

$$\min_{q \in \mathcal{P}} \left\{ L(q) := F(q) + \beta D_{\mathrm{KL}}(q \| p_{\mathrm{ref}}) \right\}, \tag{1}$$

where $D_{\mathrm{KL}}(q \| p_{\mathrm{ref}}) := \mathbb{E}_q[\log \frac{q}{p_{\mathrm{ref}}}]$ is the Kullback-Leibler divergence, and $\beta > 0$ is a regularization coefficient. We assume that $F$ is differentiable. That is, the functional $F$ has *first order variation* $\frac{\delta F}{\delta q} : \mathcal{P} \times \mathbb{R}^d \ni (q, x) \to \frac{\delta F}{\delta q}(q, x) \in \mathbb{R}$ such that for all $q, q' \in \mathcal{P}$,

$$\frac{\mathrm{d}F(q + \epsilon(q' - q))}{\mathrm{d}\epsilon} \bigg|_{\epsilon=0} = \int \frac{\delta F}{\delta q}(q, x)(q' - q)(x)\mathrm{d}x.$$

In the following we assume that there exists a unique minimizer $\hat{q}_{\mathrm{opt}} := \arg\min_{q \in \mathcal{P}} L(q)$.

**Diffusion Models.** $p_{\mathrm{ref}}$ is the output density of a pre-trained diffusion model (Sohl-Dickstein et al., 2015; Song & Ermon, 2019; Ho et al., 2020; Song et al., 2021; Vahdat et al., 2021), while $p_* \in \mathcal{P}$ is the target distribution of pre-training. A "noising" process $\{\bar{X}_t\}_{t \geq 0}$ denotes the Ornstein-Uhlenbeck

(OU) process from $p_*(x)$. The law of $\bar{X}_t$ can be written as $p_t(x) = \int \mathcal{N}(m_t \bar{X}_0, \sigma_t) \mathrm{d}p_*(\bar{X}_0)$ with $m_t = e^{-t}, \sigma_t^2 = 1 - e^{-2t}$. Then, the reverse process $\{\bar{X}_t^{\leftarrow}\}_{0 \leq t \leq T}$ ($T \geq 0$) can be defined as

$$\bar{X}_t^{\leftarrow} \sim p_t, \ \mathrm{d}\bar{X}_t^{\leftarrow} = \{\bar{X}_t^{\leftarrow} + 2\nabla \log p_{T-t}(\bar{X}_t^{\leftarrow})\}\mathrm{d}t + \sqrt{2}\mathrm{d}B_t.$$

Then it holds that $\mathrm{Law}(\bar{X}_t^{\leftarrow}) = \mathrm{Law}(\bar{X}_{T-t})$, which enables us to sample from $p_*$. In practice, we approximate the score $\nabla \log p_{T-t}(\bar{X}_t^{\leftarrow})$ function by a score network $s : \mathbb{R}^{d+1} \to \mathbb{R}^d : (x,t) \mapsto s(x,t)$. In addition, we initialize $X_0^{\leftarrow} \sim \mathcal{N}(0, I_d) \simeq p_T$ and the process is time-discretized. The random variable generated by the following dynamics with step size $h$ is denoted by $\{X_t^{\leftarrow}\}_{0 \leq t \leq T}$.

$$X_0^{\leftarrow} \sim \mathcal{N}(0, I_d), \ \mathrm{d}X_t^{\leftarrow} = \{X_{lh}^{\leftarrow} + 2s(X_{lh}^{\leftarrow}, lh)\}\mathrm{d}t + \sqrt{2}\mathrm{d}B_t, \ t \in [lh, (l+1)h], \ l = 1, ..., L = T/h.$$

In the same way, we define $q_t$ as the density of the diffusion process corresponding to $q_*$, and $\{\bar{X}_t\}_{t \geq 0}$ and $\{\bar{X}_t^{\leftarrow}\}_{0 \leq t \leq T}$ as the corresponding noising and the backward process.

Now we have two challenges in this problem (1):

(A). **Multimodality of $p_{\mathrm{ref}}$ and $\hat{q}_{\mathrm{opt}}$.** Our goal is to obtain samples from $\hat{q}_{\mathrm{opt}}$. Now we tackle with the case that $p_{\mathrm{ref}}$ and $\hat{q}_{\mathrm{opt}}$ have high *multimodality*: $p_{\mathrm{ref}}$ and $\hat{q}_{\mathrm{opt}}$ have multiple modes or maxima, in other words, the potentials $-\log p_{\mathrm{ref}}$ and $-\log \hat{q}_{\mathrm{opt}}$ are extremely far from concave. Probability distributions of real-world data like images often have such complex structures, which implies that LSI is significantly weak.

(B). **Inaccesibility to the density $p_{\mathrm{ref}}$.** We cannot directly calculate the density $p_{\mathrm{ref}}$ because diffusion models only have the score network. We only have information of $p_{\mathrm{ref}}$ as samples from $p_{\mathrm{ref}}$.

**Inapplicability of Mean-Field Langevin Dynamics.** Mean-field Langevin dynamics (MFLD) (Mei et al., 2018) is the most standard particle-based optimization method tailored to solve the special case of the problem (1) where $F$ is convex functional and $p_{\mathrm{ref}}$ is a strongly log-concave distribution such as Gaussian distribution. The convergence rate of MFLD (Nitanda et al., 2022; Chizat, 2022) has been established under the condition where the proximal distribution: $p_q \propto p_{\mathrm{ref}} \exp\left(-\beta^{-1}\frac{\delta F}{\delta q}(q, \cdot)\right)$ associated with MFLD iteration $q$ satisfies logarithmic Sobolev inequality (LSI), which says sufficient concentration of $p_q$. Typically, LSI for $p_q$ should rely on the isoperimetry of the reference distribution $p_{\mathrm{ref}}$ with Holley-Strook argument (Holley & Stroock, 1987) since $\frac{\delta F}{\delta q}(q, \cdot)$ is not expected to encourage it in general. Therefore, the replacement of $p_{\mathrm{ref}}$ to a pre-trained distribution, which is highly complex and has multi-modality, leads to failure or weak satisfaction of LSI. As a result, the convergence of MFLD is significantly slowed down. Additionally, MFLD is practically implemented so that finite particles approximately follow the ideal dynamics of MFLD, and it is not intended for resampling from the final distribution represented by these particles. In short, the lack of (i) a sampling guarantee and (ii) a isoperimetric condition limits the applicability of MFLD in our problem setting.

To address these challenges, we employ the following strategy, which will be detailed in Section 4.

(A). **Modifications of diffusion models for sampling from a multimodal distribution.** First, we obtain samples from $p_{\mathrm{ref}}$ using a diffusion model which works for a broad range of probability distributions that have complex structures such as multimodality. Then, we reconstruct the diffusion model to sample from $\hat{q}_{\mathrm{opt}}$ by simply adding a correction term.

(B). **A distributional optimization algorithm that does not require the density $p_{\mathrm{ref}}$.** Second, following the dual averaging (DA) method in the distribution optimization setting, we constract a sequence of distributions $\hat{q}^{(k)}$ converging to the optimum, which guides an aligned diffusion process in combination with the density ratio estimation using neural networks with Doob's $h$-transform. We only have to calculate the density ratio between $\hat{q}^{(k)}$ and $p_{\mathrm{ref}}$ and the samples from $p_{\mathrm{ref}}$ to run our algorithm.

## 3 APPLICATIONS

Our framework of the distributional optimization for the pre-trained diffusion models includes important fine-tuning methods for diffusion models. The first term $F(q)$ in (1) represents *the human preference*, acting as feedback from the outputs of $p_{\mathrm{ref}}$. Note that $F(q)$ can be dependent of $p_{\mathrm{ref}}$. The second term $\beta D_{\mathrm{KL}}(q\|p_{\mathrm{ref}})$ in (1) provides regularization to prevent $q$ from collapsing.

**Example 1** (Reinforcement Learning). *Our study includes the case $F$ is limited to be a linear functional:*

$$\min_{q \in \mathcal{P}} \left\{ \mathbb{E}_q[-r(x)] + \beta D_{\mathrm{KL}}(q \| p_{\mathrm{ref}}) \right\},$$

*where $r(x)$ is a reward function. In this case, the optimal distribution is obtained as $q_*(x) \propto \exp\left(\frac{r(x)}{\beta}\right) p_{\mathrm{ref}}(x)$. This type of the problems has been studied as the Reinforcement Learning (Fan et al., 2023; Black et al., 2024; Clark et al., 2024; Uehara et al., 2024b; Marion et al., 2024).*

The following two examples have not been directly solved via diffusion models:

**Example 2** (DPO). *Direct Preference Optimization (DPO) (Rafailov et al., 2023) is an effective approach for learning from human preference for not only language models but also diffusion models. Our algorithm can directly minimize the DPO objective, while Wallace et al. (2024) tried applying DPO to diffusion models via minimization of an upper bound of the original objective. In DPO, humans decide which sample is more preferred given two samples from $p_{\mathrm{ref}}$. Let $x_w$ and $x_l$ be "winning" and "losing" samples from $p_{\mathrm{ref}}$. $x_w \succ x_l$ denote the event that $x_w$ is preferred to $x_l$. The DPO objective can be written as*

$$L_{\mathrm{DPO}}(q) \coloneqq -\mathbb{E}_{x_w \sim p_{\mathrm{ref}}} \mathbb{E}_{x_l \sim p_{\mathrm{ref}}} \left[ \log \sigma \left( \gamma \log \frac{q(x_w)}{p_{\mathrm{ref}}(x_w)} - \gamma \log \frac{q(x_l)}{p_{\mathrm{ref}}(x_l)} \right) \mathbb{1}_{x_w \succ x_l}(x_w, x_l) \right],$$

*where $\mathbb{E}_{x \sim p}$ denotes expectation with respect to $x$ whose probability density is $p \in \mathcal{P}$, $\sigma$ is a sigmoid function, $\mathbb{1}_{x \succ y}(x, y)$ is one if $x \succ y$ and is zero otherwise. Precisely, the functional derivative of $L_{\mathrm{DPO}}(q)$ is calculated as*

$$\frac{\delta L_{\mathrm{DPO}}}{\delta q}(q, x) = -\gamma \mathbb{E}_{x_l \sim p_{\mathrm{ref}}} \left[ (1 - \sigma(-\gamma f(x) + \gamma f(x_l))) \frac{\int e^{-f} dp_{\mathrm{ref}}}{e^{-f(x)}} \mathbb{1}_{x \succ x_l}(x, x_l) \right]$$

$$+ \gamma \mathbb{E}_{x_w \sim p_{\mathrm{ref}}} \left[ (1 - \sigma(-\gamma f(x_w) + \gamma f(x))) \frac{\int e^{-f} dp_{\mathrm{ref}}}{e^{-f(x)}} \mathbb{1}_{x_w \succ x}(x_w, x) \right], \quad (2)$$

*where $q = e^{-f} p_{\mathrm{ref}} / \int e^{-f} \mathrm{d}p_{\mathrm{ref}}$. See Appendix B for the derivation. Therefore, we only need samples from $p_{\mathrm{ref}}$ and the log-density ratio or the potential $f$ to calculate the functional derivatives.*

**Example 3** (KTO). *Our algorithm can also minimize $L_{\mathrm{KTO}}$ directly. Assume that the whole data space $\mathbb{R}^d$ is split into a desirable domain $\mathcal{D}_{\mathrm{D}}$ and an undesirable domain $\mathcal{D}_{\mathrm{U}}$. The objective of the original KTO (Ethayarajh et al., 2024) is formulated as*

$$L_{\mathrm{KTO}}(q) = \mathbb{E}_{x \sim p_{\mathrm{ref}}} \left[ \gamma_D \left( 1 - \sigma \left( \kappa \log \frac{q}{p_{\mathrm{ref}}} - D_{\mathrm{KL}}(q \| p_{\mathrm{ref}}) \right) \right) \mathbb{1}_{\{x \in \mathcal{D}_{\mathrm{D}}\}} \right.$$

$$\left. + \gamma_U \left( 1 - \sigma \left( D_{\mathrm{KL}}(q \| p_{\mathrm{ref}}) - \kappa \log \frac{q}{p_{\mathrm{ref}}} \right) \right) \mathbb{1}_{\{x \in \mathcal{D}_{\mathrm{U}}\}} \right],$$

*where $\gamma_D$, $\gamma_U$, $\kappa$ are hyper parameters, and $\sigma$ is a sigmoid function. Li et al. (2024a) defined objectives compatible with diffusion models based on KTO, but our algorithm can directly minimize $L_{\mathrm{KTO}}$. Like the DPO objective, we only have to calculate samples from $p_{\mathrm{ref}}$ and the potential $f$ of the density ratio ($f$ is defined as $q = e^{-f} p_{\mathrm{ref}} / \int e^{-f} dp_{\mathrm{ref}}$) to calculate the functional derivatives. Please refer to Appendix B for the concrete formulation of $\frac{\delta L_{\mathrm{KTO}}}{\delta q}$.*

## 4 THE NONLINEAR DISTRIBUTIONAL OPTIMIZATION ALGORITHM

Now we make a concrete introduction of our proposed approach. Our goal of the distribution optimization (1) is to train a neural network that approximates $\hat{f}_{\mathrm{opt}} = \log \frac{\hat{q}_{\mathrm{opt}}}{p_{\mathrm{ref}}} + (\mathrm{const.})$. To achieve this, we utilize the *Dual Averaging* (DA) algorithm (Nesterov, 2009; Nitanda et al., 2021; Nishikawa et al., 2022), and we iteratively construct a tentative local potential $f_k$ by approximating the update the DA algorithm. After we obtain $f_K$, we estimate the diffusion model that generates the desired output (approximately) following $q_*$, through *Doob's h-transform technique*.

**Phase 1: Dual Averaging.** Let $f_1$ be a randomly initialized potential. First, we initialize $q^{(1)} \propto \exp(-f_1) p_{\mathrm{ref}}$, where $f_1$ is a randomly initialized neural network. Then, the distribution $q^{(k)}$ is updated recursively by pulling back the weighted sum of gradients from the dual space to the primal space. There are two options of DA methods. For a given hyper-parameter $\beta' > 0$, the update of Option 1 is given as

**(Opt. 1)** $\quad \hat{q}^{(k+1)} = \underset{q \in \mathcal{P}}{\arg\min} \left\{ \frac{2}{k(k+1)} \sum_{j=1}^{k} j \left( \mathbb{E}_q \left[ \frac{\delta F}{\delta q}(q^{(j)}) \right] + \beta D_{\mathrm{KL}}(q \| p_{\mathrm{ref}}) \right) + \frac{2\beta'}{k} D_{\mathrm{KL}}(q \| p_{\mathrm{ref}}) \right\}$

$$=: \exp\left(-\bar{g}^{(k)}\right) p_{\mathrm{ref}}, \tag{3}$$

where $\bar{g}^{(k)}(x) = \sum_{j=1}^{k} w_j^{(k)} \frac{\delta F}{\delta q}(q^{(j)}, x)$, $w_j^{(k)} = \frac{j}{\beta k(k+1)/2 + \beta'(k+1)}$ $(j = 1, \ldots, k)$. By Lemma 4 in Appendix A.2, $\bar{g}^{(k)}$ can be explicitly determined. We train a neural network $f_{k+1}$ to approximate $\bar{g}^{(k)}$[1] and define the next step as $q^{(k+1)} \propto \exp(-f_{k+1})p_{\text{ref}}$. Similarly, the update of Option 2 is given as

**(Opt. 2)** $\quad \hat{q}^{(k+1)} = \underset{q \in \mathcal{P}}{\arg\min} \left\{ \frac{2}{k(k+1)} \sum_{j=1}^{k} j \left( \mathbb{E}_q \left[ \frac{\delta F}{\delta q}(q^{(j)}) - \beta \bar{g}^{(j)} \right] \right) + \frac{2\beta'}{k} D_{\text{KL}}(q \| p_{\text{ref}}) \right\}.$ $\quad$ (4)

Here, we again express as $\hat{q}^{(k+1)}(x) \propto \exp(-\bar{g}^{(k)}(x))p_{\text{ref}}(x)$ where $\bar{g}^{(k)}(x) = \sum_{j=1}^{k} w_j^{(k)} (\frac{\delta F}{\delta q}(q^{(j)}, x) - \beta \bar{g}^{(j)}(x))$ with $w_j^{(k)} = \frac{j}{\beta'(k+1)}$. Then, $q^{(k+1)}$ is obtained in the same manner as Option 1. This phase of DA update is summarized in Algorithm 4. For the more detailed algorithm in Option 1, please refer to Algorithm D.1.

---

**Algorithm 4.1** Dual Averaging (DA)

---

**Input:** $s$: pre-trained score, $f_1$: initialized neural networks
**Output:** $f_K$: a trained potential.
$\quad$ Set $q^{(0)} = p_{\text{ref}}$ and $q^{(1)} \propto \exp(-f_1)p_{\text{ref}}$
$\quad$ **for** $k = 1, \ldots, K - 1$ **do**
$\quad\quad$ Obtain $\bar{g}^{(k)}$ via the DA algorithm with Option 1 (Eq. (3)) or Option 2 (Eq. (4)) using the recurrence formula (47), where $\hat{q}^{(k+1)} \propto \exp(-\bar{g}^{(k)})p_{\text{ref}}$ is the ideal update.
$\quad\quad$ Train a neural network $f_{k+1}$ to approximate $\bar{g}^{(k)}$, and set $q^{(k+1)} \propto \exp(-f_{k+1})p_{\text{ref}}$.
$\quad$ **end for**

---

**Phase 2: Sampling with Doob's h-transform.** After we obtain the solution $f_K$, we want to sample from $q_K \propto \exp(-f_K)p_{\text{ref}}$, which approximates the optimal solution of (1). When sampling from $q_K$, it is necessary to obtain the score function related to this distribution. However, constructing the score function of $q_*$ only from the score function of $p_{\text{ref}}$ and $f_K$ requires a particular technique. Specifically, we apply Doob's h-transform (Rogers & Williams, 2000; Chopin et al., 2023; Uehara et al., 2024b; Heng et al., 2024). By introducing the correction term $u_* : \mathbb{R}^{d+1} \to \mathbb{R}^d$ defined by

$$u_*(y, t) = \nabla \log \mathbb{E}[\exp(-f_*(\bar{X}_T^{\leftarrow})) \mid \bar{X}_t^{\leftarrow} = y],$$

the score function of $q_*$ at $(x, t)$ is written as $\nabla \log q_t(y) = \nabla \log p_t(X_t^{\leftarrow}, T - t) + u_*(y, t)$, where $q_t$ is the law of the backward process at time $t$ whose output distribution is the optimal solution $q_*$. We provide the derivation in Lemma 12 and refer readers to Rogers & Williams (2000); Chopin et al. (2023) for more details and a formal treatment of Doob's h-transform. By approximating $\log p_t(X_t^{\leftarrow}, T - t)$ by the score network $s(x, t)$ and the correction term $u_*(x, t)$ by $u(x, t)$ and discretizing the dynamics, we obtain the following update

$$Y_0^{\leftarrow} \sim \mathcal{N}(0, I_d), \quad dY_t^{\leftarrow} = \{Y_t^{\leftarrow} + 2(s(Y_{lh}^{\leftarrow}, lh) + u(Y_{lh}^{\leftarrow}, lh))\}dt + \sqrt{2}dB_t, \ t \in [lh, (l+1)h],$$

where $u(x, t)$ can be computed as $u(x, t) = \nabla_x \log \mathbb{E}[\exp(-f_K(X_T^{\leftarrow}))|X_t^{\leftarrow} = x]$, which can be estimated by running the reference diffusion model $(X_t^{\leftarrow})$. The practical treatment for this is discussed in Appendix C. For experimental information, please have a look at Section 6 and Algorithm D.2 in Appendix D.2.

## 5 Theoretical Analysis

In this section, we give theoretical justification of our proposed algorithm. More concretely, we show the rate of convergence of the (inexact) DA method and give an approximation error bound on the diffusion model based on the h-transform.

### 5.1 Convergence rate of the DA method

We give the convergence rate of the DA algorithm in the two settings: when $F$ is (I) convex and (II) non-convex, respectively.

---

[1] For DPO and KTO, it suffices to obtain the neural network $\bar{g}^{(k)}$ to minimize $\mathbb{E}_{p_{\text{ref}}}[(f - \bar{g}^{(k)})^2]$ where the expectation with respect to $p_{\text{ref}}$ is simulated by generating data from $p_{\text{ref}}$, while obtaining $\bar{g}^{(k)}$ for general settings requires Doob's h-transform similar to Phase 2. Please also refer to Section 6.

**(I): Convex objective** $F$**.** First, we show the rate when $F$ is convex. We basically follow the proof technique of Nitanda et al. (2021); Nishikawa et al. (2022). In the analysis, we put the following assumption on $F$.

**Assumption 1.** *The loss function $F$ satisfies the following conditions:*

*(i) $\frac{\delta F}{\delta q}$ is bounded: There exists $B_F > 0$ such that $\|\frac{\delta F}{\delta q}(q)\|_\infty \leq B_F$ for any $q \in \mathcal{P}$,*

*(ii) $\frac{\delta F}{\delta q}$ is Lipshitz continuous with respect to the TV distance: There exists $L_{\mathrm{TV}} > 0$ such that $\|\frac{\delta F}{\delta q}(q) - \frac{\delta F}{\delta q}(q')\|_\infty \leq L_{\mathrm{TV}}\mathrm{TV}(q, q')$ for any $q, q' \in \mathcal{P}$.*

*(iii) $F$ is convex: $F(q) \geq F(q') + \int \frac{\delta F}{\delta q}(q')\mathrm{d}(q - q')$ for any $q, q' \in \mathcal{P}$,*

Then, we can show that the (inexact) DA algorithm achieves the following convergence rate that yields $\mathcal{O}(K^{-1})$ convergence of the objective.

**Theorem 1** (Convergence of the objective in Option 1 )**.** *Suppose that $\beta' \geq \beta$ and we train the potential $f_{k+1}$ so that it is sufficiently close to $\bar{g}^{(k)}$ as $\mathrm{TV}(\hat{q}^{(k)}, q^{(k)}) \leq \epsilon_{\mathrm{TV}}$ for all $k$. Then, under Assumption 1, Option 1 satisfies the following convergence guarantee:*

$$\frac{2}{K(K+1)}\sum_{k=1}^{K} k\left[L(\hat{q}^{(k)}) - L(\hat{q}_{\mathrm{opt}})\right] \leq 2L_{\mathrm{TV}}\epsilon_{\mathrm{TV}} + \left(\frac{2B_F}{K(K+1)} + \frac{2\beta' D_{\mathrm{KL}}(q_*\|p_{\mathrm{ref}}) + 2B_F^2\beta^{-1}}{K}\right).$$

See Appendix A.1 for the proof. From this theorem, we see that the DA algorithm with Option 1 achieves $\mathcal{O}(1/K)$ convergence. The assumption (ii) in Assumption 1 is required to bound an expectation of $\bar{g}^{(k)}$ in the bound. The assumption (iii) is required to bound the difference between the exact update $\hat{q}^{(k)}$ and the inexact update $q^{(k)}$. If the update is exact, we don't need this assumption.

**(II): Non-convex objective** $F$**.** We also give a convergence for a non-convex loss $F$. Here, we put the following assumption.

**Assumption 2.** *The loss function $F$ satisfies the following conditions:*

*(i) $\frac{\delta F}{\delta q}$ is bounded: There exists $B_F > 0$ such that $\|\frac{\delta F}{\delta q}(q)\|_\infty \leq B_F$ for any $q \in \mathcal{P}$,*

*(ii) $\frac{\delta F}{\delta q}$ is Lipshitz continuous with respect to the TV distance: $\|\frac{\delta F}{\delta q}(q) - \frac{\delta F}{\delta q}(q')\|_\infty \leq L_{\mathrm{TV}}\mathrm{TV}(q, q')$ for any $q, q' \in \mathcal{P}$,*

*(iii) $F$ is lower bounded.*

Assumptions (i) and (ii) are the same as the convex case (Assumption 1), and lower boundedness (iii) is weaker than convexity in Assumption 1. Assumption (ii) induces the following type of smoothness commonly observed in standard optimization: (ii)' There exists $S_F \geq 0$ such that $F(q) \leq F(q') + \int \frac{\delta F}{\delta q}(q')\mathrm{d}(q - q') + \frac{S_F}{2}D_{\mathrm{KL}}(q\|q')$. When the inner-loop error is ignored, it is possible to prove convergence using only the smoothness assumption (ii)' instead of assumption (ii). For details, please refer to Appendix A.1.1.

Under this assumption, we can show the following convergence guarantee with respect to the sequence $(\hat{q}^{(k)})_{k=1}^{K}$ even in a non-convex setting.

**Theorem 2.** *Suppose that $\beta' > \beta + L_{\mathrm{TV}}$ and $\mathrm{TV}(\hat{q}^{(k)}, q^{(k)}) \leq \epsilon_{\mathrm{TV}}$ for all $k$, then under Assumption 2, both Option 1 and 2 yield the following convergence:*

*(i) $\lim_{k\to\infty} D_{\mathrm{KL}}(\hat{q}^{(k+1)}\|\hat{q}^{(k)}) = 0$.*

*(ii) For all $K$, it holds that*

$$\min_{k=1,\ldots,K}\left\{c_k D_{\mathrm{KL}}(\hat{q}^{(k+1)}\|\hat{q}^{(k)})\right\} \leq \frac{(\tilde{L}_1(\hat{q}^{(1)} - L(\hat{q}_{\mathrm{opt}})) + (L_{\mathrm{TV}} + B_F)K\epsilon_{\mathrm{TV}}}{K\beta'} =: \Psi_K,$$

*where $c_k = \frac{\beta k + 2\beta'}{2}$ for Option 1 and $c_k = 1$ for Option 2.*

See Appendix A.1.1 for the proof. The proof utilizes an analogous argument with Liu et al. (2023a) that analyzed convergence of a DA method in a finite dimensional non-convex optimization problem. However, we need to reconstruct a proof because we should work on the probability measure space,

which is not a vector space, and carefully make use of the property of the KL-divergence. We see that $\Psi_K = \mathcal{O}(1/K)$ if $\epsilon_{\mathrm{TV}}$ is sufficiently small as $\epsilon_{\mathrm{TV}} = \mathcal{O}(1/K)$, and thus the discrepancy between $\hat{q}^{(k+1)}$ and $\hat{q}^{(k)}$ converges. Especially, the convergence of $D_{\mathrm{KL}}(\hat{q}^{(k+1)} \| \hat{q}^{(k)})$ yields the convergence of the "dual variable" for Option 2 as in the following corollary. For that purpose, we define $\tilde{L}_k(q) := L(q) + \frac{\beta'}{k} D_{\mathrm{KL}}(q \| p_{\mathrm{ref}})$ (see its similarity to the inner objective of the DA update (3) and (4)), and define $\psi_q(g) = \log\left(\mathbb{E}_q[\exp(-g + \mathbb{E}_q[g])]\right)$ which is a "moment generating function" of a dual variable $g$.

**Corollary 1** (Convergence in Option 2 ). *Under the same condition as Theorem 2, we have that*

$$\min_{1 \leq k \leq K} \psi_{\hat{q}^{(k)}}\left(\frac{k}{\beta'(k+1)} \frac{\delta \tilde{L}_k}{\delta q}(\hat{q}^{(k)})\right) \leq \Psi_K.$$

Roughly speaking, this corollary indicates that the variance of the gradient $\frac{\delta \tilde{L}_k}{\delta q}(\hat{q}^{(k)})$ converges as $\mathrm{Var}_{\hat{q}^{(k)}}\left(\frac{\delta \tilde{L}_k}{\delta q}(\hat{q}^{(k)})\right) = \mathcal{O}(1/K)$ because we may approximate $\psi_q(g) \simeq \frac{1}{2} \mathrm{Var}_q(g)$ when $|g - \mathbb{E}_q[g]|$ is sufficiently small. Therefore, we have a convergence guarantee of the dual variable (gradient) $\frac{\delta \tilde{L}_k}{\delta q}(\hat{q}^{(k)}, x) \to 0$ (up to a constant w.r.t. $x$) even in a non-convex setting, which justifies usage of our method for general objective functions.

## 5.2 DISCRETIZATION ERROR OF DOOB'S H-TRANSFORM

We now provide the sampling error analysis after obtaining the approximate solution $f_K$. In the analysis of Dual Averaging, $p_{\mathrm{ref}}$ was assumed to be known, and the goal was to obtain the optimal solution of (1) that corresponds to $\hat{q}_{\mathrm{opt}} = \exp(-\hat{f}_{\mathrm{opt}})p_{\mathrm{ref}}$. From here, considering that $p_{\mathrm{ref}}$ estimates $p_*$, we shift our focus to sample from the optimal alignment $q_* \propto \rho_* p_*$, while the dynamics we implement involves several approximations: (i) time discretization approximation, (ii) approximation of the score $\nabla_x \log p_t$ by $s(x, T - t)$, (iii) approximation of $\rho^*$ by $\rho = \exp(-f_K)$, (iv) approximation of $u_*$ by $u$.

To evaluate how such approximation affects the final quality of our generative model, we will give a bound of the TV-distance between $q_*$ and $\hat{q} = \rho p_{\mathrm{ref}}$ by putting all those errors together. To do so, we put the following assumption.

**Assumption 3.** *1. $\nabla \log p_t$ is $L_p$-smooth at every time $t$ and it has finite second moment $\mathbb{E}[\|\bar{X}_t\|_2^2] \leq \mathsf{m} < \infty$ for all $t \in \mathbb{R}_+$ and $x \in \mathbb{R}^d$.*

*2. $\nabla \log \rho_*$ is $L_\rho$-smooth and bounded as $C_\rho^{-1} \leq \rho_* \leq C_\rho$ for a constant $C_\rho$.*

*3. The score estimation error is bounded by $\mathbb{E}_{\bar{X}^{\leftarrow}}[\|s(\bar{X}_t^{\leftarrow}, t) - \nabla \log p_{T-t}(\bar{X}_t^{\leftarrow})\|^2] \leq \varepsilon$ at each time $t$.*

*4. $\mathbb{E}_{\bar{X}^{\leftarrow}}[\|u_*(\bar{X}_t^{\leftarrow}, t) - u(\bar{X}_{lh}^{\leftarrow}, lh)\|^2] \leq \varepsilon_{\rho,l}^2$ for any $1 \leq l \leq T/h$ and $t \in [lh, (l+1)h)$.*

This assumption is rather standard, for example, Chen et al. (2023c) employed these conditions except the last condition on $u$ and $u_*$. The fourth assumption in Assumption 3 is imposed to mathematically formulate the situation: $\hat{q}_{\mathrm{opt}}/p_{\mathrm{ref}} \simeq q_*/p_*$. Then, we obtain the following error bound:

**Theorem 3.** *Suppose that Assumption 3 is satisfied. Then, we have the following bound on the distribution $\hat{q}$ of $Y_T^{\leftarrow}$:*

$$\mathrm{TV}(q_*, \hat{q})^2 \lesssim T\varepsilon^2 + \sum_{l=1}^{T/h} h\varepsilon_{\rho,l}^2 + T(L_p C_\rho^2 + L_\rho)^2 (dh + \mathsf{m}h^2) + \exp(-2T)D_{\mathrm{KL}}(q_* \| N(0, I)).$$

The proof is given in Appendix C.1. It basically follows Chen et al. (2023c;b), but we have derived the smoothness of $\nabla \log(q_{*,t})$ from that of $\nabla \log(p_t)$. In this bound, we did not give any evaluation on $\varepsilon_{\rho,l}^2$, however, this error term can be bounded as follows with additional conditions.

**Assumption 4.** *(i) $\nabla_x s(\cdot, \cdot)$ is $H_s$-Lipschitz continuous in a sense that $\|\nabla_x s(x, t) - \nabla_y s(y, t)\|_{\mathrm{op}} \leq H_s \|x - y\|$ for any $x, y \in \mathbb{R}^d$ and $0 \leq t \leq T$ and $\mathbb{E}[\|s(\tilde{X}_{kh}^{\leftarrow}, kh)\|^2] \leq Q^2$ for any $k$, (ii) There exists $R > 0$ such that $\sup_{t,x}\{\|\nabla_x^2 \log p_t(x)\|_{\mathrm{op}}, \|\nabla_x^2 \log s(x, t)\|_{\mathrm{op}}\} \leq R$.*

**Theorem 4.** *Suppose that Assumptions 3 and 4 hold and $\|\rho_* - \rho\|_\infty \leq \varepsilon'$ and $\|\rho\|_\infty \leq C_\rho$. We also assume $\nabla \rho^*$ is bounded and Lipschitz continuous. Then, for any choice of $0 \leq h \leq \delta \leq 1/(1 + 2R)$, we have that*

$$\varepsilon_{\rho,l}^2 \lesssim C_\rho^3 \left\{ \Xi_{\delta,\varepsilon} + R_\varphi^2 \left(\varepsilon^2 + L_p^2 d(\delta + \mathsf{m}\delta^2)\right) + [L_\varphi^2(\mathsf{m} + Q^2 + dh) + R_\varphi^2(1 + 2R)^2]h^2 \right\}$$
$$+ \min\{T - lh, 1/(2 + 2R)\}^{-1}\varepsilon'^2,$$

where $\Xi_{\delta,\varepsilon} := C_\rho^2(1 + 2R)^2\delta + C_\rho^2\frac{1+\delta R_\varphi^2}{\delta}[\varepsilon^2 + L_p^2 d(h + \mathsf{m}h^2)]$, and $R_\varphi$ and $L_\varphi$ are constants introduced in Lemma 10.

The proof is given in Appendix C.3. We applied the so-called *Bismut-Elworthy-Li* integration-by-parts formula (Bismut, 1984; Elworthy & Li, 1994) to obtain the discretization error. By substituting $\delta \leftarrow \sqrt{h}$ to balance the terms related to $h$ and $\delta$, we obtain a simplified upper bound of $\sum_{l=1}^{T/h} h\varepsilon_{\rho,l}^2$ as $\sum_{l=1}^{T/h} h\varepsilon_{\rho,l}^2 \lesssim \left(1 + \frac{1}{\sqrt{h}}\right)\varepsilon^2 T + T\sqrt{h} + (T + \log(1/h))\varepsilon'^2$. These results can be seen as $h$-Transform extension of the approximation error analysis given in Chen et al. (2023c). However, the approximation error corresponding to $u$ has worse dependency on $h$. This is because the computation of $u$ uses the discretized process $p_{\mathrm{ref}}$ and is affected by its error while the ordinary diffusion model does not require sampling to obtain the score.

## 6 NUMERICAL EXPERIMENTS

We conducted numerical experiments to confirm the effectiveness of minimizing nonlinear objectives. We used Option 1 and $\beta$ was set to be 0.04. We also compared the DPO objective (Rafailov et al., 2023) we minimized with the upper-bound (Wallace et al., 2024) using a toy case, Gaussian Mixture Model (GMM). For detailed experimental setting, please refer to Appendix D.

**Alignment for Gaussian Mixture Model.** We aligned a score-based diffusion model to sample from a 2-dimensional GMM using the DPO objective. The reference density was defined as $\frac{1}{2}\left(\mathcal{N}(\mu_1, \Sigma) + \mathcal{N}(\mu_2, \Sigma)\right)$, $\mu_1 = [-2.5, 0]$, $\mu_2 = [2.5, 0]$, $\Sigma = [[1, 0], [0, 5]]$. The target point was $\mu_w := \mu_2$. The preference of points $x_w$ and $x_l$ was determined by the Euclidean distance $d(\cdot, \mu_w)$ from $\mu_w$. $x_w \succ x_l$ if and only if $d(x_w, \mu_w) < d(x_l, \mu_w)$. We describe the implementation details below:

*Dual Averaging.* We set the hyperparameter $\beta'$ in [0.04, 0.2], which controls the learning speed illustrated at the middle in Figure 2. In $k$th DA iteration ($k = 1, ..., K$), using empirical samples $\{x_i = (x_{1,i}, x_{2,i})\}_{i=1}^{1000}$ from $p_{\mathrm{ref}}$ and the previous potential $f_{k-1}$ ($q^{(k-1)} \propto e^{-f_{k-1}}p_{\mathrm{ref}}$) implemented by neural networks, we prepare $\{(x_i, \frac{\delta F}{\delta q}(q^{(k-1)}, x_i))\}_{i=1}^{1000}$ to construct $f_k \simeq \bar{g}^{(k-1)}$ with the recurrence formula (47) in Appendix D. Please note that, to calculate $\frac{\delta F}{\delta q}$ for DPO, we only need the $f_k$ and empirical samples from $p_{\mathrm{ref}}$ to calculate the expectation $\mathbb{E}_{x \sim p_{\mathrm{ref}}}[\cdot]$ as described in Eq. (2). The convergence of the true DPO loss is depicted in Figure 2, compared to Diffusion-DPO (Wallace et al., 2024). The true DPO loss, upperbound, and metric loss (mean of the Euclidean distance of the particles from $\mu_w$) attained by Diffusion-DPO and our method, with optimization time and GPU memory consumption (in Phase I for ours), are summarized in Table 1. Diffusion-DPO optimizes the approximate upperbound instead of the true loss, and therefore it failed to completely control the true loss. Compared to Diffusion-DPO, our algorithm allowed us to optimize the loss to a smaller value within an acceptable computational budget.

*Doob's h-transform.* We constructed the aligned model by calculating the correction terms. The conditional expectations were calculated using naive Monte Carlo method with 30000 samples. The number of the diffusion steps was 50. The histograms of aligned samples are shown at the right of Figure 2. In the simplest phase 2 that we present (Algorithm D.2), we used significant computational resources, with $\mathcal{O}(L^2)$ time complexity. To estimate the correction term $u$ for each sample simultaneously by Monte Carlo with $N$ samples, $\mathcal{O}(N)$ memory space is required. However, this Doob's h-transform technique itself has been used in image generation (Uehara et al., 2024a;b), Bayesian samping (Heng et al., 2024), and filtering (Chopin et al., 2023). In particular, computational resources can be saved by approximating the correction term using a neural ODE (Uehara et al., 2024b).

Table 1: The quantitative comparison between Diffusion-DPO and our method

|  | Ref. | Diffusion-DPO (50 iter., w/o reg.) | Diffusion-DPO (200 iter., w/ reg.) | **Ours** ($\beta' = 0.04$) |
|---|---|---|---|---|
| True DPO loss | 0.346 | 0.340 | 0.343 | **0.328** |
| (Approx.) Upperbound | - | 0.311 | 0.337 | - |
| Metric loss | 3.256 | 3.011 | 3.303 | **2.022** |
| Opt. time (s) | - | 585 | 2280 | 1320 (Phase I) |
| GPU memory (%) | - | 6.54 | 6.54 | 8.75 (Phase I) |

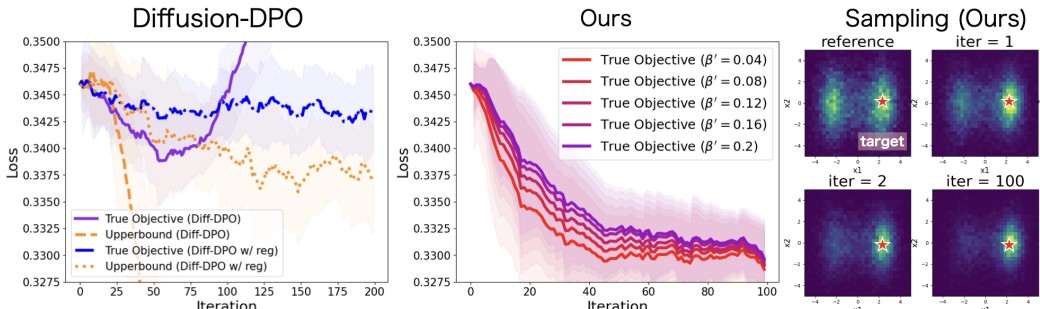

Figure 2: **Left and Middle.** The smoothed loss during optimization for Gaussian Mixture Model in Diffusion-DPO with/without regularization (left) and ours (middle). "True Objective": the true DPO loss (Rafailov et al., 2023) whose target point was $\mu_w = [2.5, 0]$. "Upperbound": An approximate upperbound of "Objective" optimized by Diffusion-DPO (Wallace et al., 2024). **Right.** Aligned samples by Doob's h-transform. "Ref." represents the empirical density of $p_{\text{ref}}$.

**Image Generation Alignment based on a Target Color.** We aligned the image generation of the basic pre-trained model available Diffusion Models Course (source: HuggingFace (2022)) to simulate specific generations for data augmentation. The pre-trained model generates RGB-colored $32 \times 32$ pixel images of butterflies. In this experiment, the model was aligned based on a target color, which is (R,G,B)= (0.9, 0.9, 0.9) while (1,1,1) corresponds to white, using the DPO objective. Please refer to Figure 7 for an illustration of the preference indicated by DPO. We leveraged 1024 images generated by the pre-trained model to train $f_k$ in each DA iteration. We see that the (regularized) objective went down during DA and the generated images reflected the target density more, on the right side of Figure 3. Please refer to Figure 8 in Appendix for the convergence results.

**Tilt Correction for Image Generation.** The goal of this experiment is data augmentation of images facing the same direction, given only a dataset of rotated images. We used 10000 images of Head CT in Medical MNIST (Lozano, 2017) and augmented it by rotating images up to 40000. Data augmentation of brain images is useful for tumor analysis and anomaly detection (Shen et al., 2024; Fontanella et al., 2024). In each DA iteration, 6400 images were generated to train $f_k$. The results are in Figure 3. For the convergence results, please refer to Figure 12 in Appendix. From the results, we see that the (regularized) objective decreased during DA and the generated images became more preferred.

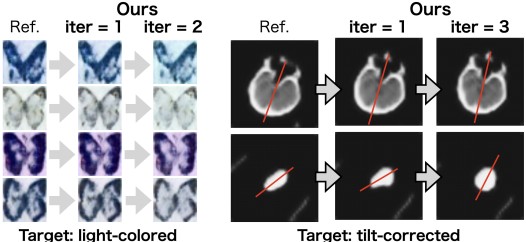

Figure 3: **Left.** Examples of aligned image generation. Our goal was to generate light-colored butterflies. "iter=2": ours with $k = 2$ DA iterations, "iter=2": ours with $k = 1$ DA iteration. "Ref.": samples from $p_{\text{ref}}$. **Right.** Tilt-corrected Head CT image generation. "iter=3": ours with $k = 3$ DA iterations, "iter=1": ours with $k = 1$ DA iteration. "Ref,": samples from $p_{\text{ref}}$.

## 7 CONCLUSION

We proved that the distributional optimization can be solved even if $p_{\text{ref}}$ and $\hat{q}_{\text{opt}}$ are mutilmodal and we only have access to the score, not the density. This setting includes important fine-tuning methodologies for diffusion models: Reinforcement Learning, DPO, and KTO. Our algorithms are based on the DA algorithm and Doob's h-transform technique and it can solve them more directly than previous works. They are guaranteed to be useful for general objective functions, the dual variable converges even in a non-convex setting. While our framework has potential applications in general distributional optimization problems, such as density ratio estimation under covariate shift and climate change tracking, further exploration of these applications is left for future work.

## ACKNOWLEDGMENTS

RK was partially supported by JST CREST (JPMJCR2015). KO is partially supported by JST ACT-X (JP-MJAX23C4). TS was partially supported by JSPS KAKENHI (24K02905, 20H00576) and JST CREST (JPMJCR2115). This research is supported by the National Research Foundation, Singapore, Infocomm Media Development Authority under its Trust Tech Funding Initiative, and the Ministry of Digital Development and Information under the AI Visiting Professorship Programme (award number AIVP-2024-004). Any opinions, findings and conclusions or recommendations expressed in this material are those of the author(s) and do not reflect the views of National Research Foundation, Singapore, Infocomm Media Development Authority, and the Ministry of Digital Development and Information.

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

NOTATIONS

Table 2: List of commonly used symbols

| Symbol | Description |
|---|---|
| $d$ | dimension of the sample space $\mathbb{R}^d$. |
| $\mathcal{B}(\mathbb{R}^d)$ | Borel set on $\mathbb{R}^d$. |
| $\mathcal{P}$ | The space of probability density functions to the Lebesgue measure on $(\mathbb{R}^d, \mathcal{B}(\mathbb{R}^d))$. |
| $F : \mathbb{R}^d : \mathcal{P} \to \mathbb{R}$ | A functional. |
| $\frac{\delta F}{\delta q}(q, x)$ | The functional derivative of $F$. We often abbreviate this as $\frac{\delta F}{\delta q}(q)$. |
| $D_{\mathrm{KL}}(q\|p)$ | Kullback-Leibler divergence between $q$ and $p$. |
| $\mathrm{TV}(q, p)$ | Total Variation between $q$ and $p$. |
| $L(q)$ | The regularized loss $F(q) + \beta D_{\mathrm{KL}}(q\|p_{\mathrm{ref}})$. |
| $\beta$ | The regularization coefficient of $L(q)$. |
| $p_* \in \mathcal{P}$ | The target distribution of pre-training. |
| $p_{\mathrm{ref}} \in \mathcal{P}$ | The output distribution at the final denoising step of the pre-trained model. |
| $q_* = \rho_* p_* \propto \exp(-f_*)p_* \in \mathcal{P}$ | The target distribution of alignment. |
| $\hat{q}_{\mathrm{opt}} = \hat{\rho}_{\mathrm{opt}} p_{\mathrm{ref}} \propto \exp(-\hat{f}_{\mathrm{opt}})p_{\mathrm{ref}} \in \mathcal{P}$ | The unique minimizer of $L(q)$, that corresponds to $q_*$ when $p_{\mathrm{ref}} = p_*$. We assume that $\rho_* \propto \exp(-f_*)$ can be estimated by minimizing $L(q)$, i.e. $\hat{\rho}_{\mathrm{opt}} \simeq \rho_*$. |
| $\hat{q}^{(k)} \propto \exp(-\bar{g}^{(k)})p_{\mathrm{ref}}$ | The optimal solution of the subproblem solved iteratively during Dual Averaging. |
| $q^{(k)} \propto \exp(-f_k)p_{\mathrm{ref}}$ | The implemented density that estimates $\hat{q}^{(k)}$. |
| $\beta'$ | The hyperparameter of Dual Averaging that controls the optimization speed. |
| $u_*$ | The optimal correction term to sample from $q_*$. |
| $u$ | The implemented correction term using the estimated output density ratio $\rho \simeq \hat{\rho}_{\mathrm{opt}}$. |
| $\mathcal{N}(\mu, \sigma)$ | Gaussian with mean $= \mu$ and variance $= \sigma^2$ |
| $T > 0$ | The diffusion time. |
| $\{B_t\}_{t \in [0,T]}$ | Brownian motion on $\mathbb{R}^d$. |
| $\{\bar{X}_t\}_{t \in [0,T]}$ | The Ornstein-Uhlenbeck (OU) process starts from $p_*$. |

| | |
|---|---|
| $\{\bar{X}_t^{\leftarrow}\}_{t \in [0,T]}$ | The reversed Ornstein-Uhlenbeck (ROU) process of $\{\bar{X}_t\}_{t \in [0,T]}$, initialized at $p_T$. |
| $\{X_t^{\leftarrow}\}_{t \in [0,T]}$ | The time-discretized, and score-estimated ROU process of $\{\bar{X}_t\}_{t \in [0,T]}$, initialized as $X_0^{\leftarrow} \sim \mathcal{N}(0, I_d)$. |
| $\{\bar{Y}_t\}_{t \in [0,T]}$ | The Ornstein-Uhlenbeck (OU) process starts from $q_*$. |
| $\{\bar{Y}_t^{\leftarrow}\}_{t \in [0,T]}$ | The reversed Ornstein-Uhlenbeck (ROU) process starts from $q_T$. |
| $\{Y_t^{\leftarrow}\}_{t \in [0,T]}$ | The time-discretized dynamics using $s$ and $u$. |
| $p_t$ | The law of $\bar{X}_t$, and the law of $\bar{X}_{T-t}^{\leftarrow}$. |
| $q_t$ | The law of the OU process $\bar{Y}_t$ corresponding to $q_* = \rho_* p_*$. |
| $h > 0$ | The discretized time step of $\{X_t^{\leftarrow}\}_{t \in [0,T]}$. |
| $L$ | The number of steps of $\{X_t^{\leftarrow}\}_{t \in [0,T]}$. |
| $s : \mathbb{R}^d \times \mathbb{R} \to \mathbb{R}^d$ | A score network to implement $\{X_t^{\leftarrow}\}_{t \in [0,T]}$ |
| $B_F$ | A constant that bounds $\frac{\delta F}{\delta q}$. |
| $L_{\mathrm{TV}}$ | Lipschitz constant of $\frac{\delta F}{\delta q}$. |
| $S_F$ | Smoothness constant of the weaker smoothness of $F$. |
| $\nabla_x f(x)$ | Gradient of $f$ with respect to $x$ |
| $\nabla_x^2 f(x)$ | Hessian matrix of $f$ |
| $\|\cdot\|$ | Euclid norm |
| $\|\cdot\|_\infty$ | $L^\infty$ norm |
| $\|\cdot\|_{\mathrm{op}}$ | The operator norm |
| $\mathbb{E}[\cdot|X]$ | the conditional expectation given a random variable $X$. |
| $\mathbb{E}[\cdot|X = x]$ | the conditional expectation given a random variable $X$, evaluated at $X = x$. |
| $\mathcal{O}$ | Big-O notation |
| $\lesssim, \gtrsim$ | inequalities ignoring constants. |

## A  CONVERGENCE ANALYSIS OF DUAL AVERAGING

**Overview for convex loss**

In Section A.1, we are interested in the convergence of Option 1 when the objective functional $F$ is convex in a distributional sense:

$$F(q) \geq F(q') + \int \frac{\delta F}{\delta q}(q') \mathrm{d}(q - q'), \quad \text{for any } q, q' \in \mathcal{P}.$$

Then, the regularized objective $L(q) = F(q) + \beta D_{\mathrm{KL}}(q\|p_{\mathrm{ref}})$ becomes strongly convex with $\beta > 0$. Formally, we assume that

(i)  $\frac{\delta F}{\delta q}$ is bounded: There exists $B_F > 0$ such that $\|\frac{\delta F}{\delta q}(q)\|_\infty \leq B_F$ for any $q \in \mathcal{P}$,

(ii)  $\frac{\delta F}{\delta q}$ is Lipshitz continuous with respect to the TV distance: There exists $L_{\mathrm{TV}} > 0$ such that $\|\frac{\delta F}{\delta q}(q) - \frac{\delta F}{\delta q}(q')\|_\infty \leq L_{\mathrm{TV}} \mathrm{TV}(q, q')$ for any $q, q' \in \mathcal{P}$.

(iii)  $F$ is convex: $F(q) \geq F(q') + \int \frac{\delta F}{\delta q}(q') \mathrm{d}(q - q')$ for any $q, q' \in \mathcal{P}$,

In the update of Option 1, we iteratively use the distributional proximal operator. We minimize

$$\mathrm{E}_q\left[\sum_{j=1}^{k} \frac{2j}{k(k+1)} g^{(j)}\right] + \beta D_{\mathrm{KL}}(q\|p_{\mathrm{ref}}) + \beta' \frac{2}{k} D_{\mathrm{KL}}(q\|p_{\mathrm{ref}}) \tag{5}$$

by the minimizer $\hat{q}^{(k)}$, where $g^{(k)}$ denotes $\frac{\delta F}{\delta q}(q^{(k)})$ with $q^{(k)} \simeq \hat{q}^{(k)}$ ($q^{(k)}$ is constructed by neural networks and samples from $p_{\mathrm{ref}}$, while we do not directly compute $q^{(k)}$.). Intuitively, with $k \to \infty$, we hope that $q$ is almost converged around the minimizer $\hat{q}_{\mathrm{opt}}$, i.e. $q \simeq \hat{q}_{\mathrm{opt}}$. Then, from the differentiability of $F$, each $\mathbb{E}_q[g^{(j)}]$ in the first term would be a linear functional that well approximates $F(q^{(j)})$, with suficiently large $j$. In addition, $\beta' \frac{2}{k} D_{\mathrm{KL}}(q\|p_{\mathrm{ref}})$ vanishes, so the equation (5) is roughly written as

$$\sum_{j=1}^{k} \frac{2j}{k(k+1)} \left[\mathrm{E}_q\left[g^{(j)}\right] + \beta D_{\mathrm{KL}}(q\|p_{\mathrm{ref}})\right] + \beta' \frac{2}{k} D_{\mathrm{KL}}(q\|p_{\mathrm{ref}})$$

$$\simeq \sum_{j=1}^{k} \frac{2j}{k(k+1)} (F(q^{(j)}) + \beta D_{\mathrm{KL}}(q\|p_{\mathrm{ref}})) + \beta' \frac{2}{k} D_{\mathrm{KL}}(q\|p_{\mathrm{ref}})$$

$$\simeq \sum_{j=1}^{k} \frac{2j}{k(k+1)} (F(\hat{q}^{(j)}) + \beta D_{\mathrm{KL}}(\hat{q}^{(j)}\|p_{\mathrm{ref}})).$$

This is the weighted average of the regularized losses $L(q) = F(q) + \beta D_{\mathrm{KL}}(q\|p_{\mathrm{ref}})$. From this concept, we will show that the convergence represented as

$$\left[\text{Weighted average of } L(\hat{q}^{(k)})\right] = [\text{Weighted average of } L(\hat{q}_{\mathrm{opt}})] + \mathcal{O}\left(\frac{1}{k}\right).$$

**Overview for nonconvex loss**

In Section A.2, we are interested in the case that $F$ is not necessarily convex (mainly) in Option 2. Alternatively, we use the assumption that $F$ is smooth in terms of KL-divergence:

(ii)'  (The weaker smoothness derived from (ii)). There exists $S_F \geq 0$ such that

$$F(q) \leq F(q') + \int \frac{\delta F}{\delta q}(q') \mathrm{d}(q - q') + \frac{S_F}{2} D_{\mathrm{KL}}(q\|q') \quad \text{for any} \quad q, q' \in \mathcal{P}, \tag{6}$$

This is induced by Lipschitz continuity of $\frac{\delta F}{\delta q}$ ((ii) of Assumption 1 and 2) and Pinsker's inequality. When the inner-loop error is ignored, it is possible to prove convergence using only the smoothness (6) instead of Lipschitz continuity of $\frac{\delta F}{\delta q}$. Please note that KL-divergence would be the "quadratic

term" in the ordinary definition of smoothness. Following the theoretical analysis of standard non-convex optimization, our goal is to show that the "derivative" of the (regularized) objective $L(q)$ goes to zero in the form of functional derivative:

$$\frac{\delta L}{\delta q}(\hat{q}^{(k)}, x) \to 0 \quad \text{(up to constant w.r.t. } x.)$$

Please note that we can ignore $\frac{\delta L}{\delta q}(\hat{q}^{(k)}, x)$ if it is a constant because of the definition of the functional derivatives.

At the first step, both in Option 1 and 2, the regularized objective $L(q)$ (roughly) monotonically decrease during the Dual Averaging. Ignoring some terms, we want to show that

$$L(\hat{q}^{(k+1)}) - L(\hat{q}^{(k)}) \lesssim -D_{\mathrm{KL}}(\hat{q}^{(k)} \| \hat{q}^{(k+1)}) < 0.$$

Next, by taking a telescoping sum, we obtain that

$$\frac{1}{K} \sum_{k=1}^{K} (\text{weight})_k D_{\mathrm{KL}}(\hat{q}^{(k)} \| \hat{q}^{(k+1)}) = \mathcal{O}\left(\frac{1}{K}\right).$$

From the above equation, it is immediately shown that

$$\min_{k=1,\ldots,K} (\text{weight})_k D_{\mathrm{KL}}(\hat{q}^{(k)} \| \hat{q}^{(k+1)}) = \mathcal{O}\left(\frac{1}{K}\right),$$

where $(\text{weight})_k \simeq k$ in the Option 1, while $(\text{weight})_k \simeq 1$ in the Option 2. Therefore, when we choose the Option 2, it holds that

$$\min_{k=1,\ldots,K} D_{\mathrm{KL}}(\hat{q}^{(k)} \| \hat{q}^{(k+1)}) = \mathcal{O}\left(\frac{1}{K}\right)$$

Finally, we get the conclusion $\frac{\delta L(\hat{q}^{(k)})}{\delta q} \to 0$ (up to constant w.r.t. $x$) because $D_{\mathrm{KL}}(\hat{q}^{(k)} \| \hat{q}^{(k+1)})$ is the approximate "moment generation function" of $\frac{\delta L(\hat{q}^{(k)})}{\delta q}$; when $k \to \infty$,

$$\min_{k=1,\ldots,K} \left( \text{"Variance" of } \frac{\delta L(\hat{q}^{(k)})}{\delta q} \right) \simeq \min_{k=1,\ldots,K} D_{\mathrm{KL}}(\hat{q}^{(k)} \| \hat{q}^{(k+1)}) = \mathcal{O}\left(\frac{1}{K}\right).$$

This result aligns with the result for standard, non-distributinal Dual Averaging (Liu et al., 2023a), $\min_{k=1,\ldots,K} \|(\text{gradient of the objective})_k\|^2 = \mathcal{O}(1/K)$ because the variance is the second moment.

## A.1 WHEN $F$ IS CONVEX

We will show that, when $F$ is convex in the distributional sense, the convergence of the Option 1 can be written as

$$\left[\text{Weighted average of } L(\hat{q}^{(k)})\right] - \left[\text{Weighted average of } L(\hat{q}_{\mathrm{opt}})\right] = \mathcal{O}\left(\frac{1}{K}\right).$$

We require $\beta$ to be positive to make the regularized objective $F(q) + \beta D_{\mathrm{KL}}(q \| p_{\mathrm{ref}})$ strongly convex. In addition, one of the necessary conditions $\beta' \geq \beta$ implies that the "learning rate" should be controlled to converge.

**Theorem** (restated - Theorem 1). *Assume that $F$ is convex, $|\frac{\delta F}{\delta q}| \leq B_F$, $\frac{\delta F}{\delta q}$ is $L_{\mathrm{TV}}$-Lipshitz with respect to $q$ in TV distance, $\mathrm{TV}(\hat{q}^{(k)}, q^{(k)}) \leq \epsilon_{\mathrm{TV}}$, and $\beta' \geq \beta$. Then,*

$$\frac{2}{K(K+1)} \sum_{k=1}^{K} k \left[ F(\hat{q}^{(k)}) + \beta D_{\mathrm{KL}}(\hat{q}^{(k)} \| p_{\mathrm{ref}}) - F(\hat{q}_{\mathrm{opt}}) - \beta D_{\mathrm{KL}}(\hat{q}_{\mathrm{opt}} \| p_{\mathrm{ref}}) \right]$$

$$\leq \frac{2}{K(K+1)} \sum_{k=1}^{K} k \left[ 2L_{\mathrm{TV}}\epsilon_{\mathrm{TV}} + \mathbb{E}_{\hat{q}^{(k)}}[g^{(k)}] - \mathbb{E}_{\hat{q}_{\mathrm{opt}}}[g^{(k)}] + \beta(D_{\mathrm{KL}}(\hat{q}^{(k)} \| p_{\mathrm{ref}}) - D_{\mathrm{KL}}(\hat{q}_{\mathrm{opt}} \| p_{\mathrm{ref}})) \right]$$

$$\leq 2L_{\mathrm{TV}}\epsilon_{\mathrm{FD}} + \frac{2}{K(K+1)} \left[ B_F + \beta(K+3)D_{\mathrm{KL}}(\hat{q}_{\mathrm{opt}} \| p_{\mathrm{ref}}) + \mathbb{E}_{\hat{q}_{\mathrm{opt}}}[g^{(1)}] + \frac{B_F^2 K}{\beta} \right].$$

First, we want to show that the $D_{\mathrm{KL}}$ regularization term plays a role of the "quadratic" regularization, which makes the regularized objective $F(q) + \beta D_{\mathrm{KL}}(q\|p_{\mathrm{ref}})$ strongly convex. We observe that the sum of the linear (convex and concave) objective and the $D_{\mathrm{KL}}$ regularization term

$$\tilde{F}(q) = \mathrm{E}_q[r(x)] + \beta D_{\mathrm{KL}}(q\|p)$$

is strongly convex in the distributional sense. In particular, if $q$ is the minimizer of $\tilde{F}$, then

$$\tilde{F}(q) + \frac{\beta}{2}\mathrm{TV}(q, q')^2 \leq \tilde{F}(q') \text{ for all } q' \in \mathcal{P}.$$

This result is useful to prove Lemma 2, which controls the regularized and linearized objective $\mathbb{E}_q[\frac{\delta F}{\delta q}] + \beta D_{\mathrm{KL}}(q\|p_{\mathrm{ref}})$:

**Lemma 1** (Nitanda et al. (2021)). *$\tilde{F} : \mathcal{P} \to \mathbb{R}$, $\tilde{F}(q) = \mathrm{E}_q[r(x)] + \beta D_{\mathrm{KL}}(q\|p)$. Assume that $r$ is bounded. We put $r_1 := \frac{dq}{dp}$, $r_2' := \frac{dq'}{dp}$,*

$$\tilde{F}(q) \leq \tilde{F}(q') + \mathrm{E}_q[(r_1 - r_2)(r(x) + \beta \log r_1)] - \frac{\beta}{2}\|r_1 - r_2\|_{L^1(p)}^2.$$

*Especially, if $q \propto \exp(-\frac{r}{\beta})p$,*

$$\tilde{F}(q) \leq \tilde{F}(q') - \frac{\beta}{2}\|r_1 - r_2\|_{L^1(p)}^2.$$

*Proof.* Omitted. See Nitanda et al. (2021). We use Pinsker's inequality $\mathrm{TV}(q, q')^2 \lesssim D_{\mathrm{KL}}(q\|q')$. $\qed$

Given that $F(q) + \beta D_{\mathrm{KL}}(q\|p_{\mathrm{ref}})$ is strongly convex, let us prove Theorem 1:

*Proof.* The main parts of the proof follow existing papers (Nitanda et al., 2021; Nishikawa et al., 2022). The key difference lies in that we are not using Langevin sampler in the inner loop, which slightly changes how the error of $q^{(k)}$ is handled.

We begin by linearizing the weighted sum of the losses. $g^{(k)}$ denotes $\frac{\delta F}{\delta q}(q^{(k)})$, and $\hat{q}_{\mathrm{opt}}$ denotes the minimizer of $F$. From the convexity of $F$,

$$\frac{2}{K(K+1)}\sum_{k=1}^{K} k\left[F(\hat{q}^{(k)}) + \beta D_{\mathrm{KL}}(\hat{q}^{(k)}\|p_{\mathrm{ref}}) - F(\hat{q}_{\mathrm{opt}}) - \beta D_{\mathrm{KL}}(\hat{q}_{\mathrm{opt}}\|p_{\mathrm{ref}})\right]$$

$$\leq \frac{2}{K(K+1)}\sum_{k=1}^{K} k\left[\int \frac{\delta F}{\delta q}(\hat{q}^{(k)})d(\hat{q}^{(k)} - \hat{q}_{\mathrm{opt}}) + \beta(D_{\mathrm{KL}}(\hat{q}^{(k)}\|p_{\mathrm{ref}}) - D_{\mathrm{KL}}(\hat{q}_{\mathrm{opt}}\|p_{\mathrm{ref}}))\right]$$

$$\leq \frac{2}{K(K+1)}\sum_{k=1}^{K} k\left[\left|\int \frac{\delta F}{\delta q}(\hat{q}^{(k)}) - \frac{\delta F}{\delta q}(q^{(k)})d(\hat{q}^{(k)} - \hat{q}_{\mathrm{opt}})\right| + \mathbb{E}_{\hat{q}^{(k)}}[g^{(k)}] - \mathbb{E}_{\hat{q}_{\mathrm{opt}}}[g^{(k)}]\right.$$

$$\left. + \beta(D_{\mathrm{KL}}(\hat{q}^{(k)}\|p_{\mathrm{ref}}) - D_{\mathrm{KL}}(\hat{q}_{\mathrm{opt}}\|p_{\mathrm{ref}}))\right].$$

Here, we bound the estimation error of $g^{(k)} = \frac{\delta F}{\delta q}(q^{(k)})$ using the Lipschitz continuity of $\frac{\delta F}{\delta q}$ with respect to $q$:

$$\left|\int \frac{\delta F}{\delta q}(\hat{q}^{(k)}) - g^{(k)}d(\hat{q}^{(k)} - \hat{q}_{\mathrm{opt}})\right|$$

$$\leq \mathrm{TV}(\hat{q}^{(k)}, \hat{q}_{\mathrm{opt}})\sup_x\left|\frac{\delta F}{\delta q}(\hat{q}^{(k)}, x) - g^{(k)}(x)\right|$$

$$\leq 2L_{\mathrm{q}}\epsilon_{\mathrm{TV}}.$$

Then we obtain

$$\frac{2}{K(K+1)} \sum_{k=1}^{K} k \left[ F(\hat{q}^{(k)}) + \beta D_{\mathrm{KL}}(\hat{q}^{(k)} \| p_{\mathrm{ref}}) - F(\hat{q}_{\mathrm{opt}}) - \beta D_{\mathrm{KL}}(\hat{q}_{\mathrm{opt}} \| p_{\mathrm{ref}}) \right]$$

$$\leq \frac{2}{K(K+1)} \sum_{k=1}^{K} k \left[ 2 L_{\mathrm{q}} \epsilon_{\mathrm{TV}} + \mathbb{E}_{\hat{q}^{(k)}}[g^{(k)}] - \mathbb{E}_{\hat{q}_{\mathrm{opt}}}[g^{(k)}] \right.$$

$$\left. + \beta (D_{\mathrm{KL}}(\hat{q}^{(k)} \| p_{\mathrm{ref}}) - D_{\mathrm{KL}}(\hat{q}_{\mathrm{opt}} \| p_{\mathrm{ref}})) \right]. \tag{7}$$

This implies that it is sufficient to bound the weighted sum of the (regularized) linearized objectives $\mathbb{E}_{\hat{q}^{(k)}}[g^{(k)}] + \beta (D_{\mathrm{KL}}(\hat{q}^{(k)} \| p_{\mathrm{ref}}), \ k = 1, ..., K$.

In each update of Option 1, $\hat{q}^{(k+1)}$ is obtained by maximizing

$$V_{k+1}(q) = -\mathrm{E}_q \left[ \sum_{j=1}^{k} j g^{(j)} \right] - \frac{\beta k(k+1)}{2} D_{\mathrm{KL}}(q \| p_{\mathrm{ref}}) - \beta'(k+1) D_{\mathrm{KL}}(q \| p_{\mathrm{ref}}).$$

We also define

$$r_*^{(k+1)} = \frac{\mathrm{d}\hat{q}^{(k+1)}}{\mathrm{d}p_{\mathrm{ref}}}, \quad V_{k+1}^* = V_{k+1}(\hat{q}^{(k+1)}).$$

Then, we can show that $V_k^*$ has the following recursive relation. Lemma 2 approximately implies

$$k \mathbb{E}_{\hat{q}^{(k)}}[g^{(k)}] + k \beta D_{\mathrm{KL}}(\hat{q}^{(k+1)} \| p_{\mathrm{ref}})) \lesssim V_k^* - V_{k+1}^* + \mathcal{O}(1),$$

then, summing from $k = 1$ to $K$, we roughly obtain,

$$\frac{2}{K(K+1)} \left[ \sum_{k=1}^{K} k (\mathbb{E}_{\hat{q}^{(k)}}[g^{(k)}] + \beta D_{\mathrm{KL}}(\hat{q}^{(k)} \| p_{\mathrm{ref}})) + V_{K+1}^* \right] \lesssim \mathcal{O}\left(\frac{1}{K}\right). \tag{8}$$

So, as long as we prove Lemma 2, we can connect this inequality (8) to the inequality (7) because $V_{K+1}^* \gtrsim -\sum_{k=1}^{K} k (\mathbb{E}_{\hat{q}_{\mathrm{opt}}}[g^{(k)}] + \beta D_{\mathrm{KL}}(\hat{q}_{\mathrm{opt}} \| p_{\mathrm{ref}}))$:

**Lemma 2.** *For $k \geq 1$,*

$$V_{k+1}^* \leq V_k^* - k \mathbb{E}_{\hat{q}^{(k)}}[g^{(k)}] - (\beta k + \beta') D_{\mathrm{KL}}(\hat{q}^{(k+1)} \| p_{\mathrm{ref}}) + \frac{B_F^2 k}{\beta(k-1) + 2\beta'}. \tag{9}$$

For the proof of Lemma 2, please refer to Section A.1.1, in which we use Lemma 1.

From here, we will rigorously explain the result derived from Lemma 2. When $k = 1$, because $\|g^{(k)}\|_\infty \leq B_F$,

$$V_1^* \leq B_F - \beta D_{\mathrm{KL}}(\hat{q}^{(1)} \| p_{\mathrm{ref}}).$$

By taking a telescoping sum of (9),

$$V_{K+1}^* \leq B_F - \sum_{k=1}^{K} k \left( \mathbb{E}_{\hat{q}^{(k)}}[g^{(k)}] + \beta D_{\mathrm{KL}}(\hat{q}^{(k)} \| p_{\mathrm{ref}}) \right) - [\beta(K+1) + \beta'] D_{\mathrm{KL}}(\hat{q}^{(k+2)} \| p_{\mathrm{ref}})$$

$$+ \sum_{k=1}^{K} \frac{B_F^2 k}{\beta(k-1) + 2\beta'}$$

$$\leq B_F - \sum_{k=1}^{K} k \left( \mathbb{E}_{\hat{q}^{(k)}}[g^{(k)}] + \beta D_{\mathrm{KL}}(\hat{q}^{(k)} \| p_{\mathrm{ref}}) \right) + \frac{B_F^2 K}{\beta}, \tag{10}$$

where we used $\beta' \geq \beta$ in the last inequality. On the other hand, we see that

$$V_{K+1}^* \geq - \mathbb{E}_{\hat{q}_{\mathrm{opt}}} \left[ \sum_{k=1}^{K} k g^{(k)} \right] - \frac{\beta K(K+1) + 2\beta'(K+1)}{2} D_{\mathrm{KL}}(\hat{q}_{\mathrm{opt}} \| p_{\mathrm{ref}})$$

$$= - \mathbb{E}_{\hat{q}_{\mathrm{opt}}} \left[ \sum_{k=1}^{K} k (g^{(k)} + \beta D_{\mathrm{KL}}(\hat{q}_{\mathrm{opt}} \| p_{\mathrm{ref}})) \right] - \beta'(K+1) D_{\mathrm{KL}}(\hat{q}_{\mathrm{opt}} \| p_{\mathrm{ref}}) \tag{11}$$

because $V_{K+2}^*$ is the maximal value. Combining the upper bound (Eq. 10) and the lower bound (Eq. 11),

$$\sum_{k=1}^{K} k \left( \mathbb{E}_{\hat{q}^{(k)}}[g^{(k)}] - \mathbb{E}_{\hat{q}_{\text{opt}}}[g^{(k)}] + \beta(D_{\text{KL}}(\hat{q}^{(k)}\|p_{\text{ref}}) - D_{\text{KL}}(\hat{q}_{\text{opt}}\|p_{\text{ref}})) \right)$$

$$\leq B_F + \beta'(K+1)D_{\text{KL}}(\hat{q}_{\text{opt}}\|p_{\text{ref}}) + \frac{B_F^2 K}{\beta}.$$

Finally we get the convergence rate:

$$\frac{2}{K(K+1)} \sum_{k=1}^{K} k \left[ F(\hat{q}^{(k)}) + \beta D_{\text{KL}}(\hat{q}^{(k)}\|p_{\text{ref}}) - F(\hat{q}_{\text{opt}}) - \beta D_{\text{KL}}(\hat{q}_{\text{opt}}\|p_{\text{ref}}) \right]$$

$$\leq \frac{2}{K(K+1)} \sum_{k=1}^{K} k \left[ 2L_{\text{TV}}\epsilon_{\text{TV}} + \mathbb{E}_{\hat{q}^{(k)}}[g^{(k)}] - \mathbb{E}_{\hat{q}_{\text{opt}}}[g^{(k)}] + \beta(D_{\text{KL}}(\hat{q}^{(k)}\|p_{\text{ref}}) - D_{\text{KL}}(\hat{q}_{\text{opt}}\|p_{\text{ref}})) \right]$$

$$\leq 2L_{\text{TV}}\epsilon_{\text{FD}} + \frac{2}{K(K+1)} \left[ B_F + \beta'(K+1)D_{\text{KL}}(\hat{q}_{\text{opt}}\|p_{\text{ref}}) + \frac{B_F^2 K}{\beta} \right].$$

This concludes the assertion. $\qquad\square$

### A.1.1 PROOF OF LEMMA 2

*Proof.* We can calculate the relation between $V_{k+1}^* = V_{k+1}(\hat{q}^{(k+1)})$ and $V_k(\hat{q}^{(k+1)})$ as

$$V_{k+1}^* = -\mathbb{E}_{\hat{q}^{(k+1)}} \left[ \sum_{j=1}^{k} jg^{(j)} \right] - \frac{(k+1)(\beta k + 2\beta')}{2} D_{\text{KL}}(\hat{q}^{(k+1)}\|p_{\text{ref}})$$

$$= -\mathbb{E}_{\hat{q}^{(k+1)}} \left[ \sum_{j=1}^{k-1} jg^{(j)} \right] - \frac{\beta k(k-1)}{2} D_{\text{KL}}(\hat{q}^{(k+1)}\|p_{\text{ref}}) - \beta' k D_{\text{KL}}(\hat{q}^{(k+1)}\|p_{\text{ref}})$$

$$\quad - k\mathbb{E}_{\hat{q}^{(k+1)}}[g^{(k)}] - (\beta k + \beta')D_{\text{KL}}(\hat{q}^{(k+1)}\|p_{\text{ref}})$$

$$= V_k(\hat{q}^{(k+1)}) - k\mathbb{E}_{\hat{q}^{(k)}}[g^{(k)}] + k(\mathbb{E}_{\hat{q}^{(k)}}[g^{(k)}] - \mathbb{E}_{\hat{q}^{(k+1)}}[g^{(k)}]) - (\beta k + \beta')D_{\text{KL}}(\hat{q}^{(k+1)}\|p_{\text{ref}}),$$
$$(12)$$

where we used the definitions of $V_k$ and $V_{k+1}$. Next, we upper bound the RHS of Eq. (12) using Lemma 1 about the convexity of $D_{\text{KL}}$:

(RHS of Eq. (12))

$$\leq V_k^* - \frac{\beta k(k-1) + 2\beta' k}{4} \|r_*^{(k+1)} - r_*^{(k)}\|_{L^1(p_{\text{ref}})}^2 - k\mathbb{E}_{\hat{q}^{(k)}}[g^{(k)}] + k(\mathbb{E}_{\hat{q}^{(k)}}[g^{(k)}] - \mathbb{E}_{\hat{q}^{(k+1)}}[g^{(k)}])$$

$$\quad - (\beta k + \beta')D_{\text{KL}}(\hat{q}^{(k+1)}p_{\text{ref}}) \quad (\because \text{ the second equation in Lemma 1 and the optimality of } V_k^*)$$

$$\leq V_k^* - \frac{\beta k(k-1) + 2\beta' k}{4} \|r_*^{(k+1)} - r_*^{(k)}\|_{L^1(p_{\text{ref}})}^2 - k\mathbb{E}_{\hat{q}^{(k)}}[g^{(k)}] + k \left| \mathbb{E}_{\hat{q}^{(k)}}[g^{(k)}] - \mathbb{E}_{\hat{q}^{(k+1)}}[g^{(k)}] \right|$$

$$\quad - \beta(k+1)D_{\text{KL}}(\hat{q}^{(k+1)}\|p_{\text{ref}})$$

$$\leq V_k^* - \frac{\beta k(k-1) + 2\beta' k}{4} \|r_*^{(k+1)} - r_*^{(k)}\|_{L^1(p_{\text{ref}})}^2 - k\mathbb{E}_{\hat{q}^{(k)}}[g^{(k)}] + B_F k \|r_*^{(k+1)} - r_*^{(k)}\|_{L^1(p_{\text{ref}})}$$

$$\quad - \beta(k+1)D_{\text{KL}}(p_*^{(t+1)}\|q) \quad (\because \|g^{(k)}\|_\infty \leq B_F)$$

$$\leq V_k^* - k\mathbb{E}_{\hat{q}^{(k)}}[g^{(k)}] - \beta(k+1)D_{\text{KL}}(\hat{q}^{(k+1)}\|p_{\text{ref}}) + \frac{B_F^2 k}{\beta(k-1) + 2\beta'},$$

where we used the arithmetic-geometric mean inequality in the last inequality. This concludes the proof. $\qquad\square$

## A.2 Convergence proof for non-convex loss $F$

Here, we give the proof of Theorem 2 and Corollary 1. In this section, we always assume Assumption 2 holds. We are mainly interested in the property that $F$ is smooth with respect to KL-divergence instead of convexity:

(ii)' (A weaker version of (ii)). There exists $S_F \geq 0$ such that

$$F(q) \leq F(q') + \int \frac{\delta F}{\delta q}(q')\mathrm{d}(q - q') + \frac{S_F}{2}D_{\mathrm{KL}}(q\|q') \quad \text{for any} \quad q, q' \in \mathcal{P}. \quad (13)$$

This property can be derived from (ii) in Assumption 2 that is Lipschitz continuity of $\frac{\delta F}{\delta q}$:

**Lemma 3.** *Assume that $\frac{\delta F}{\delta q}$ is Lipshitz continuous with respect to the TV distance: There exists $L_{\mathrm{TV}} > 0$ such that $\|\frac{\delta F}{\delta q}(q) - \frac{\delta F}{\delta q}(q')\|_\infty \leq L_{\mathrm{TV}}\mathrm{TV}(q, q')$ for any $q, q' \in \mathcal{P}$. Then,*

$$F(q) \leq F(q') + \int \frac{\delta F}{\delta q}(q')\mathrm{d}(q - q') + \frac{L_{\mathrm{TV}}}{2}\mathrm{TV}(q, q')^2 \quad \text{for any} \quad q, q' \in \mathcal{P}.$$

*From Pinsker's inequality,*

$$F(q) \leq F(q') + \int \frac{\delta F}{\delta q}(q')\mathrm{d}(q - q') + L_{\mathrm{TV}}D_{\mathrm{KL}}(q\|q') \quad \text{for any} \quad q, q' \in \mathcal{P}.$$

*Proof.* The proof is almost identical to the proof of $L_{\mathrm{TV}}$-smoothness commonly discussed in the context of standard optimization. For $q, q' \in \mathcal{P}$, we define $q_t = q' + t(q - q') \in \mathcal{P}$. Then,

$$\begin{aligned}
&F(q) - F(q') - \int \frac{\delta F}{\delta q}(q')\mathrm{d}(q - q') \\
&= \int_{t=0}^{t=1} \int \frac{\delta F}{\delta q}(q_t)\mathrm{d}(q - q')\mathrm{d}t - \int \frac{\delta F}{\delta q}(q')\mathrm{d}(q - q') \\
&= \int_{t=0}^{t=1} \int \left[\frac{\delta F}{\delta q}(q_t) - \frac{\delta F}{\delta q}(q')\right]\mathrm{d}(q - q')\mathrm{d}t \\
&\leq \int_{t=0}^{t=1} tL_{\mathrm{TV}}\mathrm{TV}(q, q')^2\mathrm{d}t \quad (L_{\mathrm{TV}}\text{-Lipschitz continuity of } \frac{\delta F}{\delta q}) \\
&= \frac{L_{\mathrm{TV}}}{2}\mathrm{TV}(q, q')^2,
\end{aligned}$$

while we used the fundamental theorem of calculus in the first equation and the assumption in the inequality. $\square$

We emphasize that when the inner-loop error is ignored, it is possible to prove convergence using only the smoothness (13) instead of (ii) in Assumption 2. Therefore, we will use the notation $S_F$ for the parts that can be derived using Assumption (ii)' instead of Assumption (ii).

Under the assumptions, our goal is to show that the functional derivative of $L(q) = F(q) + \beta D_{\mathrm{KL}}(q\|p_{\mathrm{ref}})$ goes to a constant:

$$\frac{\delta L}{\delta q} \to 0 \quad \text{(up to constant w.r.t. } x)$$

We prepare the following Lemma about the convexity of $D_{\mathrm{KL}}$:

**Lemma 4.** *The following equations hold:*

(i) $D_{\mathrm{KL}}(q\|p) = D_{\mathrm{KL}}(q'\|p) + \int \frac{\delta}{\delta q}D_{\mathrm{KL}}(q'\|p)\mathrm{d}(q - q') + D_{\mathrm{KL}}(q\|q')$,

(ii) *Let $r : \mathbb{R}^d \to \mathbb{R}$ be a smooth function. For $\tilde{F}(q) = \mathbb{E}_q[r(x)] + D_{\mathrm{KL}}(q\|p)$, $q' = \exp(-r)q$, it holds that*
$$\tilde{F}(q) = \tilde{F}(q') + D_{\mathrm{KL}}(q\|q').$$
*$\tilde{F}(q)$ is uniquely minimized at $q = q'$.*

We omit its proof because it is straight forward. Lemma 4 implies that $D_{\mathrm{KL}}$ also plays a important role as a "quadratic" penalty term of the proximal operator whose output is

$$\underset{q \in \mathcal{P}}{\operatorname{argmin}} \left\{ H(q) + D_{\mathrm{KL}}(q \| p_{\mathrm{ref}}) \right\}, \text{ for any functional } H.$$

This property allows for the use of standard nonconvex convergence analysis of Dual Averaging based on the proximal operator (Liu et al., 2023a).

Roughly speaking, the intermediate goal is to show that $L(q)$ monotonically decreases in each $k$th iteration:

$$L(\hat{q}^{(k+1)}) - L(\hat{q}^{(k)}) \lesssim 0.$$

By the weaker smoothness of $F$ (Eq.(13)), the left hand side is approximately bounded as

$$L(\hat{q}^{(k+1)}) - L(\hat{q}^{(k)}) \lesssim \int \frac{\delta F}{\delta q}(\hat{q}^{(k)}) \mathrm{d}(\hat{q}^{(k+1)} - \hat{q}^{(k)}) + D_{\mathrm{KL}}(\hat{q}^{(k+1)} \| \hat{q}^{(k)})$$

$$\lesssim \int \frac{\delta L}{\delta q}(\hat{q}^{(k)}) \mathrm{d}(\hat{q}^{(k+1)} - \hat{q}^{(k)}) + D_{\mathrm{KL}}(\hat{q}^{(k+1)} \| \hat{q}^{(k)}), \qquad (14)$$

ignoring various minor terms and constants. To bound the right hand side in (14), we show the following inequality regarding the Option 2 using Lemma 4.

**Lemma 5.** *Assume that* $\mathrm{TV}(\hat{q}^{(k)}, q^{(k)}) \le \epsilon_{\mathrm{TV}}/2$ *for all* $k$. *Then, the Option 2 achieves*

$$\int \frac{\delta L}{\delta q}(q^{(k)}) \mathrm{d}(\hat{q}^{(k+1)} - \hat{q}^{(k)})$$

$$\le B_F \epsilon_{\mathrm{TV}} + \frac{\beta'}{k} \left( D_{\mathrm{KL}}(\hat{q}^{(k)} \| p_{\mathrm{ref}}) - D_{\mathrm{KL}}(\hat{q}^{(k+1)} \| p_{\mathrm{ref}}) - k D_{\mathrm{KL}}(\hat{q}^{(k+1)} \| \hat{q}^{(k)}) - (k+1) D_{\mathrm{KL}}(\hat{q}^{(k)} \| \hat{q}^{(k+1)}) \right)$$

*Proof.* In the Option 2, $\hat{q}^{(k+1)}$ minimizes

$$r_{k+1}(q) := \sum_{j=1}^{k} \int j \frac{\delta L}{\delta q}(q^{(j)}) \mathrm{d}(q - \hat{q}^{(j)}) + \beta'(k+1) D_{\mathrm{KL}}(q \| p_{\mathrm{ref}}).$$

We interpret that Option 2 computes the proximal operator of the weighted sum of $\frac{\delta L}{\delta q}$ and this concept is justified by Lemma 4. By the definition of $r_k$ and $r_{k+1}$, it holds that

$$r_{k+1}(q) = r_k(q) + \int k \frac{\delta L}{\delta q}(q^{(k)}) \mathrm{d}(q - \hat{q}^{(k)}) + \beta' D_{\mathrm{KL}}(q \| p_{\mathrm{ref}}). \qquad (15)$$

From Lemma 4, we also have

$$r_k(q) - r_k(\hat{q}^{(k)}) = \beta' k D_{\mathrm{KL}}(q \| \hat{q}^{(k)}), \text{ for all } q \in \mathcal{P} \qquad (16)$$

because $r_k(q)$ is just the sum of the linear functional of $q$ and the KL-divergence ignoring the constant. Letting $q = \hat{q}^{(k+1)}$ and we obtain

$$0 \le r_k(\hat{q}^{(k+1)}) - r_k(\hat{q}^{(k)}) - \beta' k D_{\mathrm{KL}}(\hat{q}^{(k+1)} \| \hat{q}^{(k)}) \qquad (17)$$

$$= r_{k+1}(\hat{q}^{(k+1)}) - \int k \frac{\delta L}{\delta q}(q^{(k)}) \mathrm{d}(\hat{q}^{(k+1)} - \hat{q}^{(k)}) - \beta' D_{\mathrm{KL}}(\hat{q}^{(k+1)} \| p_{\mathrm{ref}})$$

$$- r_k(\hat{q}^{(k)}) - \beta' k D_{\mathrm{KL}}(\hat{q}^{(k+1)} \| \hat{q}^{(k)}).$$

In the last equality, we decomposed the sum using the equation (15). Thus,

$$\int k \frac{\delta L}{\delta q}(q^{(k)}) \mathrm{d}(\hat{q}^{(k+1)} - \hat{q}^{(k)})$$

$$\le r_{k+1}(\hat{q}^{(k+1)}) - r_k(\hat{q}^{(k)}) - \beta' D_{\mathrm{KL}}(\hat{q}^{(k+1)} \| p_{\mathrm{ref}}) - \beta' k D_{\mathrm{KL}}(\hat{q}^{(k+1)} \| \hat{q}^{(k)}).$$

By using the same argument as Eq. (16) for $k \leftarrow k+1$, we have

$$r_{k+1}(\hat{q}^{(k+1)}) + \beta'(k+1) D_{\mathrm{KL}}(\hat{q}^{(k)} \| \hat{q}^{(k+1)}) = r_{k+1}(\hat{q}^{(k)})$$

$$= r_k(\hat{q}^{(k)}) + \beta' D_{\mathrm{KL}}(\hat{q}^{(k)} \| p_{\mathrm{ref}}).$$

Substituting this relation to Eq. (17) and noticing $|\frac{\delta F}{\delta q}| \le B_F$, we obtain the assertion. $\qquad \square$

We also show the result for Option 1 that is similar to Lemma 5:

**Lemma 6.** *Assume that* $\mathrm{TV}(\hat{q}^{(k)}, q^{(k)}) \leq \epsilon_{\mathrm{TV}}/2$ *for all* $k$. *Then, Algorithm 1 achieves*

$$\int \frac{\delta F}{\delta q}(q^{(k)})\mathrm{d}(\hat{q}^{(k+1)} - \hat{q}^{(k)})$$

$$\leq B_F \epsilon_{\mathrm{TV}} + \frac{\beta k + \beta'}{k}\left(D_{\mathrm{KL}}(\hat{q}^{(k)}\|p_{\mathrm{ref}}) - D_{\mathrm{KL}}(\hat{q}^{(k+1)}\|p_{\mathrm{ref}})\right)$$

$$- \left(\frac{\beta(k-1) + 2\beta'}{2}\right)D_{\mathrm{KL}}(\hat{q}^{(k+1)}\|\hat{q}^{(k)}) - \left(\frac{\beta(k+1) + 2\beta'(1+1/k)}{2}\right)D_{\mathrm{KL}}(\hat{q}^{(k)}\|\hat{q}^{(k+1)})$$

*Proof.* Please refer to Appendix A.2.1. The proof is almost identical to the proof of Lamma 5. $\square$

We mentioned that our rough intermediate goal was to show that $L(q)$ monotonically decreases in each $k$th iteration:

$$L(\hat{q}^{(k+1)}) - L(\hat{q}^{(k)}) \lesssim 0.$$

Rigorously, we will show that

$$\tilde{L}_k(q) := L(q) + \frac{\beta'}{k}D_{\mathrm{KL}}(q\|p_{\mathrm{ref}})$$

decreases. This is an objective function with additional regularization imposed by the hyperparameter $\beta'$. From Lemma 5, we prove that $\tilde{L}_k(\hat{q}^{(k)})$ decreases as in the following lemma.

**Lemma 7.** *Assume that* $\mathrm{TV}(\hat{q}^{(k)}, q^{(k)}) \leq \epsilon_{\mathrm{TV}}/2$ *for all* $k$ *and* $2\beta + S_F \leq 2\beta'$. *Then, Option 2 satisfies*

$$\tilde{L}_{k+1}(\hat{q}^{(k+1)}) - \tilde{L}_k(\hat{q}^{(k)}) \leq (L_{\mathrm{TV}} + B_F)\epsilon_{\mathrm{TV}} - \frac{\beta'(k+1)}{k}D_{\mathrm{KL}}(\hat{q}^{(k)}\|\hat{q}^{(k+1)}), \quad (18)$$

*and Option 1 satisfies*

$$\tilde{L}_{k+1}(\hat{q}^{(k+1)}) - \tilde{L}_k(\hat{q}^{(k)}) \leq (L_{\mathrm{TV}} + B_F)\epsilon_{\mathrm{TV}} - \frac{\beta k(k+1) + 2\beta'(k+1)}{2k}D_{\mathrm{KL}}(\hat{q}^{(k)}\|\hat{q}^{(k+1)}), \quad (19)$$

By this lemma, it can be shown that the sum of the KL divergences $D_{\mathrm{KL}}(\hat{q}^{(k)}\|\hat{q}^{(k+1)})$ converges to 0 at a rate of $\mathcal{O}(1/K)$ through a telescoping sum: we obtain that

$$\frac{1}{K}\sum_{k=1}^{K}(\mathrm{weight})_k D_{\mathrm{KL}}(\hat{q}^{(k)}\|\hat{q}^{(k+1)}) = \mathcal{O}\left(\frac{1}{K}\right).$$

From the above equation, it is immediately shown that

$$\min_{k=1,\ldots,K}(\mathrm{weight})_k D_{\mathrm{KL}}(\hat{q}^{(k)}\|\hat{q}^{(k+1)}) = \mathcal{O}\left(\frac{1}{K}\right),$$

where $(\mathrm{weight})_k \simeq k$ in Option 1, while $(\mathrm{weight})_k \simeq 1$ in Option 2. In Option 2, we will be able to show that $\min_k D_{\mathrm{KL}}(\hat{q}^{(k)}\|\hat{q}^{(k+1)}) \to 0$, which implies that $\hat{q}^{(k)}$ converges to some point (in fact, this is the stationary point).

Now let us prove Lemma 7:

*Proof.* We only prove the inequality for Option 2. By the smoothness of $F$ (the weaker smoothness (ii)' in Eq.(13)), the Lipschitz continuity of $\frac{\delta F}{\delta q}$ and the property of the KL-divergence (Lemma 4), we have

$$L(\hat{q}^{(k+1)}) - L(\hat{q}^{(k)}) \leq \int \frac{\delta F}{\delta q}(\hat{q}^{(k)})\mathrm{d}(\hat{q}^{(k+1)} - \hat{q}^{(k)}) + \frac{S_F}{2}D_{\mathrm{KL}}(\hat{q}^{(k+1)}\|\hat{q}^{(k)})$$

$$+ \beta D_{\mathrm{KL}}(\hat{q}^{(k+1)}\|\hat{q}^{(k)}) + \int(-\beta\bar{g}^{(k-1)})\mathrm{d}(\hat{q}^{(k+1)} - \hat{q}^{(k)})$$

$$\leq \int\left(\frac{\delta F}{\delta q}(q^{(k)}) - \beta\bar{g}^{(k-1)}\right)\mathrm{d}(\hat{q}^{(k+1)} - \hat{q}^{(k)})$$

$$+ \frac{S_F}{2}D_{\mathrm{KL}}(\hat{q}^{(k+1)}\|\hat{q}^{(k)}) + \beta D_{\mathrm{KL}}(\hat{q}^{(k+1)}\|\hat{q}^{(k)}) + L_{\mathrm{TV}}\epsilon_{\mathrm{TV}}. \quad (20)$$

We have used the induced smoothness of $F$ (Eq.(13)) in the first inequality, and used the Lipschitz continuity in the second inequality. By Lemma 5, we can bound $\int \left( \frac{\delta F}{\delta q}(q^{(k)}) - \beta \bar{g}^{(k-1)} \right) \mathrm{d}(\hat{q}^{(k+1)} - \hat{q}^{(k)}) = \int \frac{\delta L}{\delta q}(\hat{q}^{(k)}) \mathrm{d}(\hat{q}^{(k+1)} - \hat{q}^{(k)})$. Thus the RHS of Eq. (20) can be bounded as

$$
\begin{aligned}
(\text{RHS}) \leq & \frac{\beta'}{k} \left( D_{\mathrm{KL}}(\hat{q}^{(k)}\|p_{\mathrm{ref}}) - D_{\mathrm{KL}}(\hat{q}^{(k+1)}\|p_{\mathrm{ref}}) - kD_{\mathrm{KL}}(\hat{q}^{(k+1)}\|\hat{q}^{(k)}) - (k+1)D_{\mathrm{KL}}(\hat{q}^{(k)}\|\hat{q}^{(k+1)}) \right) \\
& + \frac{2\beta + S_F}{2} D_{\mathrm{KL}}(\hat{q}^{(k+1)}\|\hat{q}^{(k)}) + (L_{\mathrm{TV}} + B_F)\epsilon_{\mathrm{TV}} \\
\leq & \frac{\beta'}{k} \left( D_{\mathrm{KL}}(\hat{q}^{(k)}\|p_{\mathrm{ref}}) - D_{\mathrm{KL}}(\hat{q}^{(k+1)}\|p_{\mathrm{ref}}) - (k+1)D_{\mathrm{KL}}(\hat{q}^{(k)}\|\hat{q}^{(k+1)}) \right) \\
& + \frac{(2\beta + S_F - 2\beta')}{2} D_{\mathrm{KL}}(\hat{q}^{(k+1)}\|\hat{q}^{(k)}) + (L_{\mathrm{TV}} + B_F)\epsilon_{\mathrm{TV}},
\end{aligned}
$$

which gives the assertion in Option 2 by noting $\frac{\beta'}{k}D_{\mathrm{KL}}(\hat{q}^{(k+1)}\|p_{\mathrm{ref}}) \geq \frac{\beta'}{k+1}D_{\mathrm{KL}}(\hat{q}^{(k+1)}\|p_{\mathrm{ref}})$. As for Option 1, we repeat the same argument to show the desired result. $\square$

Combining Lemma 5 and Lemma 7, we will prove that

$$
\frac{1}{K}\sum_{k=1}^{K}(\text{weight})_k D_{\mathrm{KL}}(\hat{q}^{(k)}\|\hat{q}^{(k+1)}) = \mathcal{O}\left(\frac{1}{K}\right)
$$

where $(\text{weight})_k \simeq k$ in Option 1 and $(\text{weight})_k \simeq 1$ in Option 2. When we take Option 2, it holds that $\min_k D_{\mathrm{KL}}(\hat{q}^{(k)}\|\hat{q}^{(k+1)}) = \mathcal{O}(1/K)$. It is important that $D_{\mathrm{KL}}(\hat{q}^{(k)}\|\hat{q}^{(k+1)})$ is also the approximate "moment generation function" $\psi_q(g) = \log(\mathbb{E}_q[\exp(-g + \mathbb{E}_q[g])])$ of $\frac{\delta \tilde{L}_k(\hat{q}^{(k)})}{\delta q}$; when $k \to \infty$,

$$
\min_{k=1,..,K} \left( \text{"Variance" of } \frac{\delta \tilde{L}_k}{\delta q} \right)(\hat{q}^{(k)}, x) \simeq \min_{k=1,..,K} D_{\mathrm{KL}}(\hat{q}^{(k)}\|\hat{q}^{(k+1)}) = \mathcal{O}\left(\frac{1}{K}\right).
$$

This can be interpreted as $\frac{\delta \tilde{L}_k}{\delta q}(\hat{q}^{(k)}, x) \to 0$ (up to constant w.r.t. $x$).

This result is consistent with the findings for standard, non-distributional Dual Averaging (Liu et al., 2023a), where $\min_{k=1,\ldots,K} \|(\text{gradient of the objective})_k\|^2 = \mathcal{O}(1/K)$ as the variance corresponds to the second moment..

We rigorously formulate the above discussion as the following proposition.

**Proposition 1.** *Let*

$$
\Psi_K := \frac{1}{K\beta'}(\tilde{L}_1(\hat{q}^{(1)}) - L^*) + \frac{(L_{\mathrm{TV}} + B_F)}{K\beta'}\sum_{k=1}^{K}\epsilon_{\mathrm{TV}}.
$$

*Then, Algorithm 1 satisfies*

$$
\frac{1}{K}\sum_{k=1}^{K}\frac{\beta k + 2\beta'}{2}D_{\mathrm{KL}}(\hat{q}^{(k)}\|\hat{q}^{(k+1)}) \leq \Psi_K
$$

*and Algorithm 2 satisfies*

$$
\frac{1}{K}\sum_{k=1}^{K}D_{\mathrm{KL}}(\hat{q}^{(k)}\|\hat{q}^{(k+1)}) \leq \Psi_K.
$$

*This also yields that the following bound holds for Algorithm 2:*

$$
\begin{aligned}
\min_{1 \leq k \leq K} \psi_{\hat{q}^{(k)}} & \left( \frac{k}{\beta'(k+1)}\frac{\delta \tilde{L}_k}{\delta q}(\hat{q}^{(k)}) \right) \leq \mathbb{E}_{k \sim \mathrm{Univ}([K])}\left[ \psi_{\hat{q}^{(k)}}\left( \frac{k}{\beta'(k+1)}\frac{\delta \tilde{L}_k}{\delta q}(\hat{q}^{(k)}) \right) \right] \\
& \leq \Psi_K,
\end{aligned}
$$

*where the expectation in the middle term is taken over a random index $k$ uniformly chosen from $[K] = \{1, \ldots, K\}$.*

*Proof.* Summing up (19) for $k = 1, \ldots, K$, we have that

$$\tilde{L}_{K+1}(\hat{q}^{(K+1)}) - \tilde{L}_1(\hat{q}^{(1)}) \le (L_{\mathrm{TV}} + B_F) \sum_{k=1}^{K} \epsilon_{\mathrm{TV}} - \beta' \sum_{k=1}^{K} \frac{k+1}{k} D_{\mathrm{KL}}(\hat{q}^{(k)} \| \hat{q}^{(k+1)}).$$

This yields that

$$\frac{\beta'}{K} \sum_{k=1}^{K} \frac{k+1}{k} D_{\mathrm{KL}}(\hat{q}^{(k)} \| \hat{q}^{(k+1)}) + \frac{1}{K}(\tilde{L}_{K+1}(\hat{q}^{(K+1)}) - L^*)$$

$$\le \frac{1}{K}(\tilde{L}_1(\hat{q}^{(1)}) - L^*) + \frac{(L_{\mathrm{TV}} + B_F)}{K} \sum_{k=1}^{K} \epsilon_{\mathrm{TV}}. \tag{21}$$

Please note that

$$\frac{\beta'}{K} \sum_{k=1}^{K} D_{\mathrm{KL}}(\hat{q}^{(k)} \| \hat{q}^{(k+1)}) \le \frac{\beta'}{K} \sum_{k=1}^{K} \frac{k+1}{k} D_{\mathrm{KL}}(\hat{q}^{(k)} \| \hat{q}^{(k+1)}) + \frac{1}{K}(\tilde{L}_{K+1}(\hat{q}^{(K+1)}) - L^*)$$

by the optimality of $L^*$ and that the RHS in (21) is $\mathcal{O}(1/K)$. Here, we see that

$$D_{\mathrm{KL}}(\hat{q}^{(k)} \| \hat{q}^{(k+1)})$$
$$= \mathbb{E}_{\hat{q}^{(k)}}[-\bar{g}^{(k-1)} + \bar{g}^{(k)}] - \log(\mathbb{E}_{p_{\mathrm{ref}}}[\exp(-\bar{g}^{(k-1)})]) + \log(\mathbb{E}_{p_{\mathrm{ref}}}[\exp(-\bar{g}^{(k)})])$$
$$= \mathbb{E}_{\hat{q}^{(k)}}[-\bar{g}^{(k-1)} + \bar{g}^{(k)}] + \log(\mathbb{E}_{\hat{q}^{(k)}}[\exp(-\bar{g}^{(k)} + \bar{g}^{(k-1)})])$$
$$= \log\left\{\mathbb{E}_{\hat{q}^{(k)}}\left[\exp\left(-\bar{g}^{(k)} + \bar{g}^{(k-1)} - \mathbb{E}_{\hat{q}^{(k)}}\left[-\bar{g}^{(k)} + \bar{g}^{(k-1)}\right]\right)\right]\right\},$$

Now, the term $\bar{g}^{(k)} - \bar{g}^{(k-1)}$ can be evaluated as

$$\bar{g}^{(k)} - \bar{g}^{(k-1)} = \frac{k}{\beta'(k+1)} \frac{\delta L}{\delta q}(\hat{q}^{(k)}) - \frac{1}{k+1} \bar{g}^{(k-1)} = \frac{k}{\beta'(k+1)} \frac{\delta \tilde{L}_k}{\delta q}(\hat{q}^{(k)}),$$

which yields the assertion because

$$D_{\mathrm{KL}}(\hat{q}^{(k)} \| \hat{q}^{(k+1)})$$
$$= \log\left\{\mathbb{E}_{\hat{q}^{(k)}}\left[\exp\left(\frac{k}{\beta'(k+1)} \frac{\delta \tilde{L}_k}{\delta q}(\hat{q}^{(k)}) - \mathbb{E}_{\hat{q}^{(k)}}\left[\frac{k}{\beta'(k+1)} \frac{\delta \tilde{L}_k}{\delta q}(\hat{q}^{(k)})\right]\right)\right]\right\},$$

which is the "moment generating function" $\psi_{\hat{q}^{(k)}}$ of $\frac{k}{\beta'(k+1)} \frac{\delta \tilde{L}_k}{\delta q}(\hat{q}^{(k)})$. $\qquad\square$

### A.2.1 PROOF OF LEMMA 6

*Proof.* The proof is almost identical to Lemma 5. In Algorithm 1, $\hat{q}^{(k+1)}$ is defined as the minimizer of the following quantity:

$$r_{k+1}(q) := \sum_{j=1}^{k} j\left(\int \frac{\delta F}{\delta q}(q^{(j)}) \mathrm{d}(q - \hat{q}^{(j)}) + \beta D_{\mathrm{KL}}(q \| p_{\mathrm{ref}})\right) + \beta'(k+1) D_{\mathrm{KL}}(q \| p_{\mathrm{ref}}).$$

Hence, we have

$$r_{k+1}(q) = r_k(q) + \int k \frac{\delta F}{\delta q}(q^{(k)}) \mathrm{d}(q - \hat{q}^{(k)}) + (\beta k + \beta') D_{\mathrm{KL}}(q \| p_{\mathrm{ref}}) \tag{22}$$

by the definition of $r_k$. Moreover, Lemma 4 and the optimality of $\hat{q}^{(k)}$ gives that

$$r_k(q) - r_k(\hat{q}^{(k)}) = \left(\frac{\beta k(k-1)}{2} + \beta' k\right) D_{\mathrm{KL}}(q \| \hat{q}^{(k)}), \quad \text{for all } q \in \mathcal{P}. \tag{23}$$

Substituting $q \leftarrow \hat{q}^{(k+1)}$, we obtain

$$
\begin{aligned}
0 \leq & r_k(\hat{q}^{(k+1)}) - r_k(\hat{q}^{(k)}) - \left( \frac{\beta k(k-1)}{2} + \beta' k \right) D_{\mathrm{KL}}(\hat{q}^{(k+1)} \| \hat{q}^{(k)}) \\
= & r_{k+1}(\hat{q}^{(k+1)}) - \int k \frac{\delta F}{\delta q}(q^{(k)}) \mathrm{d}(\hat{q}^{(k+1)} - \hat{q}^{(k)}) - (\beta k + \beta') D_{\mathrm{KL}}(\hat{q}^{(k+1)} \| p_{\mathrm{ref}}) \quad \text{(used (22))} \\
& - r_k(\hat{q}^{(k)}) - \left( \frac{\beta k(k-1)}{2} + \beta' k \right) D_{\mathrm{KL}}(\hat{q}^{(k+1)} \| \hat{q}^{(k)}).
\end{aligned}
\tag{24}
$$

This is equivalent to

$$
\int k \frac{\delta F}{\delta q}(q^{(k)}) \mathrm{d}(\hat{q}^{(k+1)} - \hat{q}^{(k)})
$$

$$
\leq r_{k+1}(\hat{q}^{(k+1)}) - r_k(\hat{q}^{(k)}) - (\beta k + \beta') D_{\mathrm{KL}}(\hat{q}^{(k+1)} \| p_{\mathrm{ref}}) - \left( \frac{\beta k(k-1)}{2} + \beta' k \right) D_{\mathrm{KL}}(\hat{q}^{(k+1)} \| \hat{q}^{(k)}).
$$

By using the same argument as Eq. (23) for $k \leftarrow k+1$, we have

$$
\begin{aligned}
& r_{k+1}(\hat{q}^{(k+1)}) + \left( \frac{\beta k(k+1)}{2} + \beta'(k+1) \right) D_{\mathrm{KL}}(\hat{q}^{(k)} \| \hat{q}^{(k+1)}) \\
= & r_{k+1}(\hat{q}^{(k)}) \\
= & r_k(\hat{q}^{(k)}) + (\beta k + \beta') D_{\mathrm{KL}}(\hat{q}^{(k)} \| p_{\mathrm{ref}}).
\end{aligned}
$$

Substituting this relation to Eq. (24) and noticing $|\frac{\delta F}{\delta q}| \leq B_F$, we obtain the assertion. $\qquad \square$

## B   DERIVATIONS OF FUNCTIONAL DERIVATIVES

### B.1   DIRECT PREFERENCE OPTIMIZATION

We can apply the proposed algorithm to the (populational) objective of Direct Preference Optimization (DPO) (Rafailov et al., 2023). DPO is an effective approach for learning from human preference for not only language models but also diffusion models.

**Original DPO objective**   Let $x_w, x_l$ be "winning" and "losing" outputs independently sampled from the reference model $p_{\mathrm{ref}}$. The event $\{x_w \succ x_l\}$ is determined by the human preference. The original DPO objective is formulated as

$$
L_{\mathrm{DPO,original}}(q) = -\mathbb{E}_{x_w, x_l} \left[ \log \sigma \left( \gamma \log \frac{q(x_w)}{p_{\mathrm{ref}}(x_w)} - \gamma \log \frac{q(x_l)}{p_{\mathrm{ref}}(x_l)} \right) \right],
$$

where $\gamma > 0$ is a hyperparameter. The expectation is taken by $x_w, x_l$, that are "winning" and "losing" samples from $p_{\mathrm{ref}}$. Wallace et al. (2024) derived a new objective that is the upper bound of $L_{\mathrm{DPO,original}}(q)$, but it is a specialized derivation of the optimization method for DPO.

**Reformulating of the DPO Objective**   The goal is to directly minimize $L_{\mathrm{DPO,original}}(q)$, however, in the above expression, we cannot apply DPO to diffusion models directly because the expectaion with tuple $(x_w, x_l)$ is not formulated well. We start with another expression of the objective of DPO:

$$
L_{\mathrm{DPO}}(q) := -\mathbb{E}_{x_w \sim p_{\mathrm{ref}}} \mathbb{E}_{x_l \sim p_{\mathrm{ref}}} \left[ \log \sigma \left( \gamma \log \frac{q(x_w)}{p_{\mathrm{ref}}(x_w)} - \gamma \log \frac{q(x_l)}{p_{\mathrm{ref}}(x_l)} \right) \mathbb{1}_{x_w \succ x_l}(x_w, x_l) \right],
$$

$\mathbb{1}_{x \succ y}(x, y)$ is 1 if $x \succ y$, 0 otherwise. This $L_{\mathrm{DPO}}$ is in the regime of our algorithm and the functional derivative can be derived:

**Proposition 2.**  *The functional derivative of $L_{\mathrm{DPO}}(q)$ is calculated as*

$$
\begin{aligned}
& \frac{\delta L_{\mathrm{DPO}}}{\delta q}(q, x) \\
= & -\gamma \mathbb{E}_{x_l \sim p_{\mathrm{ref}}} \left[ (1 - \sigma(-\gamma f(x) + \gamma f(x_l))) \frac{\int e^{-f} dp_{\mathrm{ref}}}{e^{-f(x)}} \mathbb{1}_{x \succ x_l}(x, x_l) \right] \\
& + \gamma \mathbb{E}_{x_w \sim p_{\mathrm{ref}}} \left[ (1 - \sigma(-\gamma f(x_w) + \gamma f(x))) \frac{\int e^{-f} dp_{\mathrm{ref}}}{e^{-f(x)}} \mathbb{1}_{x_w \succ x}(x_w, x) \right],
\end{aligned}
$$

*where $q = e^{-f} p_{\mathrm{ref}} / \int e^{-f} \mathrm{d} p_{\mathrm{ref}}$. This functional derivative is tractable in our settings.*

*Proof.* In this proof, we use the following notations:

- $p_{\text{ref}}$ : the output distribution of a pre-trained model,

- $q := e^{-f} p_{\text{ref}} / \int e^{-f} dp_{\text{ref}}$: the output distribution of an aligned model,

- $\text{LSL}(q_1, q_2) := \log \sigma(\gamma \log q_1/p_{\text{ref}} - \gamma \log q_2/p_{\text{ref}})$,

- $\partial_1 \text{LSL}(q_1, q_2) = \gamma(1 - \sigma(\gamma \log q_1/p_{\text{ref}} - \gamma \log q_2/p_{\text{ref}}))\frac{1}{q_1}$,

- $\partial_2 \text{LSL}(q_1, q_2) = -(1 - \sigma(\gamma \log q_1/p_{\text{ref}} - \gamma \log q_2/p_{\text{ref}}))\frac{1}{q_2}$,

- $\psi(r_1, r_2) := \gamma(\log r_1 - \log r_2)$,

- $\text{Inv}(f, x) = \frac{\int e^{-f} dp_{\text{ref}}}{e^{-f(x)}}$.

The objective of DPO is written as

$$L_{\text{DPO}}(q) := -\mathrm{E}_{x_w \sim p_{\text{ref}}} \mathrm{E}_{x_l \sim p_{\text{ref}}} \left[\text{LSL}(q(x_w), qp(x_l)) \mathbb{1}_{x_w \succ x_l}(x_w, x_l)\right].$$

We obtain the first variation of the objective as follows:

$$
\begin{aligned}
&L_{\text{DPO}}(q + \epsilon(\tilde{q} - q)) \\
=&L_{\text{DPO}}(q) - \epsilon \mathrm{E}_{x_w \sim p_{\text{ref}}} \mathrm{E}_{x_l \sim p_{\text{ref}}} \left[(\partial_1 \text{LSL}(q(x_w), q(x_l))(\tilde{q} - q)(x_w)\right. \\
&\qquad\qquad \left. + \partial_2 \text{LSL}(q(x_w), q(x_l))(\tilde{q} - q)(x_l)) \mathbb{1}_{x_w \succ x_l}(x_w, x_l)\right] + \mathcal{O}(\epsilon^2) \\
=&L_{\text{DPO}}(p) \\
&- \epsilon \left[ \mathrm{E}_{x_l \sim p_{\text{ref}}} \left[ \int \gamma(1 - \sigma(\psi(\frac{q(x_w)}{p_{\text{ref}}(x_w)}, \frac{q(x_l)}{p_{\text{ref}}(x_l)})))(\tilde{q} - q)(x_w)\text{Inv}(f, x_w) dx_w \mathbb{1}_{x_w \succ x_l}\right]\right. \\
&\left. + \mathrm{E}_{x_w \sim p_{\text{ref}}} \left[ \int \gamma(1 - \sigma(\psi(\frac{q(x_w)}{p_{\text{ref}}(x_w)}, \frac{q(x_l)}{p_{\text{ref}}(x_l)})))(\tilde{q} - q)(x_l)\text{Inv}(f, x_l) dx_l \mathbb{1}_{x_w \succ x_l}\right]\right] \\
&+ \mathcal{O}(\epsilon^2).
\end{aligned}
$$

Then, the first derivative of $F$ is

$$
\begin{aligned}
\frac{\delta L_{\text{DPO}}}{\delta p}(p, x) = &- \mathrm{E}_{x_l \sim p_{\text{ref}}} \left[ \gamma \left(1 - \sigma \left(\psi(\frac{q(x)}{p_{\text{ref}}(x)}, \frac{q(x_l)}{p_{\text{ref}}(x_l)})\right)\right) \mathbb{1}_{x \succ x_l}(x, x_l)\right] \text{Inv}(f, x) \\
&+ \mathrm{E}_{x_w \sim p_{\text{ref}}} \left[ \gamma \left(1 - \sigma \left(\psi(\frac{q(x_w)}{p_{\text{ref}}(x_w)}, \frac{q(x)}{p_{\text{ref}}(x)})\right)\right) \mathbb{1}_{x_w \succ x}(x_w, x)\right] \text{Inv}(f, x).
\end{aligned}
$$

$\square$

## B.2 KAHNEMAN-TVERSKY OPTIMIZATION

Assume that the whole data space $\mathbb{R}^d$ is split into a desirable domain $\mathcal{D}_{\text{D}}$ and an undesirable domain $\mathcal{D}_{\text{U}}$. The objective of original KTO (Ethayarajh et al., 2024) is formulated as

$$
\begin{aligned}
L_{\text{KTO}}(q) = &\mathbb{E}_{x \sim p_{\text{ref}}} \left[ \gamma_D \left(1 - \sigma \left(\kappa \log \frac{q}{p_{\text{ref}}} - D_{\text{KL}}(q\|p_{\text{ref}})\right)\right) \mathbb{1}_{\{x \in \mathcal{D}_{\text{D}}\}}\right. \\
&\left. + \gamma_U \left(1 - \sigma \left(D_{\text{KL}}(q\|p_{\text{ref}}) - \kappa \log \frac{q}{p_{\text{ref}}}\right)\right) \mathbb{1}_{\{x \in \mathcal{D}_{\text{U}}\}}\right],
\end{aligned}
$$

where $\gamma_D, \gamma_U, \kappa$ are hyper parameters, and $\sigma$ is a sigmoid function.

**Proposition 3.** *The functional derivative of $L_{\text{KTO}}$ is calculated as*

$$\frac{\delta L_{\text{KTO}}}{\delta q}(q, x)$$

$$= -\kappa\gamma_D\sigma_{\text{deriv}}(\phi(x))\frac{\int e^{-f}dp_{\text{ref}}}{e^{-f(x)}}\mathbb{1}_{\{x\in\mathcal{D}_{\text{D}}\}}$$

$$+ (-f(x) - \log\int e^{-f(x)}dp_{\text{ref}})\mathbb{E}_{y\sim p_{\text{ref}}}\left[\gamma_D\sigma_{\text{deriv}}(\phi(y))\mathbb{1}_{\{y\in\mathcal{D}_{\text{D}}\}}\right]$$

$$+ \kappa\gamma_U\sigma_{\text{deriv}}(-\phi(x))\frac{\int e^{-f}dp_{\text{ref}}}{e^{-f(x)}}\mathbb{1}_{\{x\in\mathcal{D}_{\text{U}}\}}$$

$$- (-f(x) - \log\int e^{-f(x)}dp_{\text{ref}})\mathbb{E}_{y\sim p_{\text{ref}}}\left[\gamma_U\sigma_{\text{deriv}}(-\phi(y))\mathbb{1}_{\{y\in\mathcal{D}_{\text{U}}\}}\right]$$

*where $\sigma_{\text{deriv}}(\cdot) := \sigma(\cdot)(1 - \sigma(\cdot))$, $\phi(x) := \kappa\log\frac{q(x)}{p_{\text{ref}}(x)} - D_{\text{KL}}(q\|p_{\text{ref}})$, $q = \frac{e^{-f}p_{\text{ref}}}{\int e^{-f}dp_{\text{ref}}}$.*

The functional derivative of KTO can be calculated if you have $f(x)$ and the samples from $p_{\text{ref}}$. Note that $\log q(x)/p_{\text{ref}}(x) = -f(x) - \log\int e^{-f(x)}dp_{\text{ref}}$.

*Proof.* The first variation of $L_{\text{KTO}}$ is

$$L_{\text{KTO}}(q + \epsilon(\tilde{q} - q)) - L_{\text{KTO}}(q)$$

$$\simeq\epsilon\mathbb{E}_{x\sim p_{\text{ref}}}\left[-\kappa\gamma_D\sigma_{\text{deriv}}(\phi(x))\left(\frac{\tilde{q}(x) - q(x)}{q(x)} + \int\log\frac{q(y)}{p_{\text{ref}}(y)}d(\tilde{q} - q)(y)\right)\mathbb{1}_{\{x\in\mathcal{D}_{\text{D}}\}}\right]$$

$$+ \epsilon\mathbb{E}_{x\sim p_{\text{ref}}}\left[\kappa\gamma_U\sigma_{\text{deriv}}(-\phi(x))\left(\frac{\tilde{q}(x) - q(x)}{q(x)} - \int\log\frac{q(y)}{p_{\text{ref}}(y)}d(\tilde{q} - q)(y)\right)\mathbb{1}_{\{x\in\mathcal{D}_{\text{U}}\}}\right]$$

$$= \epsilon\int\left\{-\kappa\gamma_D\sigma_{\text{deriv}}(\phi(x))\text{Inv}(f, x)\mathbb{1}_{\{x\in\mathcal{D}_{\text{D}}\}}\right.$$

$$+ \log\frac{q(x)}{p_{\text{ref}}(x)}\mathbb{E}\left[\kappa\gamma_D\sigma_{\text{deriv}}(\phi(y))\mathbb{1}_{\{y\in\mathcal{D}_{\text{D}}\}}\right]\left.\right\}d(\tilde{q} - q)(y)$$

$$+ \epsilon\int\left\{\kappa\gamma_U\sigma_{\text{deriv}}(-\phi(x))\text{Inv}(f, x)\mathbb{1}_{\{x\in\mathcal{D}_{\text{D}}\}}\right.$$

$$- \log\frac{q(x)}{p_{\text{ref}}(x)}\mathbb{E}\left[\kappa\gamma_D\sigma_{\text{deriv}}(-\phi(y))\mathbb{1}_{\{y\in\mathcal{D}_{\text{U}}\}}\right]\left.\right\}d(\tilde{q} - q)(y),$$

where $\text{Inv}(f, x) = \frac{\int e^{-f}dp_{\text{ref}}}{e^{-f(x)}}$. Now the desired result immediately follows. $\square$

## C  ERROR ANALYSIS OF DIFFUSION MODEL

### C.1  OVERVIEW

Here, we analyze the sampling error caused by the diffusion model. Let us organize the settings and notations used in this section.

**Target distribution.** The target distribution is $q_* = q_0$, which is decomposed as $q_*(x) = \rho_*(x)p_*(x)$ with $p_* = p_0$ and $\rho_*$. Here $p_*$ and $\rho_*$ represent the distribution of the original model and the density ratio obtained by fine-tuning, respectively.

**Sampling with score-based diffusion model.** In the score-based diffusion model, we start with the forward process, which is written as a stochastic differential equation (SDE). Choosing the Ornstein–Uhlenbeck (OU) process, $\{\bar{X}_t\}_{t\geq 0}$ follows the following SDE:

$$\bar{X}_0 \sim p_*, \ \mathrm{d}\bar{X}_t = -\bar{X}_t\mathrm{d}t + \sqrt{2}\mathrm{d}B_t, \tag{25}$$

where $\{B_t\}_{t\geq 0}$ is the standard Brownian motion. At each time $t$, the law of $X_t$ is written as

$$p_t(x) = \int p_*(y)\exp\left(-\frac{1}{2\sigma_t^2}\|m_t y - x\|^2\right)\mathrm{d}x,$$

where $m_t = e^{-t}$ and $\sigma_t^2 = 1 - e^{-2t}$. In the same way, define $\{\bar{Y}_t\}_{t \geq 0}$ by replacing $p_*$ by $q_*$ in (25) and let $q_t$ be the law of $\bar{Y}_t$.

Then, for some $T \geq 0$, we can define the reverse process for $\{\bar{X}_t^\leftarrow\}_{0 \leq t \leq T}$. (Let $B_t$ below be distinct from the one in (25).)

$$\bar{X}_0^\leftarrow \sim p_T, \ \mathrm{d}\bar{X}_t^\leftarrow = \{\bar{X}_t^\leftarrow + 2\nabla \log p_{T-t}(\bar{X}_t^\leftarrow)\}\mathrm{d}t + \sqrt{2}\mathrm{d}B_t. \tag{26}$$

Then, the law of $\bar{X}_t^\leftarrow$ equals $p_{T-t}$, which is why we call (26) as the reverse process. In the same way, we define $\{\bar{Y}_t^\leftarrow\}_{0 \leq t \leq T}$ as the reverse process of $\{\bar{Y}_t\}_{t \geq 0}$.

**Doob's h-transform.** By applying Doob's h-transform to $\nabla \log q_{T-t}$, it can be decomposed into the original score $\nabla \log p_{T-t}$ and a correction term.

$$\nabla \log q_{T-t}(\bar{Y}_t^\leftarrow) = \nabla \log p_{T-t}(\bar{Y}_t^\leftarrow) + \nabla_x \log(\mathbb{E}[\rho_*(\bar{X}_0)|\bar{X}_{T-t} = x])|_{x=\bar{Y}_t^\leftarrow}. \tag{27}$$

See Lemma 12 for derivation.

**Approximation of the score and correction term.** Because we do not have access to the exact value of $p_t$ and $\rho_*$ and therefore cannot implement (27) exactly, we consider approximating them by, e.g., neural networks. We approximate $\nabla \log p_{T-t}(x)$ by $s(x,t): \mathbb{R}^{d+1} \to \mathbb{R}^d$. Also, we approximate $u_*(x,t) = \nabla_x \log(\mathbb{E}[\rho_*(\bar{X}_0)|\bar{X}_{T-t} = x])$ by $u(x,t): \mathbb{R}^{d+1} \to \mathbb{R}^d$.

**Discretization.** Also, we need to discretize the stochastic differential equation. We finally obtain the approximation of (26), denoted by $\{Y_t^\leftarrow\}_{0 \leq t \leq T}$, as

$$Y_0^\leftarrow \sim \mathcal{N}(0, I_d), \ \mathrm{d}Y_t^\leftarrow = \{Y_t^\leftarrow + 2s(Y_{kh}^\leftarrow, kh) + 2u(Y_{kh}^\leftarrow, kh)\}\mathrm{d}t + \sqrt{2}\mathrm{d}B_t, \ t \in [kh, (k+1)h].$$

**Obtaining the correction term (approximately).** Given the score network $s(x,t)$ approximating $\nabla \log p_t(x)$ and the function $\bar{h}$ that approximates $h_*$, we can approximate the correction term $u(x,t)$. Remind that, for fixed $x \in \mathbb{R}^d$, $t = kh$, $s = k(h+1) \in \mathbb{R}$, the correction term is calculated as

$$\nabla_x \log(\mathbb{E}[\rho_*(\bar{X}_0)|\bar{X}_{T-t} = x]) = \frac{\int \mathbb{E}[\rho_*(\bar{X}_0)|\bar{X}_{T-s} = x']\frac{\partial}{\partial x}\mathbb{P}[\bar{X}_{T-s} = x'|\bar{X}_{T-t} = x]\mathrm{d}x'}{\mathbb{E}[\rho_*(\bar{X}_0)|\bar{X}_{T-t} = x]}. \tag{28}$$

One way to approximate (28) starts from approximating $\mathbb{E}[\rho_*(\bar{X}_0)|\bar{X}_{T-t} = x]$. If we run the reverse diffusion process

$$\tilde{X}_t^\leftarrow = x, \ \mathrm{d}\tilde{X}_\tau^\leftarrow = \{\tilde{X}_\tau^\leftarrow + 2\nabla \log p_{T-\tau}(\tilde{X}_\tau^\leftarrow)\}\mathrm{d}\tau + \sqrt{2}\mathrm{d}B_\tau,$$

we obtain that the law of $\tilde{X}_T^\leftarrow$ is equal to that of $\bar{X}_0|\bar{X}_{T-t} = x$. Therefore, by running the approximated reverse process (with a slight abuse of notation)

$$\tilde{X}_t = x, \ \mathrm{d}\tilde{X}_\tau = \{\tilde{X}_\tau + 2s(\tilde{X}_{lh}, lh)\}\mathrm{d}t + \sqrt{2}\mathrm{d}B_\tau, \ \tau \in [lh, (l+1)h],$$

multiple times, the sample of $\tilde{X}_T$, denoted by $\{\tilde{x}_{T,i}\}_{i=1}^n$, can approximate $\mathbb{E}[\rho_*(\bar{X}_0)|\bar{X}_{T-t} = x]$ as

$$\mathbb{E}[\rho_*(\bar{X}_0)|\bar{X}_{T-t} = x] \approx \frac{1}{n}\sum_{i=1}^n \rho'(\tilde{x}_{T,i}), \tag{29}$$

where $\rho'$ is the approximation of $\rho_*$.

On the other hand, we approximate $\frac{\partial}{\partial x}\mathbb{P}[\bar{X}_{T-s} = x'|\bar{X}_{T-t} = x]$ by approximating $\mathbb{P}[\bar{X}_{T-s} = x'|\bar{X}_{T-t} = x]$ with a Gaussian distribution. Specifically, because $\bar{X}_{T-s} = x'|\bar{X}_{T-t} = x$ is obtained by the following reverse diffusion process

$$\tilde{X}_t^\leftarrow = x, \ \mathrm{d}\tilde{X}_\tau^\leftarrow = \{\tilde{X}_\tau^\leftarrow + 2\nabla \log p_{T-\tau}(\tilde{X}_\tau^\leftarrow)\}\mathrm{d}\tau + \sqrt{2}\mathrm{d}B_\tau,$$

we approximate $\nabla \log p_{T-\tau}(\tilde{X}_\tau^\leftarrow)$ by $s(x_{kh}, kh)$ to obtain

$$\dot{X}_t^\leftarrow = x_{kh}, \ \mathrm{d}\dot{X}_\tau^\leftarrow = \{\dot{X}_\tau^\leftarrow + 2s(x_{kh}, kh)\}\mathrm{d}\tau + \sqrt{2}\mathrm{d}B_\tau.$$

The distribution of $\tilde{X}_s^\leftarrow$ is denoted by

$$\mathcal{N}(e^h x_{kh} + 2(e^h - 1)s(x_{kh}, kh), e^{2h} - 1). \tag{30}$$

Using this, our approximation is

$$\frac{\partial}{\partial x}\mathbb{P}[\bar{X}_{T-s} = x'|\bar{X}_{T-t} = x]$$

$$\approx \frac{\partial}{\partial x}\frac{1}{(2\pi(e^{2h}-1))^{\frac{d}{2}}}\exp\left(-\frac{(x' - (e^h x + 2(e^h-1)s(x_{kh}, kh)))^2}{2(e^{2h}-1)}\right)$$

$$= \frac{e^h(x' - (e^h x + 2(e^h-1)s(x_{kh}, kh)))}{(e^{2h}-1)(2\pi(e^{2h}-1))^{\frac{d}{2}}}\exp\left(-\frac{(x' - (e^h x + 2(e^h-1)s(x_{kh}, kh)))^2}{2(e^{2h}-1)}\right).$$

This implies that, if we sample $\{x'_j\}_{j=1}^m$ from (30),

$$\int \mathbb{E}[\rho_*(\bar{X}_0)|\bar{X}_{T-s} = x']\frac{\partial}{\partial x}\mathbb{P}[\bar{X}_{T-s} = x'|\bar{X}_{T-t} = x]\mathrm{d}x'$$

$$\approx \frac{e^h}{(e^{2h}-1)}\frac{1}{m}\sum_{j=1}^m \mathbb{E}[\rho_*(\bar{X}_0)|\bar{X}_{T-s} = x'_j](x'_j - (e^h x + 2(e^h-1)s(x_t, t))),$$

and approximate each $\mathbb{E}[\rho_*(\bar{X}_0)|\bar{X}_{T-s} = x'_j]$ in the same way as (29), we can approximate the correction term (28).

Now, our goal is to bound the error of the whole pipeline under the following assumptions of $p_*$ and $\rho_*$ and the score approximation error.

**Assumption 5** (Assumption 3, restated). *1. $\nabla \log p_t$ is $L_p$-smooth at every time $t$ and it has finite second moment $\mathbb{E}[\|\bar{X}_t\|_2^2] \leq \mathsf{m} < \infty$ for all $t \in \mathbb{R}_+$ and $x \in \mathbb{R}^d$.*

*2. $\nabla \log \rho_*$ is $L_\rho$-smooth and bounded as $C_\rho^{-1} \leq \rho_* \leq C_\rho$ for a constant $C_\rho$.*

*3. The score estimation error is bounded by $\mathbb{E}_{\bar{X}\leftarrow}[\|s(\bar{X}_t^\leftarrow, t) - \nabla \log p_{T-t}(\bar{X}_{T-t}^\leftarrow)\|^2] \leq \varepsilon$ at each time $t$.*

*4. $\mathbb{E}_{p_t}[\|u_*(x, lh) - u(x, lh)\|^2] \leq \varepsilon_{\rho,l}^2$ for any $1 \leq l \leq T/h$.*

**Theorem 5** (Theorem 3, restated). *Suppose that Assumption 5 is satisfied. Then, we have the following bound on the distribution $\hat{q}$ of $Y_T^\leftarrow$:*

$$\mathrm{TV}(q_*, \hat{q})^2 \lesssim T\varepsilon^2 + \sum_{l=1}^{T/h} h\varepsilon_{\rho,l}^2 + T(L_p C_\rho^2 + L_\rho)^2(dh + \mathsf{m}^2 h^2) + \exp(-2T)D_{\mathrm{KL}}(\hat{q}_{\mathrm{opt}}\|N(0,I)).$$

*Proof.* Suppose that $\|s(\cdot, t) + u(\cdot, t) - \nabla \log q_{T-t}\|_{L^\infty} \leq \varepsilon'$ and $\nabla \log q_{T-t}$ is $L_q$-smooth at every time $t$. According to Chen et al. (2023c) and Pinsker's inequality, the distribution $\hat{q}$ of $Y_T$ satisfies

$$\mathrm{KL}(\hat{q}\|q_*)^2 \lesssim T\varepsilon^2 + \sum_{l=1}^{T/h} h\varepsilon_{\rho,l}^2 + TL_p^2(dh + \mathsf{m}^2 h^2) + \exp(-2T)D_{\mathrm{KL}}(\hat{q}_{\mathrm{opt}}\|N(0,I)).$$

According to Lemma 8, $L_q$ is bounded by

$$L_q \lesssim L_p C_\rho^2 + L_\rho,$$

which yields the assertion. $\qquad\square$

In the bound, we assumed that the term $\varepsilon_{\rho,l}^2$ can be bounded, however this approximation error can be derived as in the following theorem with additional technical conditions.

**Assumption 6** (Assumption 4 restated). *(i) $\nabla_x s(\cdot, \cdot)$ is $H_s$-Lipschitz continuous in a sense that $\|\nabla_x s(x,t) - \nabla_y s(y,t)\|_{\mathrm{op}} \leq H_s\|x-y\|$ for any $x, y \in \mathbb{R}^d$ and $0 \leq t \leq T$ and $\mathbb{E}[\|s(\tilde{X}_{kh}^\leftarrow, kh)\|^2] \leq Q^2$ for any $k$, (ii) There exists $R > 0$ such that $\sup_{t,x}\{\|\nabla_x^2 \log p_t(x)\|_{\mathrm{op}}, \|\nabla_x^2 \log s(x,t)\|_{\mathrm{op}}\} \leq R$.*

**Theorem 6** (Theorem 4 restated). *Suppose that Assumptions 3 and 4 hold. Assume that $\|\rho_* - \rho\|_\infty \leq \varepsilon'$, $\|\rho\|_\infty \leq C_\rho$, and $\sup_x\|\nabla\rho_*(x)\| \leq R_\rho$, $\|\nabla\rho_*(x) - \nabla\rho_*(y)\| \leq L_\rho\|x-y\| \; (\forall x, y)$. Then, for any choice of $0 \leq h \leq \delta \leq 1/(1+2R)$, we have that*

$$\varepsilon_{\rho,l}^2 \lesssim C_\rho^3 \left\{\Xi_{\delta,\varepsilon} + R_\varphi^2\left(\varepsilon^2 + L_p^2 d(\delta + \mathsf{m}\delta^2)\right) + [L_\varphi^2(\mathsf{m} + Q^2 + dh) + R_\varphi^2(1+2R)^2]h^2\right\}$$
$$+ \min\{T - lh, 1/(2+2R)\}^{-1}\varepsilon'^2,$$

*where $\Xi_{\delta,\varepsilon} := C_\rho^2(1+2R)^2\delta + C_\rho^2\frac{1+\delta R_\varphi^2}{\delta}[\varepsilon^2 + L_p^2 d(h + \mathsf{m}h^2)]$, and $R_\varphi$ and $L_\varphi$ are constants introduced in Lemma 10.*

## C.2 Bounding the smoothness

The proof of Theorem 3 (i.e., Theorem 5) utilizes the smoothness of the density $q_t$ corresponding to the aligned model. The following lemma gives its bound.

**Lemma 8.** *Suppose that $\nabla \log p_t$ is $L_p$-Lipschitz for all $t$ and $\nabla \log \rho_*$ is $L_\rho$-Lipschitz, and $C_\rho^{-1} \leq \rho_* \leq C_\rho$. Then, $\nabla \log q_t$ is $L_q$-Lipschitz for all $t$, where $L_q$ is bounded by*

$$L_q \leq \min \left\{ \frac{4(L_p + L_\rho)^2}{2(L_p + L_\rho) - 1}, (4 + C_\rho^2)L_p + 4L_\rho \right\} \lesssim L_p C_\rho^2 + L_\rho.$$

*Proof.* We divide the proof into two parts, with $\sigma_t^2 = \frac{1}{2(L_p + L_\rho)}$ as the boundary.

First consider the case when $\sigma_t^2 \leq \frac{1}{2(L_p + L_\rho)}$. Remind that

$$q_t(x) = \int q_*(y) \frac{1}{(2\pi\sigma_t^2)^{\frac{d}{2}}} \exp\left( -\frac{1}{2\sigma_t^2} \|m_t y - x\|^2 \right) dy,$$

with $\sigma_t^2 = 1 - e^{-2t}$ and $m_t = e^{-t}$. From this, $\nabla_x q_t(x)$ and $\nabla_x^2 q_t(x)$ are computed as

$$\nabla_x q_t(x) = \nabla_x \int q_*(y) \frac{1}{(2\pi\sigma_t^2)^{\frac{d}{2}}} \exp\left( -\frac{1}{2\sigma_t^2} \|m_t y - x\|^2 \right) dy$$

$$= m_t^{-1} \int (\nabla_y q_*(y)) \frac{1}{(2\pi\sigma_t^2)^{\frac{d}{2}}} \exp\left( -\frac{1}{2\sigma_t^2} \|m_t y - x\|^2 \right) dy$$

and

$$\nabla_x^2 q_t(x) = m_t^{-2} \int (\nabla_y^2 q_*(y)) \frac{1}{(2\pi\sigma_t^2)^{\frac{d}{2}}} \exp\left( -\frac{1}{2\sigma_t^2} \|m_t y - x\|^2 \right) dy.$$

Thus, we can compute $\nabla_x^2 \log q_t(x)$ as

$$\nabla_x^2 \log q_t(x)$$

$$= \frac{\nabla_x^2 q_t(x)}{q_t(x)} - \frac{\nabla_x q_t(x)(\nabla_x q_t(x))^\top}{(q_t(x))^2}$$

$$= \frac{m_t^{-2} \int \left( \frac{\nabla_y^2 q_*(y)}{q_*(y)} - \frac{\nabla_y q_*(y) \nabla_y q_*(y)^\top}{q_*(y)^2} \right) q_*(y) \exp\left( -\frac{1}{2\sigma_t^2} \|m_t y - x\|^2 \right) dy}{\int q_*(y) \exp\left( -\frac{1}{2\sigma_t^2} \|m_t y - x\|^2 \right) dy} \tag{31}$$

$$+ \frac{m_t^{-2} \int \frac{\nabla_y q_*(y) \nabla_y q_*(y)^\top}{q_*(y)^2} q_*(y) \exp\left( -\frac{1}{2\sigma_t^2} \|m_t y - x\|^2 \right) dy}{\int q_*(y) \exp\left( -\frac{1}{2\sigma_t^2} \|m_t y - x\|^2 \right) dy} \tag{32}$$

$$- \frac{m_t^{-2} \int \frac{\nabla_y q_*(y)}{q_*(y)} q_*(y) \exp\left( -\frac{1}{2\sigma_t^2} \|m_t y - x\|^2 \right) dy \int \frac{\nabla_y q_*(y)}{q_*(y)} q_*(y) \exp\left( -\frac{1}{2\sigma_t^2} \|m_t y - x\|^2 \right) dy}{(\int q_*(y) \exp\left( -\frac{1}{2\sigma_t^2} \|m_t y - x\|^2 \right) dy)^2}. \tag{33}$$

Eq. (31) is an expectation of $\nabla_y^2 \log q_*(y) = \frac{\nabla_y^2 q_*(y)}{q_*(y)} - \frac{\nabla_y q_*(y) \nabla_y q_*(y)^\top}{q_*(y)^2}$ with respect to a distribution

$$A(y|x) \propto q_*(y) \exp\left( -\frac{1}{2\sigma_t^2} \|m_t y - x\|^2 \right).$$

Therefore, (31) is bounded by $m_t^{-2}(L_p + L_\rho)$ from the assumption. On the other hand, the other two terms are regarded as the covariance of $\nabla \log q_*(y)$ with respect to that distribution. Because $\sigma_t^2 \leq \frac{1}{2(L_p + L_\rho)}$, $A(y|x)$ is $(L_p + L_\rho)$-strongly concave, and therefore satisfies the Poincaré inequality with a constant $\frac{1}{L_p + L_\rho}$. Therefore, for any $a \in \mathbb{R}^d$, we have

$$a^\top((32) + (33))a$$

$$= m_t^{-2} a^\top (\mathbb{E}_{A(y|x)}[(\nabla \log q_*(y))(\nabla \log q_*(y))^\top] - \mathbb{E}_{A(y|x)}[\nabla \log q_*(y)]\mathbb{E}_{A(y|x)}[\nabla \log q_*(y)]^\top)a$$

$$\leq \frac{m_t^{-2}}{L_p + L_\rho} \cdot \mathbb{E}[\|a\nabla^2 \log q_*(y)\|^2] \leq \frac{m_t^{-2}}{L_p + L_\rho}(L_p + L_\rho)^2 = m_t^{-2} L_p + L_\rho.$$

This implies that (32)+(33) is $m_t^{-2}(L_p + L_\rho)$-smooth and $\nabla_x^2 \log q_t(x)$ is $2m_t^{-2}(L_p + L_\rho)$-smooth. Because $m_t = \sqrt{1 - \sigma_t^2}$, we have $m_t \geq \sqrt{1 - \frac{1}{2(L_p + L_\rho)}}$. By applying this, we have $2m_t^{-2}(L_p + L_\rho) \leq \frac{4(L_p + L_\rho)^2}{2(L_p + L_\rho) - 1}$.

Next, let us consider the case when $\sigma_t^2 \geq \frac{2}{(L_p + L_\rho)}$. Note that $\nabla_x q_t(x)$ and $\nabla_x^2 q_t(x)$ are also written as

$$\nabla_x q_t(x) = -\int q_*(y) \frac{1}{(2\pi\sigma_t^2)^{\frac{d}{2}}} \exp\left(-\frac{1}{2\sigma_t^2}\|m_t y - x\|^2\right) \frac{x - m_t y}{\sigma_t^2} \mathrm{d}y$$

and

$$\nabla_x^2 q_t(x)$$
$$= -\int q_*(y) \frac{1}{(2\pi\sigma_t^2)^{\frac{d}{2}}} \exp\left(-\frac{1}{2\sigma_t^2}\|m_t y - x\|^2\right) \frac{I}{\sigma_t^2} \mathrm{d}y$$
$$+ \int q_*(y) \frac{1}{(2\pi\sigma_t^2)^{\frac{d}{2}}} \exp\left(-\frac{1}{2\sigma_t^2}\|m_t y - x\|^2\right) \frac{(x - m_t y)(x - m_t y)^\top}{\sigma_t^4} \mathrm{d}y.$$

Thus, we can compute $\nabla_x^2 \log q_t(x)$ as

$$\nabla_x^2 \log q_t(x)$$
$$= \frac{\nabla_x^2 q_t(x)}{q_t(x)} - \frac{\nabla_x q_t(x)(\nabla_x q_t(x))^\top}{(q_t(x))^2}$$
$$= -\frac{\int q_*(y) \exp\left(-\frac{1}{2\sigma_t^2}\|m_t y - x\|^2\right) \frac{I}{\sigma_t^2} \mathrm{d}y}{\int q_*(y) \exp\left(-\frac{1}{2\sigma_t^2}\|m_t y - x\|^2\right) \mathrm{d}y} \tag{34}$$
$$+ \frac{\int q_*(y) \exp\left(-\frac{1}{2\sigma_t^2}\|m_t y - x\|^2\right) \frac{(x - m_t y)(x - m_t y)^\top}{\sigma_t^4} \mathrm{d}y}{\int q_*(y) \exp\left(-\frac{1}{2\sigma_t^2}\|m_t y - x\|^2\right) \mathrm{d}y} \tag{35}$$
$$- \frac{\left(\int q_*(y) \exp\left(-\frac{1}{2\sigma_t^2}\|m_t y - x\|^2\right) \frac{(x - m_t y)}{\sigma_t^2} \mathrm{d}y\right)\left(\int q_*(y) \exp\left(-\frac{1}{2\sigma_t^2}\|m_t y - x\|^2\right) \frac{(x - m_t y)}{\sigma_t^2} \mathrm{d}y\right)^\top}{\left(\int q_*(y) \exp\left(-\frac{1}{2\sigma_t^2}\|m_t y - x\|^2\right) \mathrm{d}y\right)^2}. \tag{36}$$

Eq. (34) is bounded by $\sigma_t^{-2} \leq 2(L_p + L_\rho)$. On the other hand, (35)+(36) are transformed into

$$(35)+(36)$$
$$= \frac{m_t^2}{\sigma_t^4} \frac{\int q_*(y) \exp\left(-\frac{1}{2\sigma_t^2}\|m_t y - x\|^2\right) yy^\top \mathrm{d}y}{\int q_*(y) \exp\left(-\frac{1}{2\sigma_t^2}\|m_t y - x\|^2\right) \mathrm{d}y}$$
$$- \frac{m_t^2}{\sigma_t^4} \frac{\left(\int q_*(y) \exp\left(-\frac{1}{2\sigma_t^2}\|m_t y - x\|^2\right) y\mathrm{d}y\right)\left(\int q_*(y) \exp\left(-\frac{1}{2\sigma_t^2}\|m_t y - x\|^2\right) y\mathrm{d}y\right)^\top}{\left(\int q_*(y) \exp\left(-\frac{1}{2\sigma_t^2}\|m_t y - x\|^2\right) \mathrm{d}y\right)^2}$$
$$= \frac{m_t^2}{\sigma_t^4} \mathrm{Var}_{q_{0|t}(y|x)}(y),$$

where $\mathrm{Var}_{q_{0|t}(y|x)}(y)$ means the variance of $X_0$ conditioned on $X_t = x$, with respect to $q_t$.

Thus, bounding $\mathrm{Var}_{q_{0|t}(y|x)}(x)$ yields the conclusion. Lemma 9 implies that

$$\mathrm{Var}_{q_{0|t}(y|x)}(y) \leq C_\rho^2 \mathrm{Var}_{p_{0|t}(y|x)}(y). \tag{37}$$

Similarly to $\nabla_x^2 \log q_t(x)$, $\nabla_x^2 \log p_t(x)$ satisfies

$$\nabla_x^2 \log p_t(x) = -\sigma_t^{-2} I_d + \frac{m_t^2}{\sigma_t^4} \mathrm{Var}_{p_{0|t}(y|x)}(y). \tag{38}$$

By combining (37) and (38), we have

$$\frac{m_t^2}{\sigma_t^4} \mathrm{Var}_{q_{0|t}(y|x)}(y) \leq C_\rho^2 \sigma_t^{-2} I_d + C_\rho^2 \nabla_x^2 \log p_t(x).$$

Therefore, from the assumption that $\nabla_x \log p_t(x)$ is $L_p$-Lipschitz and $\sigma_t^{-2} \geq 2(L_p + L_\rho)$, we obtain that $\|\frac{m_t^2}{\sigma_t^4} \mathrm{Var}_{q_{0|t}(y|x)}(y)\| \leq (2 + C_\rho^2)L_p + 2L_\rho$.

By putting it all together, $\nabla \log q_t$ is $((4 + C_\rho^2)L_p + 4L_\rho)$-Lipschitz. □

**Lemma 9.** *When $C_\rho^{-1} \leq h_*(x) \leq C_\rho$, we have*

$$\frac{q_{0|t}(x|x')}{p_{0|t}(x|x')} \leq C_\rho^2$$

*for all $x, x'$ and $t$.*

*Proof.* We can write $q_{0|t}(x|x')$ as

$$q_{0|t}(x|x') = \frac{q_{0,t}(x,x')}{\int q_0(x'')q_{t|0}(x|x'')dx''} = \frac{p_*(x)\rho_*(x)q_{t|0}(x'|x)}{\int p_*(x'')\rho_*(x'')q_{t|0}(x|x'')dx''}.$$

Because $C_\rho^{-1} \leq \rho_*(x) \leq C_\rho$ and $q_{t|0}(x'|x) = p_{t|0}(x'|x)$, we have

$$q_{0|t}(x|x') \leq C_\rho^2 \frac{p_*(x)p_{t|0}(x'|x)}{\int p_*(x'')p_{t|0}(x|x'')dx''} = C_\rho^2 p_{0|t}(x|x'),$$

which concludes the proof. □

## C.3 ESTIMATION ERROR OF THE CORRECTION TERM

Since we have shown the time discretization error in the previous section, what we remain to show is just an upper bound of $\varepsilon_{\rho,l}^2$. For that purpose, we put an additional assumption which is almost same as Assumption 4 except the condition (iii). A bound of $\varepsilon_{\mathrm{TV}}^2$ in the third condition (iii) will be given as $\varepsilon_{\mathrm{TV}}^2 = \mathcal{O}(\varepsilon^2 + h)$ by Chen et al. (2023c).

**Assumption 7.** *(i) $\nabla_x s(\cdot, \cdot)$ is $H_s$-Lipschitz continuous in a sense that $\|\nabla_x s(x,t) - \nabla_y s(y,t)\|_{\mathrm{op}} \leq H_s \|x - y\|$ for any $x, y \in \mathbb{R}^d$ and $0 \leq t \leq T$ and $\mathbb{E}[\|s(\bar{X}_{kh}^\leftarrow, kh)\|^2] \leq Q^2$ for any $k$.*

*(ii) There exists $R > 0$ such that $\sup_{t,x}\{\|\nabla_x^2 \log p_t(x)\|_{\mathrm{op}}, \|\nabla_x^2 \log s(x,t)\|_{\mathrm{op}}\} \leq R$.*

*(iii) $\mathbb{E}_{\bar{X}_t}[\mathrm{TV}(\bar{X}_T^\leftarrow, X_T^\leftarrow | \bar{X}_{T-t}^\leftarrow = X_{T-t}^\leftarrow = \bar{X}_t)^2] \leq \varepsilon_{\mathrm{TV}}^2$ for any $t \in [0, T]$.*

**Theorem 7.** *Suppose that $0 \leq h \leq \delta \leq 1/(1 + 2R)$ and Assumptions 7 and 5 hold. Let $L_\varphi$ and $R_\varphi$ be as given in Lemma 10. Then, it holds that*

$$\mathbb{E}_{\bar{X}_\cdot^\leftarrow}[\|\nabla_x \mathbb{E}[\rho_*(\bar{X}_T^\leftarrow)|\bar{X}_t] - \nabla_x \mathbb{E}[\rho_*(X_T^\leftarrow)|X_{kh}^\leftarrow]\|^2]$$
$$\lesssim R_\varphi^2 \left(\varepsilon^2 + L_p^2 d(\delta + \mathsf{m}\delta^2)\right) + \Xi_{\delta,\varepsilon} + [L_\varphi^2(\mathsf{m} + 4Q^2 + dh) + R_\varphi^2(1 + 2R)^2]h^2$$
$$= \mathcal{O}\left(\varepsilon^2 + \delta + \frac{\varepsilon_{\mathrm{TV}}^2}{\delta}\right),$$

*where*

$$\Xi_{\delta,\varepsilon} := \frac{4c_\eta^2 C_\rho^2(1 + 2R)^2}{3}\delta + 2\exp(2)\left\{C_\rho^2 \frac{\varepsilon_{\mathrm{TV}}^2}{\delta} + CR_\varphi^2[\varepsilon^2 + L_p^2 d(\delta + \mathsf{m}\delta^2)]\right\},$$

*and $c_\eta > 0$ is a universal constant.*

By Chen et al. (2023c), $\varepsilon_{\mathrm{TV}}^2 = \mathcal{O}(\varepsilon^2 + h)$, and thus by substituting $\delta \leftarrow \sqrt{h}$, we finally obtain an error estimate as

$$\left(1 + \frac{1}{\sqrt{h}}\right)\varepsilon^2 + \sqrt{h}.$$

*Proof.* Let $t \in [kh, (k+1)h)$ and $t^* = kh + \delta$ where $\delta$ is larger than or equal to $h$: $\delta \geq h$. We only consider a situation where $T - t \geq \delta$. The situation where $\delta < T - t$ can be treated in the same manner by noticing a trivial relation $\nabla_x \mathbb{E}[\rho_*(\bar{X}_T^{\leftarrow})|\bar{X}_T^{\leftarrow} = x] = \nabla \rho_*(x)$. Then, for a given initial state $x \in \mathbb{R}^d$, we define the stochastic processes as

$$\bar{X}_t^{\leftarrow} = x, \ \mathrm{d}\bar{X}_\tau^{\leftarrow} = \{\bar{X}_\tau^{\leftarrow} + 2\nabla \log p_{T-\tau}(\bar{X}_\tau^{\leftarrow})\}\mathrm{d}\tau + \sqrt{2}\mathrm{d}B_\tau, \quad (t \leq \tau \leq T),$$

and its numerical approximation as

$$X_t^{\leftarrow} = x, \ \mathrm{d}X_\tau^{\leftarrow} = \{X_\tau^{\leftarrow} + 2s(X_{k_\tau h}^{\leftarrow}, k_\tau h)\}\mathrm{d}\tau + \sqrt{2}\mathrm{d}B_\tau. \quad (t \leq \tau \leq T),$$

where $k_\tau$ is the integer such that $\tau \in [k_\tau h, (k_\tau + 1)h)$.

Note that

$$\begin{aligned}
&\nabla_x \mathbb{E}[\rho_*(\bar{X}_T^{\leftarrow})|\bar{X}_t] - \nabla_x \mathbb{E}[\rho_*(X_T^{\leftarrow})|X_{kh}^{\leftarrow}] \\
&= \underbrace{(\nabla_x \mathbb{E}[\rho_*(\bar{X}_T^{\leftarrow})|\bar{X}_t] - \nabla_x \mathbb{E}[\rho_*(X_T^{\leftarrow})|X_t^{\leftarrow}])}_{(a)} + \underbrace{(\nabla_x \mathbb{E}[\rho_*(X_T^{\leftarrow})|X_t^{\leftarrow}] - \nabla_x \mathbb{E}[\rho_*(X_T^{\leftarrow})|X_{kh}^{\leftarrow}])}_{(b)}.
\end{aligned}$$
$$(39)$$

We first evaluate the term (a):

$$\nabla_x \mathbb{E}[\rho_*(\bar{X}_T^{\leftarrow})|\bar{X}_t^{\leftarrow} = x] - \nabla_x \mathbb{E}[\rho_*(X_T^{\leftarrow})|X_t^{\leftarrow} = x].$$

As we have seen above, the derivative can be expressed by the following recursive formula of the conditional expectation:

$$\nabla_x \mathbb{E}[\mathbb{E}[\rho_*(\bar{X}_T^{\leftarrow})|\bar{X}_{t^*}]|\bar{X}_t^{\leftarrow} = x].$$

For a notation simplicity, we let $\varphi_X(x) := \mathbb{E}[\rho_*(\bar{X}_T^{\leftarrow})|\bar{X}_{t^*}^{\leftarrow} = x]$ and $\varphi_Y(x) := \mathbb{E}[\rho_*(X_T^{\leftarrow})|X_{t^*}^{\leftarrow} = x]$. Then, the Bismut-Elworthy-Li formula (Bismut, 1984; Elworthy & Li, 1994) yields that, for any $v \in \mathbb{R}^d$,

$$v^\top \nabla_x \mathbb{E}[\rho_*(\bar{X}_0^{\leftarrow})|\bar{X}_t^{\leftarrow} = x] = \mathbb{E}\left[\frac{1}{\delta}\int_0^\delta \langle \eta_{\bar{X},\tau}, \mathrm{d}B_\tau \rangle \varphi_X(\bar{X}_{t^*}^{\leftarrow}) \mid \bar{X}_t^{\leftarrow} = x\right],$$

where $\eta_{\bar{X},\tau}$ is the solution of

$$\begin{aligned}
\mathrm{d}\eta_{\bar{X},\tau} &= (I + 2\nabla^2 \log p_{T-t-\tau}(\bar{X}_{t+\tau}^{\leftarrow}))\eta_{\bar{X},\tau}\mathrm{d}\tau, \\
\eta_{\bar{X},0} &= v.
\end{aligned}$$

Similarly, we define $\eta_{X,\tau}$ for the process $X_\tau^{\leftarrow}$ as

$$\begin{aligned}
\mathrm{d}\eta_{X,\tau} &= (\eta_{X,\tau} + 2\nabla_x^\top s(X_{k_\tau h}^{\leftarrow}, k_\tau h)\eta_{X,k_\tau h-t})\mathrm{d}\tau, \\
\eta_{X,0} &= v.
\end{aligned}$$

Then,

$$\begin{aligned}
&v^\top \nabla_x \mathbb{E}[\rho_*(\bar{X}_T^{\leftarrow})|\bar{X}_t^{\leftarrow} = x] - v^\top \nabla_x \mathbb{E}[\rho_*(X_T^{\leftarrow})|X_t^{\leftarrow} = x] \\
&= \mathbb{E}\left[\frac{1}{\delta}\int_0^\delta \langle \eta_{\bar{X},\tau} - \eta_{X,\tau}, \mathrm{d}B_\tau \rangle \varphi_Y(X_{t^*}^{\leftarrow}) \mid \bar{X}_t^{\leftarrow} = X_t^{\leftarrow} = x\right] \\
&\quad + \mathbb{E}\left[\frac{1}{\delta}\int_0^\delta \langle \eta_{\bar{X},\tau}, \mathrm{d}B_\tau \rangle (\varphi_X(\bar{X}_{t^*}^{\leftarrow}) - \varphi_Y(X_{t^*}^{\leftarrow})) \mid \bar{X}_t^{\leftarrow} = X_t^{\leftarrow} = x\right].
\end{aligned}$$
$$(40)$$

By the Ito isometry, the first term of the right hand side can be bounded as

$$\left(\mathbb{E}\left[\frac{1}{\delta}\int_0^\delta\langle\eta_{\bar{X},\tau}-\eta_{X,\tau},\mathrm{d}B_\tau\rangle\varphi_Y(X_{t^*})\mid\bar{X}_t^\leftarrow=X_t^\leftarrow=x\right]\right)^2$$

$$\leq\mathbb{E}\left[\left(\frac{1}{\delta}\int_0^\delta\langle\eta_{\bar{X},\tau}-\eta_{X,\tau},\mathrm{d}B_\tau\rangle\varphi_Y(X_{t^*})\right)^2\mid\bar{X}_t^\leftarrow=X_t^\leftarrow=x\right]$$

$$\leq C_\rho^2\mathbb{E}\left[\left(\frac{1}{\delta}\int_0^\delta\langle\eta_{\bar{X},\tau}-\eta_{X,\tau},\mathrm{d}B_\tau\rangle\right)^2\mid\bar{X}_t^\leftarrow=X_t^\leftarrow=x\right]$$

$$=C_\rho^2\mathbb{E}\left[\frac{1}{h^2}\int_\tau^h\|\eta_{\bar{X},\tau}-\eta_{X,\tau}\|^2\mathrm{d}\tau\mid\bar{X}_t^\leftarrow=X_t^\leftarrow=x\right]$$

$$\leq2C_\rho^2\mathbb{E}\left[\frac{1}{\delta^2}\int_0^\delta(\|\eta_{\bar{X},\tau}-v\|^2+\|\eta_{X,\tau}-v\|^2)\mathrm{d}\tau\mid\bar{X}_t^\leftarrow=X_t^\leftarrow=x\right].$$

Hence, we just need to bound $\|\eta_{\bar{X},\tau}-v\|^2$ in the right hand side. We note that it obeys the following differential equation:

$$\frac{\mathrm{d}\|\eta_{\bar{X},\tau}-v\|^2}{\mathrm{d}\tau}$$

$$=2(\eta_{\bar{X},\tau}-v)^\top\frac{\mathrm{d}\eta_{\bar{X},\tau}}{\mathrm{d}\tau}$$

$$=2(\eta_{\bar{X},\tau}-v)^\top(I+2\nabla^2\log p_{T-t-\tau}(\bar{X}_{T-t-\tau}^\leftarrow))\eta_{\bar{X},\tau}$$

$$=2(\eta_{\bar{X},\tau}-v)^\top(I+2\nabla^2\log p_{T-t-\tau}(\bar{X}_{T-t-\tau}^\leftarrow))[(\eta_{\bar{X},\tau}-v)+v]$$

$$\leq2(1+2R)\|\eta_{\bar{X},\tau}-v\|^2+2(1+2R)\|v\|\|\eta_{\bar{X},\tau}-v\|,$$

which also yields that

$$2\|\eta_{\bar{X},\tau}-v\|\frac{\mathrm{d}\|\eta_{\bar{X},\tau}-v\|}{\mathrm{d}\tau}\leq2(1+2R)\|\eta_{\bar{X},\tau}-v\|^2+2(1+2R)\|v\|\|\eta_{\bar{X},\tau}-v\|$$

$$\Rightarrow\frac{\mathrm{d}\|\eta_{\bar{X},\tau}-v\|}{\mathrm{d}\tau}\leq2(1+2R)(\|\eta_{\bar{X},\tau}-v\|+\|v\|)$$

$$\Rightarrow\|\eta_{\bar{X},\tau}-v\|\leq[\exp(2(1+2R)\tau)-1]\|v\|$$

$$\Rightarrow\|\eta_{\bar{X},\tau}-v\|^2\leq[\exp(2(1+2R)\tau)-1]^2\|v\|^2.$$

Therefore, if $\delta$ is sufficiently small (such as $\delta\leq1/(1+2R)$), then we arrive at

$$\|\eta_{\bar{X},\tau}-v\|^2\leq c_\eta(1+2R)^2\tau^2\|v\|^2,$$

with a universal constant $c_\eta$, for any $0\leq\tau\leq\delta$. In the same vein, we also have

$$\|\eta_{X,\tau}-v\|^2\leq c_\eta(1+2R)^2\tau^2\|v\|^2,$$

$$\|\eta_{\bar{X},\tau}\|^2\leq\exp(4(1+2R)\tau)\|v\|^2,\ \|\eta_{X,\tau}\|^2\leq\exp(4(1+2R)\tau)\|v\|^2,$$

for $0\leq\tau\leq\delta$. These bounds yield that

$$2C_\rho^2\mathbb{E}\left[\frac{1}{\delta^2}\int_0^\delta(\|\eta_{\bar{X},\tau}-v\|^2+\|\eta_{X,\tau}-v\|^2)\mathrm{d}\tau\mid\bar{X}_t^\leftarrow=X_t^\leftarrow=x\right]$$

$$\leq4c_\eta^2C_\rho^2(1+2R)^2\frac{\int_0^\delta\tau^2\mathrm{d}\tau}{\delta^2}=\frac{4c_\eta^2C_\rho^2(1+2R)^2}{3}\delta.$$

Next, we bound the second term of the right hand side in Eq. (40):

$$
\mathbb{E}\left[\frac{1}{\delta}\int_0^\delta \langle \eta_{\bar{X},\tau}, \mathrm{d}B_\tau\rangle(\varphi_X(\bar{X}_{t^*}^\leftarrow) - \varphi_Y(X_{t^*}^\leftarrow)) \mid \bar{X}_t^\leftarrow = X_t^\leftarrow = x\right]^2
$$

$$
\leq \mathbb{E}\left[\frac{1}{\delta^2}\int_0^\delta \|\eta_{\bar{X},\tau}\|^2 \mathrm{d}\tau \mid \bar{X}_t^\leftarrow = X_t^\leftarrow = x\right]\mathbb{E}\left[(\varphi_X(\bar{X}_{t^*}^\leftarrow) - \varphi_Y(X_{t^*}^\leftarrow))^2 \mid \bar{X}_t^\leftarrow = X_t^\leftarrow = x\right]
$$

$$
\leq \frac{\exp(2(1+2R)\delta)\|v\|^2}{\delta}\mathbb{E}\left[(\varphi_X(\bar{X}_{t^*}^\leftarrow) - \varphi_Y(X_{t^*}^\leftarrow))^2 \mid \bar{X}_t^\leftarrow = X_t^\leftarrow = x\right]. \tag{41}
$$

Otherwise, we also have the following inequality:

$$
\mathbb{E}\left[\frac{1}{\delta}\int_0^\delta \langle \eta_{\bar{X},\tau}, \mathrm{d}B_\tau\rangle(\varphi_X(\bar{X}_{t^*}^\leftarrow) - \varphi_Y(X_{t^*}^\leftarrow)) \mid \bar{X}_t^\leftarrow = X_t^\leftarrow = x\right]^2
$$

$$
\leq 2\mathbb{E}\left[\frac{1}{\delta}\int_0^\delta \langle \eta_{\bar{X},\tau}, \mathrm{d}B_\tau\rangle(\varphi_Y(\bar{X}_{t^*}^\leftarrow) - \varphi_Y(X_{t^*}^\leftarrow)) \mid \bar{X}_t^\leftarrow = X_t^\leftarrow = x\right]^2
$$

$$
+ 2\mathbb{E}\left[\eta_{\bar{X},\delta}^\top \nabla(\varphi_X(\bar{X}_{t^*}^\leftarrow)) - \varphi_Y(\bar{X}_{t^*}^\leftarrow)) \mid \bar{X}_t^\leftarrow = X_t^\leftarrow = x\right]^2
$$

$$
\leq \frac{\exp(2(1+2R)\delta)\|v\|^2}{\delta}\mathbb{E}\left[(\varphi_Y(\bar{X}_{t^*}^\leftarrow) - \varphi_Y(X_{t^*}^\leftarrow))^2 \mid \bar{X}_t^\leftarrow = X_t^\leftarrow = x\right]
$$

$$
+ 2\mathbb{E}\left[\|\eta_{\bar{X},\delta}\|\|\nabla(\varphi_X(\bar{X}_{t^*}^\leftarrow)) - \varphi_Y(\bar{X}_{t^*}^\leftarrow))\| \mid \bar{X}_t^\leftarrow = X_t^\leftarrow = x\right]^2. \tag{42}
$$

For bounding these quantities, we need to bound the discrepancy $\|\bar{X}_\tau^\leftarrow - X_\tau^\leftarrow\|^2$. Note that this quantity follows the following ODE:

$$
\frac{\mathrm{d}\|\bar{X}_\tau^\leftarrow - X_\tau^\leftarrow\|^2}{\mathrm{d}\tau}
$$

$$
= 2(\bar{X}_\tau^\leftarrow - X_\tau^\leftarrow)^\top[(\bar{X}_\tau^\leftarrow - 2\nabla_x \log p_{T-\tau-t}(\bar{X}_\tau^\leftarrow)) - (X_\tau^\leftarrow - 2s(x,kh))]
$$

$$
= 2\|\bar{X}_\tau^\leftarrow - X_\tau^\leftarrow\|^2 - 4(\bar{X}_\tau^\leftarrow - X_\tau^\leftarrow)^\top(\nabla_x \log p_{T-\tau-t}(\bar{X}_\tau^\leftarrow) - s(x,kh))
$$

$$
\leq 4\|\bar{X}_\tau^\leftarrow - X_\tau^\leftarrow\|^2 + 2\|\nabla_x \log(p_{T-\tau-t}(\bar{X}_\tau^\leftarrow)) - s(x,kh)\|^2.
$$

Therefore, it satisfies that

$$
\|\bar{X}_\tau^\leftarrow - X_\tau^\leftarrow\|^2 \leq 4\int_0^\tau \|\bar{X}_s^\leftarrow - X_s^\leftarrow\|^2 \mathrm{d}s + 2\int_0^\tau \|\nabla_x \log(p_{T-\tau-t}(\bar{X}_s^\leftarrow)) - s(x,kh)\|^2 \mathrm{d}s.
$$

Taking its expectation, we see that

$$
\mathbb{E}[\|\bar{X}_\tau^\leftarrow - X_\tau^\leftarrow\|^2] \leq 4\int_0^\tau \mathbb{E}[\|\bar{X}_s^\leftarrow - X_s^\leftarrow\|^2]\mathrm{d}s + 2\underbrace{\int_0^\tau (\varepsilon^2 + L_p^2 ds + L_p^2 \mathrm{m}s^2)\mathrm{d}s}_{=\mathcal{O}(\varepsilon^2\tau + L_p^2 d(\tau^2 + \mathrm{m}\tau^3))=:\xi(\tau)},
$$

where we used Theorem 10 (and its proof) of Chen et al. (2023b) for obtaining $\xi(\tau)$. Then, Gronwall inequality yields

$$
\mathbb{E}[\|\bar{X}_\tau^\leftarrow - X_\tau^\leftarrow\|^2] \leq \xi(\tau) + \int 4\xi(s)e^{4(\tau-s)}\mathrm{d}s \lesssim \varepsilon^2\tau + L_p^2 d(\tau^2 + \mathrm{m}\tau^3),
$$

(see Mischler (2019) for example). Then, the Lipschitz continuity of $\varphi_Y$ (Lemma 10) yields that

$$
\mathbb{E}\left[(\varphi_Y(\bar{X}_{t^*}^\leftarrow) - \varphi_Y(X_{t^*}^\leftarrow))^2 \mid \bar{X}_t^\leftarrow = X_t^\leftarrow = x\right] \lesssim L_\varphi^2[\varepsilon^2\tau + L_p^2 d(\tau^2 + \mathrm{m}\tau^3)].
$$

**Bound for $t = kh$:** First, we show a bound for $t = kh$. The right hand side of Eq. (41) with $\delta = h$ can be bounded by

$$
\frac{\exp(2(1+2R)\delta)\|v\|^2}{\delta}\mathbb{E}_{\bar{X}_t^\leftarrow}\left[\mathbb{E}\left[(\varphi_X(\bar{X}_{t^*}^\leftarrow) - \varphi_Y(X_{t^*}^\leftarrow))^2 \mid \bar{X}_t^\leftarrow = X_t^\leftarrow]\right]\right]
$$

$$
\leq \frac{\exp(2(1+2R)\delta)\|v\|^2}{\delta}2\mathbb{E}_{\bar{X}_t^\leftarrow}\left[\mathbb{E}\left[(\varphi_X(\bar{X}_{t^*}^\leftarrow) - \varphi_Y(\bar{X}_{t^*}^\leftarrow))^2 + (\varphi_Y(\bar{X}_{t^*}^\leftarrow) - \varphi_Y(X_{t^*}^\leftarrow))^2 \mid \bar{X}_t^\leftarrow = X_t^\leftarrow]\right]\right]
$$

$$
\leq \frac{\exp(2(1+2R)\delta)\|v\|^2}{\delta}2\left\{C_\rho^2\varepsilon_{\mathrm{TV}}^2 + R_\varphi^2\mathbb{E}_{\bar{X}_t^\leftarrow}\left[\mathbb{E}\left[(\bar{X}_{t^*}^\leftarrow - X_{t^*}^\leftarrow)^2 \mid \bar{X}_t^\leftarrow = X_t^\leftarrow]\right]\right]\right\},
$$

where we used
$$\mathbb{E}\left[(\varphi_X(\bar{X}_{t^*}^{\leftarrow}) - \varphi_Y(\bar{X}_{t^*}^{\leftarrow}))^2\right] \leq C_\rho^2 \mathbb{E}_{\bar{X}_{T-t^*}}\left[\mathrm{TV}(\bar{X}_T^{\leftarrow}, X_T^{\leftarrow}|X_{t^*}^{\leftarrow} = \bar{X}_{t^*}^{\leftarrow} = \bar{X}_{T-t^*}^{\leftarrow})^2\right]$$
$$\leq C_\rho^2 \varepsilon_{\mathrm{TV}}.$$

Here, by using Theorem 10 of Chen et al. (2023c) again, the right hand side can be bounded as
$$2\frac{\exp(2(1+2R)\delta)\|v\|^2}{\delta}\left\{C_\rho^2 \varepsilon_{\mathrm{TV}}^2 + CR_\varphi^2[\varepsilon^2\delta + L_p^2 d(\delta^2 + \mathsf{m}\delta^3)]\right\}$$
$$= 2\exp(2(1+2R)\delta)\|v\|^2\left\{C_\rho^2 \frac{\varepsilon_{\mathrm{TV}}^2}{\delta} + CR_\varphi^2[\varepsilon^2 + L_p^2 d(\delta + \mathsf{m}\delta^2)]\right\}.$$

Then, with the contraint $h \leq 1/(1+2R)$, it can be further simplified as
$$2\exp(2)\|v\|^2\left\{C_\rho^2 \frac{\varepsilon_{\mathrm{TV}}^2}{\delta} + CR_\varphi^2[\varepsilon^2 + L_p^2 d(\delta + \mathsf{m}\delta^2)]\right\}.$$

Therefore, by taking maximum with respect to $v \in \mathbb{R}^d$ with a constraint $\|v\| = 1$,
$$\mathbb{E}_{\bar{X}_t^{\leftarrow}}[\|\nabla_x \mathbb{E}[\rho_*(\bar{X}_T^{\leftarrow})|\bar{X}_t^{\leftarrow}] - \nabla_x \mathbb{E}[\rho_*(X_T^{\leftarrow})|X_t^{\leftarrow} = \bar{X}_t^{\leftarrow}]\|^2]$$
$$\leq \frac{4c_\eta^2 C_\rho^2 (1+2R)^2}{3}\delta + 2\exp(2)\left\{C_\rho^2 \frac{\varepsilon_{\mathrm{TV}}^2}{\delta} + CR_\varphi^2[\varepsilon^2 + L_p^2 d(\delta + \mathsf{m}\delta^2)]\right\} =: \Xi_{\delta,\varepsilon}. \quad (43)$$

We see that $\Xi_{\delta,\varepsilon} = \mathcal{O}(\delta + \varepsilon^2 + \varepsilon_{\mathrm{TV}}^2/\delta)$.

**Bound for general** $t \in (kh, (k+1)h)$**:** In this setting, we utilize the inequality (42). Using the constraint $\delta \leq 1/(1+2R)$ and $\|v\| = 1$, the right hand side of (42) can be bounded by
$$\frac{\exp(2)}{\delta}\mathbb{E}\left[(\varphi_Y(\bar{X}_{t^*}^{\leftarrow}) - \varphi_Y(X_{t^*}^{\leftarrow}))^2 \mid \bar{X}_t^{\leftarrow} = X_t^{\leftarrow} = x\right]$$
$$+ 2\mathbb{E}\left[\exp(2)\|\nabla(\varphi_X(\bar{X}_{t^*}^{\leftarrow})) - \varphi_Y(\bar{X}_{t^*}^{\leftarrow}))\| \mid \bar{X}_t^{\leftarrow} = X_t^{\leftarrow} = x\right]^2$$
$$\leq \frac{\exp(2)}{\delta}R_\varphi^2 \mathbb{E}\left[(\bar{X}_{t^*}^{\leftarrow} - X_{t^*}^{\leftarrow})^2 \mid \bar{X}_t^{\leftarrow} = X_t^{\leftarrow} = x\right] + 2\exp(2)\Xi_{\delta,\varepsilon} \quad (\because Eq.~(43)).$$

By taking the expectation with respect to $x = \bar{X}_t^{\leftarrow}$, we arrive at
$$\mathbb{E}_{\bar{X}_t^{\leftarrow}}[\|\nabla_x \mathbb{E}[\rho_*(\bar{X}_T^{\leftarrow})|\bar{X}_t^{\leftarrow}] - \nabla_x \mathbb{E}[\rho_*(X_T^{\leftarrow})|X_t^{\leftarrow}]\|^2]$$
$$\leq C\frac{\exp(2)}{\delta}R_\varphi^2\left(\varepsilon^2\delta + L_p^2 d(\delta^2 + \mathsf{m}\delta^3)\right) + 2\exp(2)\Xi_{\delta,\varepsilon}$$
$$\leq C\exp(2)R_\varphi^2\left(\varepsilon^2 + L_p^2 d(\delta + \mathsf{m}\delta^2)\right) + 2\exp(2)\Xi_{\delta,\varepsilon}. \quad (44)$$

This gives an upper bound of the term (a) in Eq. (39). Then, we just need to bound the remaining term (b) in Eq. (39):
$$\mathbb{E}_{\bar{X}_t^{\leftarrow}, \bar{X}_{kh}^{\leftarrow}}[\|\nabla_x \mathbb{E}[\rho_*(X_T^{\leftarrow})|X_t^{\leftarrow}] - \nabla_x \mathbb{E}[\rho_*(X_T^{\leftarrow})|X_{kh}^{\leftarrow}]\|^2].$$

For that purpose, we define $\varphi_{Y,t}(x) = \mathbb{E}[\rho_*(X_T^{\leftarrow})|X_t^{\leftarrow} = x]$. Then, using the Bismut-Elworthy-Li formula again,
$$v^\top(\nabla\varphi_{Y,t}(x) - \nabla\varphi_{Y,kh}(x))$$
$$= v^\top \nabla\varphi_{Y,t}(x) - \mathbb{E}[\eta_{X,h(k+1)-t}^\top \nabla\varphi_{Y,t}(X_t^{\leftarrow})|X_{kh}^{\leftarrow} = x]$$
$$= \mathbb{E}[v^\top(\nabla\varphi_{Y,t}(x) - \nabla\varphi_{Y,t}(X_t^{\leftarrow})) + (\eta_{X,(k+1)h-t}^\top - v)\nabla\varphi_{Y,t}(X_t^{\leftarrow})|X_{kh}^{\leftarrow} = x]$$
$$\leq \mathbb{E}[L_\varphi\|x - X_t^{\leftarrow}\| + R_\varphi\|\eta_{X,(k+1)h-t}^\top - v\| \mid X_{(k+1)h-t}^{\leftarrow} = x]$$
$$\leq \mathbb{E}[L_\varphi\|((k+1)h-t)(x - 2s(x,kh)) + \sqrt{(h-\delta)}B_{kh}\| + R_\varphi\|\eta_{X,(k+1)h-t}^\top - v\| \mid X_{kh}^{\leftarrow} = x],$$

which yields that
$$\mathbb{E}_{\bar{X}^{\leftarrow}}[\|\nabla\varphi_{Y,t}(\bar{X}_t^{\leftarrow}) - \nabla\varphi_{Y,kh}(\bar{X}_{(k+1)h-t}^{\leftarrow})\|^2]$$
$$\leq 2L_\varphi^2((k+1)h-t)^2\mathbb{E}_{\bar{X}^{\leftarrow}}[\|\bar{X}_{kh}^{\leftarrow}\|^2 + 4\|s(\bar{X}_{kh}^{\leftarrow}, kh)\|^2 + d((k+1)h-t)]$$
$$+ R_\varphi^2 c_\eta(1+2R)^2((k+1)h-t)^2$$
$$\leq 2L_\varphi^2 h^2(\mathsf{m} + 4Q^2 + dh) + R_\varphi^2 c_\eta(1+2R)^2 h^2$$
$$= [2L_\varphi^2(\mathsf{m} + 4Q^2 + dh) + R_\varphi^2 c_\eta(1+2R)^2]h^2. \quad (45)$$

Combining (44) and (45) gives the assertion. $\qquad\square$

**Lemma 10.** *Suppose that* $\sup_x \|\nabla\rho_*(x)\| \leq R_\rho$, $\|\nabla\rho_*(x) - \nabla\rho_*(y)\| \leq L_\rho\|x - y\|$ ($\forall x, y$), *and* $\nabla_x s(\cdot, \cdot)$ *is* $H_s$-*Lipschitz continuous with respect to* $x$. *Let* $\varphi_{Y,t}(x) = \mathbb{E}[\rho_*(X_T^{\leftarrow})|X_t^{\leftarrow} = x]$. *Then,* $\nabla_x\varphi_{Y,t}(x)$ *is bounded by* $R_\varphi$ *and* $L_\varphi$-*Lipschitz continuous for any* $0 \leq t \leq T$, *where*

$$R_\varphi = \max\{C_\rho 2\sqrt{(1 + 2R)e}, e^{1/2}R_\rho\},$$

$$L_\varphi = \max\left\{\left(\frac{2C_\eta^2 H_s^2 C_\rho^2}{1 + 2R} + 2(1 + 2R)\exp(6)R_\varphi^2\right)^{1/2}, \left(2C_\eta^2 H_s^2 R^2 + e^2 L_\rho^2\right)^{1/2}\right\},$$

*for a universal constant* $C_\eta > 0$.

*Proof.* We show it only when $t = kh$ for a positive integer $k$ just for simplicity. The proof for a general $t$ can be obtained in the same manner.

(i) First, we assume that $T - t \geq 1/4(1 + 2R)$. In the following, we let $v \in \mathbb{R}^d$ be an arbitrary vector with $\|v\| = 1$. We again utilize the Bismut-Elworthy-Li formula:

$$v^\top \nabla\varphi_{Y,t}(x)$$
$$= v^\top \nabla_x\mathbb{E}[\rho_*(X_T^{\leftarrow}) \mid X_t^{\leftarrow} = x]$$
$$= \mathbb{E}\left[\frac{1}{S}\int_0^S \langle\eta_\tau, \mathrm{d}B_\tau\rangle\varphi_{Y,S}(X_S^{\leftarrow}) \mid X_t^{\leftarrow} = x\right].$$

Hence,

$$(v^\top \nabla\varphi_{Y,t}(x))^2$$
$$\leq C_\rho^2 \mathbb{E}\left[\frac{1}{S^2}\left(\int_0^S \langle\eta_\tau, \mathrm{d}B_\tau\rangle\right)^2 \mid X_t^{\leftarrow} = x\right]$$
$$\leq C_\rho^2 \mathbb{E}\left[\frac{1}{S^2}\int_0^S \|\eta_\tau\|^2\mathrm{d}\tau \mid X_t^{\leftarrow} = x\right].$$

Here, we know that $\|\eta_\tau\|^2 \leq \exp(4(1 + 2R)\tau)\|v\|^2$, and thus

$$(v^\top \nabla\varphi_{Y,t}(x))^2 \leq C_\rho^2 \frac{1}{S}\exp(4(1 + 2R)S)\|v\|^2.$$

Hence, by taking $S = \frac{1}{4(1+2R)}$, we have that

$$(v^\top \nabla\varphi_{Y,t}(x))^2 \leq C_\rho^2 4(1 + 2R)e\|v\|^2.$$

This shows that $\|\nabla\varphi_{Y,t}(x)\|$ is bounded by $R_\varphi = C_\rho 2\sqrt{(1 + 2R)e}$.

Next, we show its Lipschitz continuity. For that purpose, we define two stochastic processes

$$X_t^{\leftarrow} = x, \; \mathrm{d}X_\tau^{\leftarrow} = \{X_\tau^{\leftarrow} + 2s(X_{kh}^{\leftarrow}, kh)\}\mathrm{d}\tau + \sqrt{2}\mathrm{d}B_\tau \; (\tau \in [kh, k(h+1)]),$$
$$\tilde{Z}_t^{\leftarrow} = y, \; \mathrm{d}\tilde{Z}_\tau^{\leftarrow} = \{\tilde{Z}_\tau^{\leftarrow} + 2s(\tilde{Z}_{kh}^{\leftarrow}, kh)\}\mathrm{d}\tau + \sqrt{2}\mathrm{d}B_\tau \; (\tau \in [kh, k(h+1)]),$$

where $x, y \in \mathbb{R}^d$ with $\|x - y\| \leq \varepsilon$. Accordingly, we also define

$$\eta_{X,0} = v, \; \frac{\mathrm{d}\eta_{X,\tau}}{\mathrm{d}\tau} = (I + 2\nabla_x^\top s(X_{kh}^{\leftarrow}, kh))\eta_{X,\tau},$$

$$\eta_{Z,0} = v, \; \frac{\mathrm{d}\eta_{Z,\tau}}{\mathrm{d}\tau} = (I + 2\nabla_x^\top s(\tilde{Z}_{kh}^{\leftarrow}, kh))\eta_{Z,\tau}.$$

Thus,

$$X_{(k+1)h}^{\leftarrow} - \tilde{Z}_{(k+1)h}^{\leftarrow} = X_{kh}^{\leftarrow} - \tilde{Z}_{kh}^{\leftarrow} + h[X_{kh}^{\leftarrow} - \tilde{Z}_{kh}^{\leftarrow} + 2(s(X_{kh}^{\leftarrow}, kh) - s(\tilde{Z}_{kh}^{\leftarrow}, kh))],$$

which yields

$$\|X_{(k+1)h}^{\leftarrow} - \tilde{Z}_{(k+1)h}^{\leftarrow}\| \leq (1 + h(1 + R))\|X_{kh}^{\leftarrow} - \tilde{Z}_{kh}^{\leftarrow}\|$$
$$\leq (1 + h(1 + R))^{k+1}\|x - y\|.$$

Now, we assume $k \leq S/h$ so that we have $\|X^{\leftarrow}_{(k+1)h} - \tilde{Z}^{\leftarrow}_{(k+1)h}\| \leq \exp(S(1+R))\|x - y\|$ for $k = 1, \ldots, S/h$. Hence,

$$
\frac{\mathrm{d}(\eta_{X,\tau} - \eta_{Z,\tau})}{\mathrm{d}\tau}
$$
$$
= (\eta_{X,\tau} - \eta_{Z,\tau}) + 2\nabla_x^\top s(X^{\leftarrow}_{kh}, kh))\eta_{X,\tau} - 2\nabla_x^\top s(\tilde{Z}^{\leftarrow}_{kh}, kh))\eta_{Z,\tau}
$$
$$
= (\eta_{X,\tau} - \eta_{Z,\tau}) + 2(\nabla_x^\top s(X^{\leftarrow}_{kh}, kh) - \nabla_x^\top s(\tilde{Z}^{\leftarrow}_{kh}, kh))\eta_{X,\tau} - 2\nabla_x^\top s(\tilde{Z}^{\leftarrow}_{kh}, kh)(\eta_{Z,\tau} - \eta_{X,\tau}),
$$

which also yields that

$$
\frac{\mathrm{d}\|\eta_{X,\tau} - \eta_{Z,\tau}\|^2}{\mathrm{d}\tau}
$$
$$
= 2\|\eta_{X,\tau} - \eta_{Z,\tau}\|^2 + 4(\eta_{X,\tau} - \eta_{Z,\tau})^\top (\nabla_x^\top s(X^{\leftarrow}_{kh}, kh) - \nabla_x^\top s(\tilde{Z}^{\leftarrow}_{kh}, kh))\eta_{X,\tau}
$$
$$
\quad - 4(\eta_{X,\tau} - \eta_{Z,\tau})^\top \nabla_x^\top s(\tilde{Z}^{\leftarrow}_{kh}, kh)(\eta_{Z,\tau} - \eta_{X,\tau})
$$
$$
\leq 2\|\eta_{X,\tau} - \eta_{Z,\tau}\|^2 + 4\|\eta_{X,\tau} - \eta_{Z,\tau}\|H_s \exp(S(1+R))\varepsilon \exp(2(1+2R)S)
$$
$$
\quad + 4R\|\eta_{X,\tau} - \eta_{Z,\tau}\|^2.
$$

Therefore,

$$
\frac{\mathrm{d}\|\eta_{X,\tau} - \eta_{Z,\tau}\|}{\mathrm{d}\tau}
$$
$$
\leq (1+2R)\left[\|\eta_{X,\tau} - \eta_{Z,\tau}\| + 2H_s \exp(S(1+R)) \exp(2(1+2R)S)\varepsilon/(1+2R)\right],
$$

and thus by noticing $\|\eta_{X,0} - \eta_{Z,0}\| = 0$, we have

$$
\|\eta_{X,\tau} - \eta_{Z,\tau}\| \leq [\exp(S(1+2R)) - 1]\frac{2H_s \exp(S(1+R)) \exp(2(1+2R)S)}{1+2R}\varepsilon,
$$

for any $\tau \leq S$. Then, by setting $S = 1/(1+2R)$, the right hand side can be rewritten as

$$
\|\eta_{X,\tau} - \eta_{Z,\tau}\| \leq C_\eta \frac{H_s}{1+2R}\varepsilon,
$$

for an absolute constant $C_\eta$. Therefore, we arrive at

$$
(v^\top(\nabla\varphi_{Y,t}(x) - \nabla\varphi_{Y,t}(y)))^2
$$
$$
= \mathbb{E}\left[\frac{1}{S}\int_0^S \langle\eta_{X,\tau}, \mathrm{d}B_\tau\rangle\varphi_{Y,S}(X^{\leftarrow}_S) - \frac{1}{S}\int_0^S \langle\eta_{Z,\tau}, \mathrm{d}B_\tau\rangle\varphi_{Y,S}(\tilde{Z}^{\leftarrow}_S)\right]^2
$$
$$
\leq 2\mathbb{E}\left[\frac{1}{S^2}\int_0^S (\eta_{X,\tau} - \eta_{Z,\tau})^2\mathrm{d}\tau\right]\mathbb{E}\left[\varphi_{Y,S}(X^{\leftarrow}_S)^2\right]
$$
$$
\quad + 2\mathbb{E}\left[\frac{1}{S^2}\int_0^S \eta_{Z,\tau}^2\mathrm{d}\tau\right]\mathbb{E}\left[(\varphi_{Y,S}(X^{\leftarrow}_S) - \varphi_{Y,S}(\tilde{Z}^{\leftarrow}_S))^2\right]
$$
$$
\leq \frac{2}{S}C_\eta^2\frac{H_s^2}{(1+2R)^2}\varepsilon^2 \cdot C_\rho^2 + \frac{2}{S}\exp(4(1+2R)S)\|v\|^2 R_\varphi^2 \exp(2S(1+R))\varepsilon^2.
$$

Then, for the choice of $S = 1/(1+2R)$, the right hand side can be bounded by

$$
\left(\frac{2C_\eta^2 H_s^2 C_\rho^2}{1+2R} + 2(1+2R)\exp(6)R_\varphi^2\right)\varepsilon^2.
$$

This implies that $\nabla\varphi_{Y,t}(\cdot)$ is Lipschitz continuous with a constant $L_\varphi = \left(\frac{2C_\eta^2 H_s^2 C_\rho^2}{1+2R} + 2(1+2R)\exp(6)R_\varphi^2\right)^{1/2}$.

(ii) Next, we assume that $T - t \leq S = 1/4(1+2R)$. In this situation, we may use the following relation:

$$
v^\top\nabla\varphi_{Y,t}(x) = \mathbb{E}[\eta_\tau^{T-t}\nabla\rho_*(X^{\leftarrow}_T) \mid X^{\leftarrow}_t = x].
$$

And, tracing an analogous argument by replacing $\varphi_{Y,S}$ with $h$, we obtain the assertion with

$$R_\varphi = e^{1/2} R_\rho, \ L_\varphi = \left(2C_\eta^2 H_s^2 R^2 + e^2 L_\rho^2\right)^{1/2}.$$

$\square$

**Lemma 11.** *If $\|\rho_* - \rho\|_\infty \le \varepsilon'$, then*

$$\|\nabla_x \mathbb{E}[\rho_*(X_T^\leftarrow)|X_t^\leftarrow = x] - \nabla_x \mathbb{E}[\rho(X_T^\leftarrow)|X_t^\leftarrow = x]\| \le \frac{e}{\sqrt{\min\{T-t, 1/(2+2R)\}}} \varepsilon'.$$

*Proof.* It can be proved by the Bismut-Elworthy-Li formula again. We omit the details. $\square$

Combining all inequalities, we arrive at (the formal version of) Theorem 4.

**Theorem 8** (Formal statement of Theorem 4). *Assume that Assumptions 7 and 5 hold and the conditions in Lemma 10 are satisfied. and $\|\rho_* - \rho\|_\infty \le \varepsilon'$ and $\|\rho\|_\infty \le C_\rho$. Let $L_\varphi$ and $R_\varphi$ be as given in Lemma 10. Then, for $0 \le h \le \delta \le 1/(1 + 2R)$, we have that*

$$\mathbb{E}_{\bar{Y}_t^\leftarrow}[\|u_*(\bar{Y}_t^\leftarrow, t) - u(\bar{Y}_{k_t h}^\leftarrow, t)\|^2]$$
$$\lesssim C_\rho^3 \left\{ R_\varphi^2 \left(\varepsilon^2 + L_p^2 d(\delta + \mathsf{m}\delta^2)\right) + \Xi_{\delta,\varepsilon} + [(R_\varphi^2 + L_\varphi^2)(\mathsf{m} + 4Q^2 + dh) + R_\varphi^2(1+2R)^2]h^2 \right\}$$
$$+ \frac{e^2}{\min\{T-t, 1/(2+2R)\}} \varepsilon'^2 + C_\rho(1+2R)\sqrt{\frac{\log(T/(h\delta))}{nh}},$$

*where*

$$\Xi_{\delta,\varepsilon} := \frac{4c_\eta^2 C_\rho^2(1+2R)^2}{3}\delta + 2\exp(2)\left\{C_\rho^2\left(R_\varphi^2 + \frac{1}{\delta}\right)\varepsilon_{\mathrm{TV}}^2 + CR_\varphi^2[\varepsilon^2 + L_p^2 d(\delta + \mathsf{m}\delta^2)]\right\},$$

*and $c_\eta > 0$ is a universal constant.*

*Proof.* Define

$$\rho_{*,t}(x) = \mathbb{E}[\rho_*(\bar{X}_T^\leftarrow)|\bar{X}_t^\leftarrow = x], \ \rho_t(x) = \mathbb{E}[\rho(X_T^\leftarrow)|X_t^\leftarrow = x].$$

First, note that

$$\|u^*(x,t) - u(x,t)\|^2 = \left\|\frac{\nabla\rho_{*,t}(x) - \nabla\rho_t(x)}{\rho_{*,t}(x)} + \frac{\nabla\rho_t(x)(\rho_{*,t}(x) - \rho_t(x))}{\rho_{*,t}(x)\rho_t(x)}\right\|^2$$
$$\le 2\left\|\frac{\nabla\rho_{*,t}(x) - \nabla\rho_t(x)}{\rho_{*,t}(x)}\right\|^2 + 2\left\|\frac{\nabla\rho_t(x)(\rho_{*,t}(x) - \rho_t(x))}{\rho_{*,t}(x)\rho_t(x)}\right\|^2$$
$$\le 2C_\rho^2 \|\nabla\rho_{*,t}(x) - \nabla\rho_t(x)\|^2 + 2\frac{\|\nabla\rho_t(x)\|^2 |\rho_{*,t}(x) - \rho_t(x)|^2}{(\rho_{*,t}(x)\rho_t(x))^2}$$
$$\le 2C_\rho^2 \|\nabla\rho_{*,t}(x) - \nabla\rho_t(x)\|^2 + 2R_\varphi^2 C_\rho^2 |\rho_{*,t}(x) - \rho_t(x)|^2.$$

Therefore, the expectation of the right hand side with respect to $\bar{X}_t^\leftarrow$ can be bounded by

$$\mathbb{E}_{\bar{X}_t^\leftarrow}\left[\|u^*(\bar{X}_t^\leftarrow, t) - u(\bar{X}_t^\leftarrow, t)\|^2\right]$$
$$\le 2C_\rho^2 \mathbb{E}_{\bar{X}_t^\leftarrow}\left[\|\nabla\rho_{*,t}(x) - \nabla\rho_t(x)\|^2\right] + 2R_\varphi C_\rho^2 \mathbb{E}_{\bar{X}_t^\leftarrow}\left[|\rho_{*,t}(x) - \rho_t(x)|^2\right]$$
$$\le 2C_\rho^2 \mathbb{E}_{\bar{X}_t^\leftarrow}\left[\|\nabla\rho_{*,t}(x) - \nabla\rho_t(x)\|^2\right] + 2R_\varphi C_\rho^2 \mathbb{E}_{\bar{X}_t^\leftarrow}\left[\mathrm{TV}(\bar{X}_T^\leftarrow, X_T^\leftarrow|\bar{X}_t^\leftarrow = X_t^\leftarrow = x)|_{x = \bar{X}_t^\leftarrow}^2\right].$$

The first term of the right hand side can be bounded by Theorem 7 and Lemma 11. The second term can be bounded by $\varepsilon_{\mathrm{TV}}^2$ by Assumption 7.

In the same vein, we can bound the difference

$$\|u(x,t) - u(x, k_t h)\|^2$$
$$\le 2C_\rho^2 \mathbb{E}_{\bar{X}_t^\leftarrow}\left[\|\nabla\rho_t(x) - \nabla\rho_{k_t h}(x)\|^2\right] + 2R_\varphi C_\rho^2 \mathbb{E}_{\bar{X}_t^\leftarrow}\left[|\rho_t(x) - \rho_{k_t h}(x)|\right]$$
$$\le [2L_\varphi^2(\mathsf{m} + 4Q^2 + dh) + R_\varphi^2 c_\eta(1+2R)^2]h^2$$
$$\quad + 2R_\varphi^2 h^2(\mathsf{m} + 4Q^2 + dh)$$
$$\le [2(L_\varphi^2 + R_\varphi^2)(\mathsf{m} + 4Q^2 + dh) + R_\varphi^2 c_\eta(1+2R)^2]h^2,$$

where we used (45) and the same argument as (45) in the last inequality with $R_\varphi$ Lipschitz continuity of $\rho_t$.

Finally, we convert the expectation w.r.t. $\bar{X}_t^{\leftarrow}$ to that w.r.t. $\bar{X}_t^{\leftarrow}$. However, the density ratio between $p_t$ and $q_t$ is bounded by $C_\rho$, which yields the assertion. □

### C.4 Doob's H-transform

**Lemma 12.** *For all $t \in [0, T]$, the following relationship holds.*

$$\nabla_x \log q_t(x) = \nabla_x \log p_t(x) + \nabla_x \log(\mathbb{E}[\rho_*(\bar{X}_0)|\bar{X}_t = x]).$$

*Proof.* Let us denote the joint distribution of $\bar{X}_0$ and $\bar{X}_t$ as $p_{0,t}(\bar{X}_0, \bar{X}_t)$, the conditional distributions of $\bar{X}_0$ given $\bar{X}_t$ as $p_{0|t}(\bar{X}_0|\bar{X}_t)$, and the conditional distributions of $\bar{X}_t$ given $\bar{X}_0$ as $p_{t|0}(\bar{X}_t|\bar{X}_0)$. Define $q_{t|0}$ in the same way. It is straightforward to see that

$$\log p_t(x) + \log(\mathbb{E}[\rho_*(\bar{X}_0)|\bar{X}_t = x]) = \log(p_t(x)\mathbb{E}[\rho_*(\bar{X}_0)|\bar{X}_t = x])$$
$$= \log p_t(x) \int_{x'} \rho_*(x')p_{0|t}(x'|x)\mathrm{d}x'$$
$$= \log \int_{x'} \rho_*(x')p_{0,t}(x', x)\mathrm{d}x'$$
$$= \log \int_{x'} \rho_*(x')p_0(x')p_{t|0}(x|x')\mathrm{d}x'$$
$$= \log \int_{x'} q_0(x')p_{t|0}(x|x')\mathrm{d}x'. \tag{46}$$

Note that $p_{t|0}(x|x')$ and $q_{t|0}(x|x')$ are the same in (46). Therefore,

$$\log p_t(x) + \log(\mathbb{E}[\rho_*(\bar{X}_0)|\bar{X}_t = x]|) = \log \int_{x'} q_0(x')q_{t|0}(x|x')\mathrm{d}x'$$
$$= \log q_t(x),$$

which concludes the proof. □

## D Details of Numerical Experiments

In this Appendix, we compare our method with existing approaches in a very simple numerical experiment and confirm that our method is also capable of performing image generation alignment. A more practical and clearer comparison using realistic benchmarks is left for future work.

### D.1 Alignment for Gaussian Mixture Models

We explain how to align the pre-trained diffusion model for Gaussian Mixture Models and the technical details of our algorithm. Almost the same algorithm was used for the other experiments.

**A Pre-Trained Score-Based Diffusion Model** We pre-trained a score-based diffusion model to sample from the 2 dimensional mixture of Gaussian Mixture Models. The target density was $\frac{1}{2}(\mathcal{N}(\mu_1, \Sigma) + \mathcal{N}(\mu_2, \Sigma))$, $\mu_1 = [-2.5, 0]$, $\mu_2 = [2.5, 0]$, $\Sigma = [[1, 0], [0, 5]]$. The score model was implemented as simple 4 layer neural networks. The learning rate of pre-train was 0.0005, the batch size was 100, the number of epochs was 1000. The pre-train MSE loss is shown in the left side in Figure 4. The minimum losses until the current epoch were plotted. The histogram of 20000 samples from the pre-trained score-based diffusion model is the right figure in Figure 4. For sampling, $T$ was set to be 10 and the number of sampling steps was 100.

**The objective** The target was the mean of the Gaussian in the right side, $\mu_w := \mu_2 = [2.5, 0]$. The preference of point $x_w$ and $x_l$ were determined by the Euclidean distance $d(\cdot, \mu_w)$ from $\mu_w := [2.5, 0]$. $x_w \succ x_l$ if and only if $d(x_w, \mu_w) < d(x_l, \mu_w)$. The DPO objective was used in this setting. 2000 points from $p_{\mathrm{ref}}$ were sampled to calculate the expectation of the functional derivative. Note

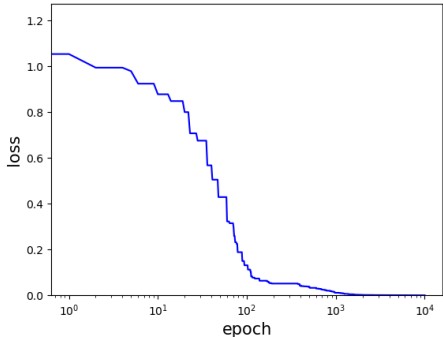 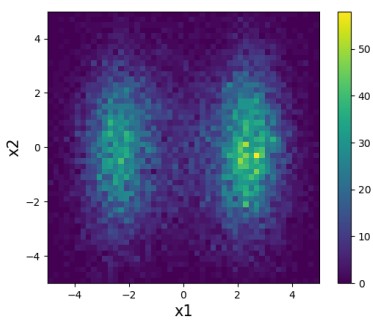

Figure 4: **Left.** Pre-train MSE loss of denoising score matching. The minimum losses until the current epoch were plotted. **Right.** The histogram of 20000 samples from the pre-trained DDPM.

that the reference objectives[2] can be (and were) analytically calculated because $q/p_{\text{ref}} = 1$. The regularization terms $\beta$ and $\gamma$ were 0.04 and 0.1. The metric loss was calculated with the samples s.t. $d(x, \mu_w) < 10$ in random 1000 samples.

As a counter method, Diffusion-DPO (Wallace et al., 2024) was implemented with the learning rate = 0.0005 and 0.0001 for w/ reg and w/o reg respectively, and batch size = 5000. We remark that, in practice, it is hardly realistic to compute the true DPO objective during Diffusion-DPO, but in this case, we forcefully carried out the computation by estimating the density ratio $q(x)/p_{\text{ref}}(x)$ with $100 \times 2$ samples for each $x$: The densities $q$ and $p_{\text{ref}}$ were estimated by repeatedly computing the denoising path and empirically obtaining their marginal densities, which implies some degree of numerical instability in Figure 2 and additional computational cost[3] in Table 1.

Please note that the reason why the values of "Upper bound"[4] and "True Objective" in the existing method shown in Figure 2 are reversed is that their "Upper bound" approximates the true DPO loss by replacing the current reverse process with the ROU process (Wallace et al., 2024). This implies that the upperbound may be too loose to be used as a surrogate of the true loss.

The phenomenon that the "True Objective w/o reg." of the existing method diverges upward in Figure 2 is likely due to the absence of additional entropy regularization, leading to catastrophic forgetting Uehara et al. (2024a); Tang (2024); Li et al. (2024b); Zhao et al. (2024). On the other hand, when applying our method, we incorporate entropy regularization.

Additionally, some degree of numerical instability is unavoidable due to the experimental methodology because we empirically estimated the density ratio to compute the true DPO loss in using Diffusion-DPO.

**Dual Averaging in Option 1** We implemented Option 1 for the experiment. We used 8 NVIDIA V100 GPUs with 32GB memory. The plotted losses (smoothed with EMA) in Figure 2 demonstrated that DA algorithm decreased the losses. Please refer to Figure 5 for losses without smoothing. We estimated the log-density ratio $\bar{g}^{(k-1)} = -\log \hat{q}^{(k)}/p_{\text{ref}} + \text{const.}$ with neural networks $f_k \simeq \bar{g}^{(k-1)}$. In each loop, the potential was trained by 1000 points in $x_1, x_2 \in [-5, 5]$, the learning rate was 0.0005, the number of epochs was 1000. We kept $\bar{g}^{(k-1)}$ as neural networks and generate the dataset

---

[2]the DPO loss of Ref. in Table 1, our algorithm at $k = 0$ before the random initialization of the potential, and the initial (true and approx.) objectives in Diffusion-DPO in Figure 2 and 5

[3]"Opt.time" and "GPU memory "also include the time and space required to compute the "True Objective"

[4]We took $\omega(\lambda_t)$ in (2) in Wallace et al. (2024) to be 0.5 (constant, as recommended in the paper). We will leave the investigation of the phenomena that arise when $\omega$ is turned into a time-dependent function for future work.

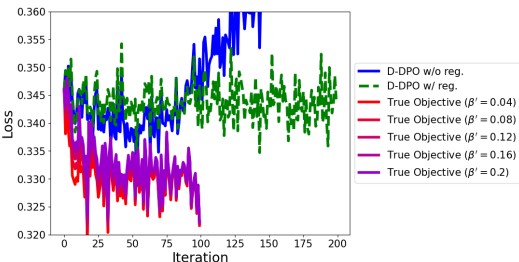

Figure 5: (For Reference) Losses without smoothing corresponding to Figure 2. "D-DPO": Diffusion-DPO. "Ours": Proposed DA.

for $\bar{g}^{(k)}$ from the equation ($\beta = \beta'$ for simplicity)

$$\bar{g}^{(k)} = \frac{2}{\beta(k+1)(k+2)} \left[ \frac{\beta k(k+1)}{2} \bar{g}^{(k-1)} + k \frac{\delta F}{\delta q}(q^{(k)}) \right]$$

$$= \frac{k}{k+2} \bar{g}^{(k-1)} + \frac{2k}{\beta k(k+2)} \frac{\delta F}{\delta q}(q^{(k)}). \tag{47}$$

So, we only need one model to store for DA loops (we don't need $\bar{g}^{(k-2)}, ..., \bar{g}^{(1)}$). The psudocode for this phase is described as Algorithm D.1.

---

**Algorithm D.1** Dual Averaging (Option 1)

---

**Input:** $F$: an objective, $s$: pre-trained score, $\beta$: Regularization scale, $K$: number of loops.
**Output:** $f_K$: a trained potential.
  Define $q^{(0)} = p_{\text{ref}}$ as the reference density obtained by the pre-trained score $s$
  Initialize NN $f_1$ randomly
  Collect samples $(x_i)_i$ (e.g. collect data at the final denoising step from $p_{\text{ref}}$ by score function $s$)
  **for** $k = 1, ..., K - 1$ **do**
    $q^{(k)} \propto \exp(-f_k)p_{\text{ref}}$ (no actual computation)
    Construct dataset $\{(x_i, \frac{2k}{\beta k(k+2)} \frac{\delta F}{\delta q}(q^{(k)}, x_i)\}_i$ with $(x_i)_i$ and $f_k$ using equation (2)
    Train $f_{k+1}$ to approximate $\frac{k}{k+2} f_k + \frac{2k}{\beta k(k+2)} \frac{\delta F}{\delta q}(q^{(k)})$ by minimizing MSE.
  **end for**
  **End**

---

The heat map of the trained potential $f_k$ is shown in Figure 6.

**Doob's h-transform** We sampled the aligned images with 50 diffusion steps. The guidance term of Doob's h-transform was calculated in every diffusion steps. The conditional expectation $\mathbb{E}[h_T(X_T^{\leftarrow}) \mid x]$ was calculated by Monte Carlo with 30000 samples. The psudocode for this phase is described as Algorithm D.2. In phase 2 (the sampling phase), our simplest solution (i.e., estimating the correction term at each time step using Monte Carlo) has a time complexity of $\mathcal{O}(L^2)$, where $L$ represents the number of time steps in the denoising process. When we set $N$ as the number of particles needed to estimate one correction term, to compute the correction term for each sample simultaneously, $\mathcal{O}(N)$ memory space is required. The sampling error for each correction term would be $\mathcal{O}(1/\sqrt{N})$. This leads to severer additional computational effort in phase 2 than the time and space consumed as in Table 1 for Phase I[5]. However, this Doob's h-transform technique itself has been used in image generation (Uehara et al., 2024a;b), Bayesian samping (Heng et al., 2024), and filtering (Chopin et al., 2023). As a more practical alternative of our phase 2, the idea of approximating the correction term using neural ODE solvers for faster test-time implementation has also been proposed in (Uehara et al., 2024b;a).

---

[5]In Table 1, we showed the computational cost for Phase I, not for Phase II because we only require Phase I to compute the true loss. Note that we used Phase II to compute the metric loss, however, the true loss is much more important.

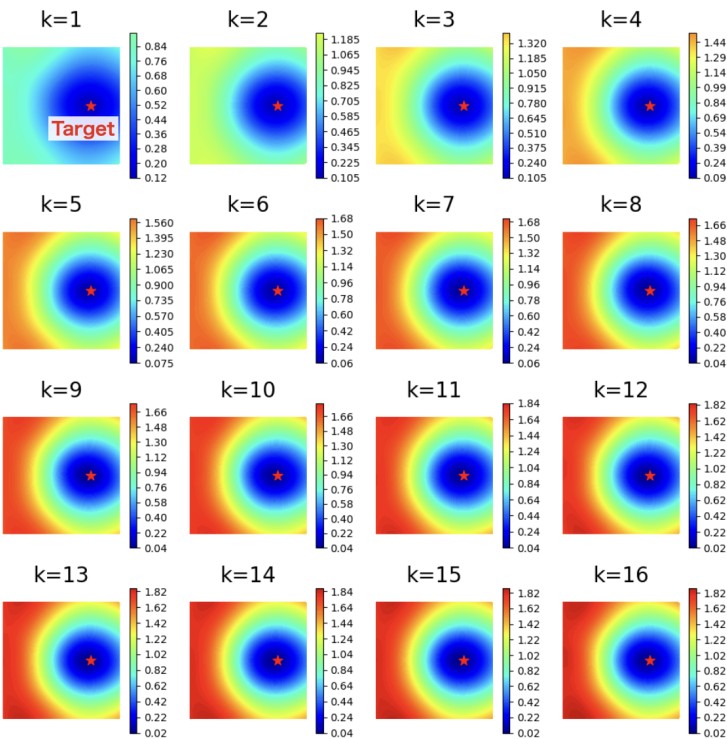

Figure 6: The heatmap of the potential $f_k$ in the $k$th loop in Dual Averaging for Gaussian Mixture Model. The target point was $[2.5, 0]$. Note that $f_k$ is the negated log-density ratio: the aligned density is $\exp(-f_k)p_{\text{ref}}$.

---

**Algorithm D.2** Doob's h-transform (A simplest implementation)

---

**Input:** $F$: an objective, $f_K$: a trained potential, $s$: pre-trained score,
**Output:** $x_T$: an output approximately from $\exp(-f_K)p_{\text{ref}}$.
  Initialize $x_0$ as white noise
  Set number of steps $L$ and the time $T$.
  Set the step size $h = T/L$
  **for** $l = 0, ..., L-1$ **do**
    Initialize $N$ samples $(x_{l,i})_{i \leq N}$ as $x_l$.
    **for** $l' = l, ..., L-1$ **do**
      (denoising step of $(x_{l,i})_{i \leq N}$)
    **end for**
    Store $N$ samples of $X_T^{\leftarrow}$ as $(x_{l,i})_{i \leq N}$.
    Approximate $u(x_l, lh) = \nabla \log \overline{\mathbb{E}}[\exp(-f_K(X_T^{\leftarrow}) \mid X_{lh}^{\leftarrow} = x_l]$ by Monte Carlo with $(x_{l,i})_{i \leq N}$.
    Sample white noise $\xi_{\text{noise}}$
    $x_{l+1} := x_l + \delta(x_l + 2s(x_l, T - lh) + 2u(x_l, lh)) + \sqrt{2h}\xi_{\text{noise}}$
  **end for**
  **End**

---

## D.2    IMAGE GENERATION ALIGNMENT

We aligned the image generation of the basic pre-trained model in Diffusion Models Course (source: HuggingFace (2022)). The pipeline path we utilized was "`johnowhitaker/ddpm-butterflies-32px`". The summarized results are show in Figure 8.

**The pre-trained Model** The model samples the images of butterflies of $32 \times 32$ pixels. Number of the sampling step was 1000.

**The objective** The target color was $[0.9, 0.9, 0.9]$ in RGB. The reward is visualized in Figure 7. 6400 samples from $p_{\text{ref}}$ to calculate the expectation of $F$ and $\frac{\delta F}{\delta q}$. $\beta$ and $\gamma$ were 0.05 and 1. The output of the functional derivative was clipped in $\pm 20$ to stabilize the training step. In calculation of the DPO objective, the sample that has a higher reward is the "winning" sample.

**Dual Averaging** In each loop, $f_k$ was trained by 1024 images from pooled 6400 images, the learning rate was 0.0001, the batch size was 64, and the number of epochs was 5. $f_k$s were implemented by Unet2Dmodel in Diffusers library von Platen et al. (2022). How the potential $f_k$ learned the reward, the distance from the target color, was shown in the right side of Figure 7. The DA algorithm succeeded to extract the target images from the true reward.

Figure 7: **Left.** Output images of $p_{\text{ref}}$ sorted by the distance from the target color ($= -$reward). **Right.** Images sorted by the learned potential in $k = 2$.

**Doob's h-transform** We sampled the aligned images with 1000 diffusion steps. The guidance term of Doob's h-transform was calculated in every 10 diffusion steps for faster sampling. We calculated the drift term of Doob's h-transform once in 10 diffusion steps for faster sampling. In addition, We defined a decay rate $r_d = 0.95$ and a strongness $s = 5$ for $\nabla \log \rho_t$ and finally we added $r_d^{m'} \nabla \log \mathbb{E}[\exp(-s f_K(X_T^{\leftarrow})) \mid x_{10m\delta}]$ as a drift term in $l = 10m + m'$th diffusion step to balance the computational cost and stability. The conditional expectation $\mathbb{E}[\rho_T(X_T^{\leftarrow}) \mid x]$ was calculated by Monte Carlo with 128 samples.

## D.3    TILT CORRECTION FOR GENERATION OF MEDICAL IMAGE DATA

Our goal of this experiment was generating images with no rotation with an unconditional pre-trained model that generates rotated images. The objective was based on DPO. The rotation angle of each image was predicted with CNN. The summarized results are in Figure 12.

**Dataset** 10000 images of Head CT ($64 \times 64$ pixel) in Medical MNIST Lozano (2017) were leveraged. They include some rotated images, the angles are up to $90°$. To aggravate the existing situation, we augmented the data by rotating images (up to $45°$) from this dataset to be 40000. A predictor of angles were trained with the augmented dataset to define the reward in place of human preference due to preparation difficulties. The number of epochs was 10, 95% and 5% of the dataset were used for training and validation, the training MSE loss was 3.79, and the validation MSE loss was 3.25.

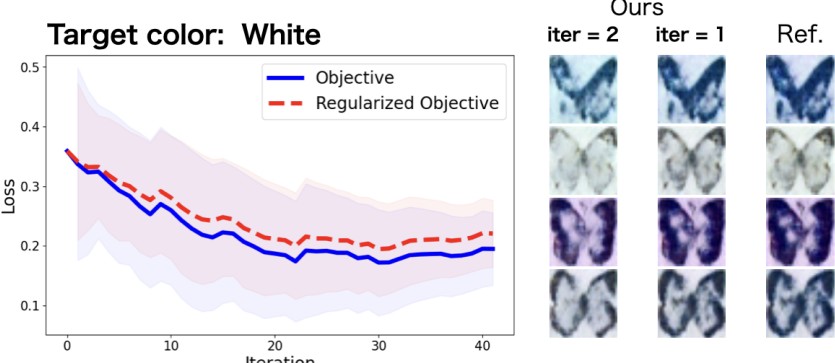

Figure 8: **Left.** The smoothed loss during DA for image generation alignment from $k = 1$. "Objective": DPO objective. "Regularized Objective": "Objective" $+ \beta D_{\mathrm{KL}}(q \| p_{\mathrm{ref}})$, $\beta = 0.05$ **Right.** Examples of aligned image generation. "iter=2": ours with $k = 2$ DA iterations, "iter=2": ours with $k = 1$ DA iteration. "Reference": samples from $p_{\mathrm{ref}}$.

Note that there were rotated images in the original dataset, so the labels of the rotation angles made in the augmentation were noisy.

**Pre-trained Autoencoder** We pre-trained autoencoder from scratch. It encodes gray-scaled $64 \times 64$ pixels into latent $32 \times 32$ pixels. It only has convolution layers so that geometrical features were preserved for simplicity. The training data was $95\%$ of augmented 40000 images and the validation data was $5\%$ of them. This autoencoder was fed white-noised data to make the model robust with noise. The pretraining MSE loss is shown in the left side of Figure 9.

**Latent Diffusion Model** We also pre-trained a (latent) diffusion model based on Unet2DModel in Diffusers von Platen et al. (2022). Number of sampling steps was 1000. The beta scheduler was set to be squaredcos_cap_v2. In pre-training, we leveraged the augmented dataset up to 20000 images, the number of epochs was 15, the batch size was 8, the learning rate was 0.0001. The pretraining MSE loss is shown in right side of Figure 9 and the generated images are displayed in Figure 10.

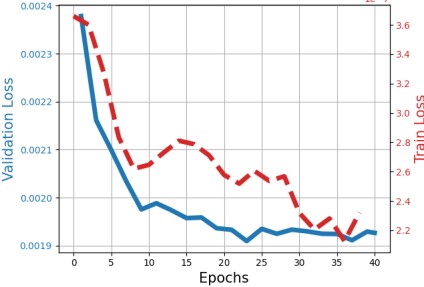 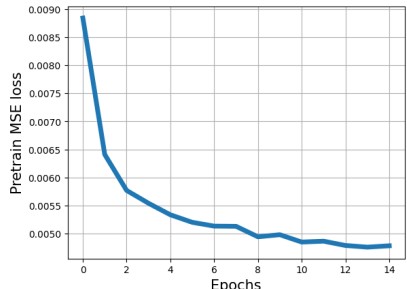

Figure 9: **Left.** The MSE loss in pretraining the Autoencoder. The solid blue line and the dashed red represent the validation loss and the train loss. **Right.** The MSE loss in pretraining the diffusion model.

**The objective** The reward was defined to be $-|\text{predicted angle}|$ of the pre-trained predictor (in place of humans), described in the left side of Figure 11. 6400 samples from $p_{\mathrm{ref}}$ to calculate the expectation of $F$ and $\frac{\delta F}{\delta q}$. $\beta$ and $\gamma$ were 0.01 and 0.1. The output of the functional derivative was clipped in $\pm 5$ to stabilize the training step. In calculation of the DPO objective, the sample that has a higher reward (in the more vertical direction) is the "winning" sample.

**Dual Averaging** In each loop, $f_k$ was trained by 6400 images from $p_{\mathrm{ref}}$. All the generated images were reused during DA. The learning rate was 0.0001, the batch size was 64, and the number of

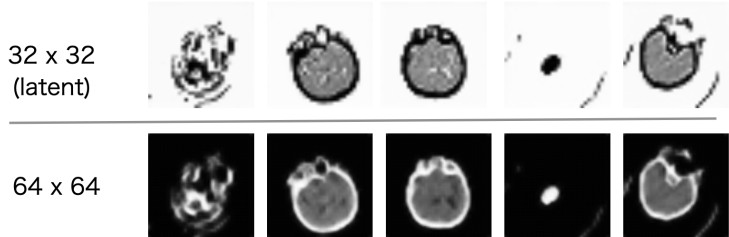

Figure 10: **Top.** Outputs of pre-trained latent diffusion. **Bottom.** Decoded images of the outputs.

epochs was 5. We compared the learned potential with the reference reward in Figure 11. We see that DA iterations worked well to replicate the reference reward.

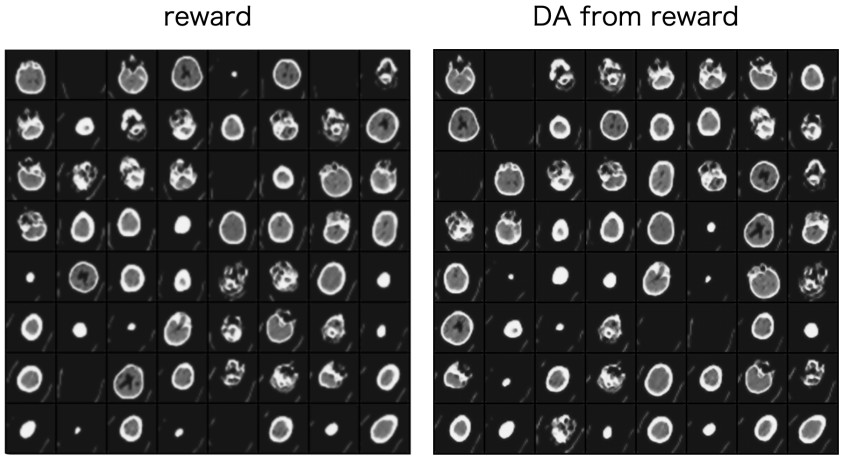

Figure 11: **Left.** Output CT images of the pre-trained model, sorted by the absolute values of estimated angles ($= -$reward). **Right.** Output CT Images sorted by the trained potential in the 3rd loop in DA.

**Doob's h-transform** We sampled the aligned images with 1000 diffusion steps. The guidance term of Doob's h-transform was calculated once in 10 diffusion steps. Technical settings were the same with . The conditional expectation was calculated by Monte Carlo with 128 samples.

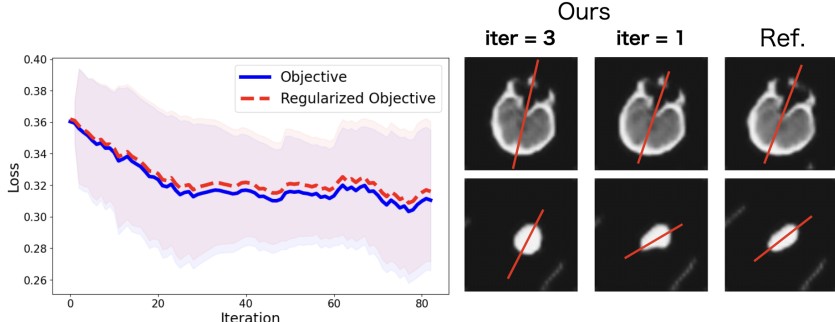

Figure 12: **Left.** The smoothed loss during DA for tilt correction from $k = 1$. "Objective": DPO objective. The target point was $[2.5, 0]$. "Regularized Objective": "Objective" $+ \beta D_{\mathrm{KL}}(q\|p_{\mathrm{ref}})$, $\beta = 0.01$ **Right.** Tilt-corrected Head CT image generation. "iter=3": ours with $k = 3$ DA iterations, "iter=1": ours with $k = 1$ DA iteration. "Reference": samples from $p_{\mathrm{ref}}$.

