# OpenReview forum: "Direct Distributional Optimization for Provable Alignment of Diffusion Models"
_ICLR.cc/2025/Conference — ICLR 2025 Poster_

### Official Review · Reviewer_tyit · 2024-10-28

**Soundness:** 3
**Presentation:** 2
**Contribution:** 3
**Rating:** 6
**Confidence:** 4

**Summary:**

This paper proposes a new optimization procedure to minimize entropy-regularized versions of finetuning losses for diffusion models such as DPO or KTO losses. This optimization method is an approximation of Dual Averaging (DA), where, instead of tracking a set of particles to approximate the distribution as done in prior work, an estimate of the unnormalized density ratio between the iterate and the original reference model is tracked. These changes make it possible to sample from finetuned model (e.g. the final iterate) by simulating the backward SDE of a OU forward process (à la diffusion model), where the score of the forward process is estimated by a Monte Carlo approximation its expression obtained using Doob's h-transform formula, whose ``correction'' terms can be estimated from the density ratio-estimate.

On the theoretical side, global convergence (in TV) of the final DA iterate to the true minimizer is provided when the loss is Convex, its first variation is bounded and smooth, and the density ratio estimates are accurate enough, while non global convergence (convergence of the gradient) to some limit is established if the loss in nonconvex. Finally, the TV between the distribution of the final finetuned model samples and true finetuning loss minimizer is controlled, assuming a bound on the error of score of the reference OU process, and a bound in the error of the Doob correction term.

The method is implemented on a synthetic 2d mixture of Gaussians reference model with mode alignment, an image dataset with color alignment, and Head CT scans from Medial MNIST with orientation alignment, where it is show that the loss trends downwards, and that the finetuned samples are more aligned than the ones from the reference models.=

**Strengths:**

The present submission tackles an important problem (diffusion model alignment). The method is an interesting twist on existing dual averaging approaches, whose setting does not involve estimating density ratios.

A significant effort is spent providing end-to-end guarantees for both the optimization procedure and the sampling procedure. These guarantees do not directly require a Log-Sobolev inequality on the target.

Finally, on the experimental side, the method is tested on high-dimensional dataset, and display reasonable performance (although there is no relative point of comparison due to the lack of results reporting the performance of existing methods on the datasets and tasks explored by the paper).

**Weaknesses:**

My main concern is the lack of comparison against existing methods to finetune diffusion models, both in terms of performance and in terms of space and memory complexity. I personally like the paper, however I believe it needs comparison against the applicable baselines before acceptance.
An investigation of the impact of $\beta$ would also be interesting.

The method involves some computationally expensive operations such as Monte Carlo estimation at every iteration of the DA loop. It would be nice to have the total time and space complexity of the algorithm described somwehere, complementing a Computational time comparison with other methods. The fact that only one previous density ratio estimate (and not k) are needed to approximate the next one (see l 1953, algorithm D1) seems important to cap the total time and space complexity of the method and not directly obvious from equation 2 and 3, should be mentioned in the main.

The presentation needs to be clarified. Currently,
  - How is the reader supposed to know that the equality on equation (2) holds? It seems hidden as specific instances of Lemma 1, but this should be highlghted more.
  - The recipe for estimating the density ratios during DA is not clearly described in the main body.
  - The presentation of the image generation results could be improved: what is the color (0.9/0.9/0.9)? this deserves to be on one of the figures IMO. Figure 7 definitely helps to understand how much the model has learned, and should be at least referenced in the main text.
  - I did not understand the meaning of the ("Linear/nonlinear", "k=1/2/3 DA Loops") terms. Do you mean number of DA iterations? Why linear and nonlinear?
  - The reason "our method works without isoperimetry conditions such as Log-Sobolev inequality" could be explained more clearly: my understanding is that the original DA update for MLFD required running an inner Langevin sampler, which is sensitive to LSI. On the other hand, the current method estimates the density ratio by using the backward SDE to sample from the reference model, which does not rely on an LSI to converge.

Additional minor style/typos improvements are suggested in minor.




### **Minor**
The style could be substantially improved. The english grammar mistakes and imprecision should be corrected.

- the differences between the two DA techniques are not explained. Turns out some results hold for Opt 1 or Opt 2.
- Typo l 1235, 1236, 1249: $\nabla p_t$ -> $\nabla \log p_t$
- "Particle-based optimization methods are not designed for resampling from the distribution where obtained particles follow" -> "...that the obtained particle follow"
- "pre-trained distribution is typically highly complex multimodal distributions, failing to satisfy isoperimetric conditions". Not super accurate. Isoperimetric conditions are verified for Mixture of Gaussians for instance. But the isoperimetric constant is very large.
- "The convergence rate of MFLD (Nitanda et al., 2022; Chizat & Bach, 2018) has been established under the condition where the proximal"... I don't think (Chizat & Bach, 2018) tackle MFLD.
- Consider using ``test'' for correct quotes formatting.

**Questions:**

.

---

> ### Author Response · Authors · 2024-11-20
> **Official Comment (1/2)**
>
> We thank the reviewer for the thoughtful comment and constructive feedback. We address the technical concerns below.
>
> ---
>
> **Comparison against existing methods**
>
> We absolutely agree that we need to compare our methods against existing methods, so we have done experiments with an existing DPO optimization method, called Diffusion-DPO [R1].
>
> - We empirically compared the performance of the methods with Gaussian Mixture Model, setting the same DPO objective.
> - The Dual Averaging was done with $\beta'\geq\beta=0.04$. We have illustrated results for the case where $\beta'>\beta$ in the revised manuscript.
> - As a counter method, following Diffusion-DPO [R1], we have optimized the score networks by minimizing the approximate upperbound, which can be calculated with the score, of the (not regularized / $\beta$-regularized) true DPO loss. In practice, it is hardly realistic to compute the true loss during Diffusion-DPO, but in this case, we forcefully carried out the computation. The densities $q$ and $p_\mathrm{ref}$ were estimated by repeatedly computing the denoising path and empirically obtaining their marginal densities.
> - We evaluated how the true DPO loss of Diffusion-DPO behaves while minimizing the approximated upperbound.
> - In addition, we evaluated the averaged squared Euclidean distance from the target mean of Gaussian as the Metric Loss.
>
> We have compared the performance and the computational costs in the optimization phase:
>
> |                       | Ref.   | Diffusion-DPO (50 iter., w/o reg.) | Diffusion-DPO (200 iter., w/ reg.)           | Ours ($\beta$=0.04)   |
> | --------------------- | ------ | ---------------------------------- | -------------------------------------------- | --------------------- |
> | True DPO Loss         | 0.346  | 0.340             | 0.343                        | 0.328          |
> | (Approx.) Upperbound  | -      | 0.342             | 0.337                        | -              |
> | Metric Loss           | 3.226  | 2.828             | 3.283                        | 2.098          |
> | Opt. Time (s)         | -      | 509.99            | 1457.00                      | 1166.00        |
> | GPU memory (\%)       | -      | 6.54              | 6.54                         | 8.71           |
>
> From this result, we observe the following:
>
> - We got a smaller value of the true DPO loss and the metric loss for our method.
> - It is crucial to note that the prior approach makes it difficult to control the true loss. That is, minimizing their approximated upperbound failed to decrease the true loss indicating that the upperbound is too lose to be used as a surrogate of the true loss (please refer to the loss curves in the revised manuscript).
> - In our approach, we directly optimize the (regularized) true loss, ensuring that the iteration can continue until it reaches a small value albeit at a higher computational cost.
> In phase 1 (the optimization phase), regarding time and space complexity, our method was comparable to the existing method to achieve roughly the same loss. As you mentioned, while we only need to store one previous density ratio model, our method does require more space to be utilized.
>
> In phase 2 (the sampling phase), our simplest solution (i.e., estimating the correction term at each time step using Monte Carlo) has a time complexity of $\mathcal{O}(L^2)$, where $L$ represents the number of time steps in the denoising process. When we set $N$ as the number of particles needed to estimate one correction term, to compute the correction term for each sample simultaneously, $ \mathcal{O}(N)$ memory space is required. The sampling error for each correction term would be $\mathcal{O}(1/\sqrt{N})$. This leads to severe computational effort in phase 2.
>
> However, we emphasize that Doob's $h$-transform techniques used in our phase 2 have been empirically applied in image generation tasks [R2], Bayesian computation [R3], and online filtering [R4]  with efficient sampling schemes. In particular, as a more practical alternative of our phase 2, the idea of approximating the correction term using neural ODE solvers for faster test-time implementation has also been proposed in [R2]. This would enhance the practicality of our phase 2.

---

> ### Author Response · Authors · 2024-11-20
> **Official Comment (2/2)**
>
> **Clarification of the presentation**
>
> We appreciate your feedback on the presentation. We have made the modifications as suggested:
>
> - We have clarified the differences between Opt.1 and Opt.2 in the theoretical analysis of Dual Averaging. In particular, when the objective $F$ is convex, we have $\mathcal{O}(1/K)$ convergence of the weighted sum of the losses in Opt.1 with $K$ iterations.
> On the other hand, when $F$ is not necessarily convex (but smooth), the functional derivative of the regularized loss $\delta L/ \delta q (\hat{q}^{(K)})$ converges to zero, up to a constant w.r.t $x$, in Opt.2.
>
> - We have also updated the figures and the implementation details from the previously conducted experiments to make them more comprehensible. Just to clarify, the terms *linear* and *nonlinear* refer to the fact that when the target functional is linear, a single iteration of Dual Averaging is sufficient, whereas in the nonlinear case, multiple iterations are required.
>
> - We have refined the expressions related to the LSI to make them more precise.
>
> ---
>
> If you feel that all of your concerns have been properly addressed, we would deeply appreciate your consideration of a higher rating score, if you believe it is warranted.
>
> ---
>
> [R1] Bram Wallace, Meihua Dang, Rafael Rafailov, Linqi Zhou, Aaron Lou, Senthil Purushwalkam, Stefano Ermon, Caiming Xiong, Shafiq Joty, Nikhil Naik (2024). Diffusion model alignment using direct preference optimization. In Proceedings of the IEEE/CVF Conference on Computer Vision and Pattern Recognition (pp. 8228-8238).
>
> [R2] Masatoshi Uehara, Yulai Zhao, Kevin Black, Ehsan Hajiramezanali, Gabriele Scalia, Nathaniel Lee Diamant, Alex M Tseng, Tommaso Biancalani, Sergey Levine: Fine-Tuning of Continuous-Time Diffusion Models as Entropy-Regularized Control. arXiv, 2024.
>
> [R3] Jeremy Heng and Valentin De Bortoli and Arnaud Doucet: Diffusion Schrödinger Bridges for Bayesian Computation. Statistical Science, 2024.
>
> [R4] Nicolas Chopin, Andras Fulop, Jeremy Heng, Alexandre H. Thiery: Computational Doob $h$transforms for Online Filtering of Discretely Observed Diffusions. ICML, 2023.

---

> ### Comment · Area_Chair_4jV8 · 2024-11-26
> **Please reply to authors**
>
> Thank you very much for your work on reviewing this submission.
>
> Could you read the author's reply, check if they have addressed your comments, and acknowledge it?
>
> Further, please make sure, if necessary, to explain what your new stance is based on this information.

---

> > ### Comment · Reviewer_tyit · 2024-11-28
> > **Reply**
> >
> > I thank the authors for their efforts in comparing to existing alternatives. I have some questions:
> >
> > - In figure 2 of the updated manuscript. The wording makes it seem like the "upper bounds" are upper bound of the "True objective" (I briefly checked the paper and it seems like so). However, the lines "objectives" are above the lines "upper bound". Can the authors elaborate on that?
> > - Can the authors provide a few more sentences on the landscape of alternative, and of why they picked Diffusion DPO as an alternative?
> >
> > Minor:
> > - The Chizat & Bach, 2018 reference p. 181 has not been removed by the authors - I think it should be (they do not study MFLD)

---

> ### Author Response · Authors · 2024-12-01
> **Reply to Reviewer tyit**
>
> Thank you for replying our rebuttal. Below, we address your questions in detail:
>
> Numerical Experiments
>
> - We note that “Upperbound” refers to the upper bound on the approximation to the DPO loss, not on the true DPO loss. We refer you to paper [R1], which approximates the true DPO loss by replacing the current reverse process with the ROU process and derives the upper bound based on this approximation. Hence, the upper bound could lie below the true DPO loss since the aligned reverse process will deviate from ROU.
>
> - In fact, we empirically observed that the upper bound successfully constrained the true objective during the early stages (up to approximately 25 steps). However, the true loss began to increase and diverge without additional regularization, implying "catastorophic forgetting" or "overoptimization" to the upper bound, and the true loss ceased to decrease with additional regularization.
>
> - As far as we know, Diffusion-DPO [R1] is the first paper that applied Direct Preference Optimization to diffusion models. Diffusion-DPO was presented as a computationally efficient alternative of Reinforcement Learning approach. This method is empirically powerful: for example, utilizing Diffusion-DPO, [R2] aligned diffusion models for text-to-audio generations. However, their method approximates the true DPO loss by replacing the current (aligned) reverse process with the ROU process and derives the upper bound based on this approximation, which can be crude. Therefore, we selected Diffusion-DPO as an alternative because this issue has not been thoroughly studied in prior work.
>
> Citation
>
> - As you have suggested, we will replace the citation in p.181 to [R3] in the final version.
>
> We will clarify and fix these points you have mentioned in the final version.
>
> ---
>
> [R1] Bram Wallace et al. (2024). Diffusion Model Alignment Using Direct Preference Optimization. ArXiv preprint. https://arxiv.org/abs/2311.12908
>
> [R2] Navonil Majumder, Chia-Yu Hung, Deepanway Ghosal, Wei-Ning Hsu, Rada Mihalcea, and Soujanya Poria. 2024. Tango 2: Aligning Diffusion-based Text-to-Audio Generations through Direct Preference Optimization. In Proceedings of the 32nd ACM International Conference on Multimedia (MM '24). Association for Computing Machinery, New York, NY, USA, 564–572. https://doi.org/10.1145/3664647.3681688
>
> [R3] Chizat, L. (2022). Mean-field Langevin dynamics: Exponential convergence and annealing. *Transactions on Machine Learning Research*. https://openreview.net/forum?id=BDqzLH1gEm

---

> > ### Comment · Reviewer_tyit · 2024-12-03
> > **Response**
> >
> > I thank the authors for the clarifications. I encourage the authors to make these facts clear in the paper (especially regarding the different objectives/upper bounds of the DPO loss). I also think that briefly restating the advantages of DPO compared to RLHF (if DPO is indeed a computationally efficient alternative of Reinforcement Learning approach") is important as the experiments focus on DPO. Finally, I'd encourage the authors to clearly state (in the main text) the computational bottleneck of the method, and to suggest ways to address this, as done by the reviewer in their reply to my comments.
> >
> > Overall, the experiments ran by the authors cleared some of my concerns regarding comparison to prior work. I have raised my score.

---

### Official Review · Reviewer_ei8r · 2024-11-02

**Soundness:** 3
**Presentation:** 3
**Contribution:** 4
**Rating:** 8
**Confidence:** 4

**Summary:**

From my understanding, the paper presents a novel distribution alignment technique for score-based diffusion models based on optimization techniques over probability measures. Standard measure optimization techniques face challenges in the diffusion model setting because both the density governed by the model and the density of the true reference distribution cannot be evaluated. The model parametrizes its distribution using the score function/networks, whereas the true reference distribution is accessible only through samples. The work reformulates the measure optimization as an optimization problem over potentials. Then, sampling using the learned potential proceeds by the Doob's $h$-transform technique.

**Strengths:**

- The proposed method is usable on relevant examples such as reward-based alignment, DPO, and KTO. There are contributions even in the small details of this derivation, including rewriting the DPO objective in a way that is more amenable to statistical analysis.
- The sampling error is bounded while taking into account all sources of approximation: the discretization of the SDE, approximation of the score, and approximation of the Gibbs distribution resulting from the potential.
- While the theoretical derivations in the paper are already a strong contribution, the experiments are helpful in confirming that the method works on simple tasks.

**Weaknesses:**

Broadly speaking, the significant portions of the paper are unreadable. While I applaud the authors effort in producing this work, I request an equal effort in presenting it well. Some specific points are outlined below.
- The "two main challenges" addressed by paper are stated at least three times (045-048, 162-168, 185-194). While I believe this was done for clarity sake, the "challenge" is never described in an accessible way. While I interpret them as what I wrote in the summary, it is not explicitly clear to me. To say "particle-based optimization methods are not designed for resampling from the distribution where obtained particles follow" is meaningless without having described what particle-based methods are. The pre-trained distribution "failing to satisfy isoperimetric conditions" is meaningless without clearly explaining the implication of this (for example, are there no distributional optimization guarantees for distributions that do not satisfy isoperimetric conditions)?
- The appendix is a literally a list of theorems and proofs. Whether for optimization or sampling-style analysis, it is important to give context to all results. Concretely, this means: 1) providing a proof outline, 2) describing which technical steps are identical to/adapted from previous work (with references), and what proof techniques are specific to this paper. In its current state, it is hard to even verify the technical correctness of the paper.
- The notation is excessive, and not always introduced. Unless I am misunderstanding, $\hat{q}$ is presented as the minimizer of the objective, but seems to be changed to $q^*$ towards the end. I am also not sure that $\tilde{X}_{kh}^{\leftarrow}$ is introduced in 416. In the final version, please provide a global notation table in the first appendix containing all symbols and a verbal description.

I am recommending 5 as a score.

I will **raise my score from 5 to 6** if the authors can provide in their rebuttal **a completely rewritten version of 036 - 048 in an accessible form**, including equations, for readers who are unfamiliar with optimization over probability measures, as this is a more niche field than traditional optimization theory or diffusion models at this point (feel free to ignore references).

I will **raise my score from 6 to 8** if the authors provide **all missing text** from Appendix A. This includes
- text between A and A.1
- text between A.1 and Theorem 5 (which should be renamed to Theorem 1).
- description of the significance of Lemma 1 (which seems to be a strong convexity-like result).
- description of the significance of Lemma 2
- any in-line explanations.
- similar for all of A.2.
Please look to the outline in the weaknesses for this. For examples of papers with well-presented optimization/sampling proofs, see [Schmidt et al. (2016)](https://arxiv.org/pdf/1309.2388), [Diakonikolas et al. (2021)](https://arxiv.org/pdf/2011.00364), [Balasubramanian et al. (2022)](https://proceedings.mlr.press/v178/balasubramanian22a/balasubramanian22a.pdf).

**Questions:**

- In Assumption 2, are both (i) and (iii) necessary? Can we not assume (iii), the derive (i) with TV^2 instead of KL, and then apply Pinkser's inequality to achieve the smoothness relative to KL?
- The nonconvex convergence analysis results only in convergence of the successive gaps of iterates. What would it take to achieve convergence to an appropriately-defined stationary point?

---

> ### Author Response · Authors · 2024-11-20
> **Official Comment (1/2)**
>
> We thank the reviewer for the thoughtful comment and constructive feedback. We address the technical concerns below.
>
> ---
>
> We are grateful to the reviewer ei8r for highly valuing our contributions and for suggesting improvements to the readability of the paper.
>
>
> Following your suggestions, we have revised the manuscript:
>
> - We have revised the introduction to make it more accessible to readers who are not familiar with distribution optimization.
> - We have also revised the convergence analysis of Dual Averaging, including a proof sketch, to ensure it is fully understandable.
>
> We would appreciate it if you could review the revised manuscript.
>
> Below, we will provide responses to the valuable questions we received:
>
> **Challenges in our paper**
>
> Alignment of diffusion models as a minimization problem over the probability space has two challenges:
>
> - (i) inaccessibility of the output densities.
>
> As you summarized in the summary, the issue lies in the fact that the output distribution of diffusion models is parametrized by the score network.
>
> - (ii) muitimodality of the densities.
>
> Existing distributional optimization methods such as mean-field Langevin dynamics [Mei et al., 2018] and particle dual averaging [Nitanda et al., 2021] resolved this minimization problem by adapting a Langevin type sampling procedure into its optimization procedure so that we can calculate a functional derivative of the objective. Concretely, we find the functional derivative $\delta F/ \delta q$ in mean-field Langevin dynamics that is described as
> $$
> \mathrm{d}X_t = - \nabla \frac{\delta F}{\delta q}(q_t,X_t)\mathrm{d}t + \sqrt{2\lambda}\mathrm{d}B_t,
> $$
> where $X_t \sim q_t$ and $(B_t)_t$ is Brownian motion. This density $q_t$ approaches the minimizer of the optimization problem.
>
> However, the densities in our problem setting $q$ and $q_\mathrm{ref}$ are highly complex and multimodal from which it is extremely hard to generate data by a standard MCMC type methods including the Langevin dynamics.
> This difficulty can also be mathematically characterized by isoperimetric conditioins, such as logarithmic Sobolev inequality (LSI), that has usually exponential dependency on the data dimension $d$ for multimodal data yielding the curse of dimensionality.
> Unfortunately, the existing distribution optimization methods mentioned above are sensitive to the LSI constant, so they suffer from severely slow convergence, failing to align diffusion models.
>
> **Necessity of assumptions for Dual Averaging**
>
> The first assumption (i) is quite standard assumption known as a Lipschitz smoothness condition.
> This is required to obtain $O(1/k)$ convergence, otherwise, we would only have $O(1/\sqrt{k})$ for convex settings and no guarantee for non-convex settings.
> The condition (ii) is required to take the approximation error of $q^{(k)}$ from $\hat{q}^{(k)}$ into account. If we have the exact solution (i.e., $q^{(k)} = \hat{q}^{(k)}$), then this is not required.

---

> ### Author Response · Authors · 2024-11-20
> **Official Comment (2/2)**
>
> **Is it possible to deduce (i) from (iii)?**
>
> Thank you very much for your insightful comment.
> Yes, that is true. We can derive the condition (i) from the third one (iii).
> Acutally, $q,q'\in\mathcal{P}$, we define $q_t = q' + t (q-q')$ that is mixture of $q$ and $q'$. Then,
>
> $F(q) - F(q') - \int \frac{\delta F}{\delta q}(q')\mathrm{d}(q-q')
>     = \int_{t=0}^{t=1}  \int \frac{\delta F}{\delta q}(q_t) \mathrm{d}(q-q') \mathrm{d}t
>     - \int \frac{\delta F}{\delta q}(q')\mathrm{d}(q-q')$
>
> by the fundamental theorem of calculus. Then, the RHS is
>
> $\int_{t=0}^{t=1}  \int \left[ \frac{\delta F}{\delta q}(q_t) - \frac{\delta F}{\delta q}(q') \right] \mathrm{d}(q-q') \mathrm{d}t.$
>
> Using the $L_\mathrm{TV}$-Lipschitz continuity of $\frac{\delta F}{\delta q}$, it is bounded by
>
> $\int_{t=0}^{t=1} L_\mathrm{TV} \mathrm{TV}(q,q')^2 \mathrm{d}t = \frac{L_\mathrm{TV}}{2}\mathrm{TV}(q,q')^2,$
>
> which implies the smoothness
>
> $F(q) \leq F(q') + \int \frac{\delta F}{\delta q}(q')\mathrm{d}(q-q') + \frac{L_\mathrm{TV}}{2}\mathrm{TV}(q,q')^2.$
>
> From Pinsker's inequality, we also obtain
>
> $F(q) \leq F(q') + \int \frac{\delta F}{\delta q}(q')\mathrm{d}(q-q') + L_\mathrm{TV}D_\mathrm{KL}(q\|q').$
>
> Here, we emphasize that when the inner-loop error is ignored, it is possible to prove convergence using only the smoothness w.r.t. $D_\mathrm{KL}$ instead of the Lipschitz continuity of $\frac{\delta F}{\delta q}$.
> Your excellent suggestions have been incorporated into the manuscript.
>
> **Characterization of convergence in the nonconvex setting**
>
> Thanks for your insightful question.
> It is indeed shown that the solution converges to a (near) stationary point as we presented in Corollary 1.
> What Corollary 1 claims is that
> the functional derivative of the regularized loss also converges to zero:
> with sufficiently large $k$, we can approximately evaluate
>
> $\min_{k=1,...,K} \mathbb{E}\_{x\sim q^{(k)}}\left[ \left(
> \frac{\delta L}{\delta q}(q^{(k)},x) - \mathbb{E}_{x'\sim q^{(k)}}\left[\frac{\delta L}{\delta q}(q^{(k)},x')\right]\right)^2\right] = \mathcal{O}\left(\frac{1}{K}\right),$
>
> because $D_\mathrm{KL}(\hat{p}^{(k)}\|\hat{p}^{(k+1)})$ will be approximately the *moment generating function* $\psi_{q^{(k)}}$ of $\frac{\delta L}{\delta q}(q^{(k)})$ and asymptotically approaches the *variance* of it. Therefore, we interpret the above equation as
>
> $\frac{\delta L}{\delta q}(q^{(k)},x) \to 0~~~(\text{up to constant w.r.t. $x$}),$
>
> which implies that the solution $q^{(k)}$ converges to a stationary point as in the standard finite dimensional optimization.
>
> ---
>
> In the event that all your concerns have been fully resolved, we would be grateful if you could think about adjusting the rating score positively, if it aligns with your judgment.

---

> > ### Comment · Reviewer_ei8r · 2024-11-26
> > **Response to Rebuttal**
> >
> > Thank you for your hard work on the rebuttal. I have raised my score to 8. Please propagate these types of changes in the final version.

---

### Official Review · Reviewer_141C · 2024-11-04

**Soundness:** 2
**Presentation:** 3
**Contribution:** 2
**Rating:** 5
**Confidence:** 4

**Summary:**

The paper formulates the diffusion model alignment problem as a general distribution optimization problem. The proposed general objective, which involves regularized loss minimization over probability distributions, encompasses important tasks in language model alignment, including RLHF, DPO, and KTO. The authors propose an algorithm that first employs Dual Averaging to approximate the optimization objective function over a distribution into a potential function (thereby transforming the variable from a distribution to a point), and then uses this learned potential function to fine-tune diffusion models. The approach is supported by theoretical guarantees.

**Strengths:**

1. The paper's formulation of a general distribution optimization problem is significant as it unifies several widely-studied alignment tasks under a single framework, making it relevant to current research priorities in the field.
2. The paper is well-written and easy to follow.

**Weaknesses:**

1. The paper lacks crucial implementation details for applying Dual Averaging to diffusion models. Specifically, while the objective involves  $KL(q|p_{ref})$ and $exp(-f)p_{ref}$, it doesn't address how to compute these terms when only time-dependent score functions are available by diffusion models. The potential Monte Carlo alternatives would be computationally intensive.

2. In phase 2,  the proposed correction term u(x,t) in equation 17 is computationally impractical, requiring multiple reverse process simulations at each inference timestep.

3. Based on 2, the paper's key Assumption 3.4 regarding the small approximation error of u* is questionable, particularly given the gradient terms and complexity of the approximation.

**Questions:**

1. According to Alg. 4.1, we've already got $q_K \propto exp(-f_K)p_{ref}$, which is exactly the optimal solution of (1) (also the target of phase 2), why do we need to conduct phase 2?

2. The paper claims to provide convergence guarantees for diffusion alignment problems without relying on isoperimetric conditions. However, Assumption 3 requires that $\nabla \log p_t$ is $L_p$-smooth at every time $t$. Could this requirement serve as an alternative to isoperimetric conditions?

---

> ### Author Response · Authors · 2024-11-20
> **Official Comment (1/2)**
>
> We thank the reviewer for the thoughtful comment and constructive feedback. We address the technical concerns below.
>
> ---
>
> **Implementation details for applying Dual Averaging to diffusion models**
>
> This is a crucial question that addresses how we designed our algorithm.
> First of all, we would like to emphasize that our algorithm is designed so that we can avoid direct computation of the quantities $D_\mathrm{KL}(q\|p_\mathrm{ref})$ and $q \propto \exp(-f)p_\mathrm{ref}$
> because they are hardly obtained as you pointed out.
>
> We addressed these computational challenges by focusing on estimation of the density ratio of samples at the final step $q(x_T) / p_\mathrm{ref}(x_T) \propto \exp(-f(x_T))$ that is tractable with Dual Averaging.
> Actually, we highlight that $f$ is the weighted sum of the functional derivatives of $F$, which can be calculated with the estimated $f$ in the previous iterations within the Dual Averaging framework.
>
> For instance, when $F$ is the DPO objective function, the functional derivative of $F$ can be calculated by samples generated from $p_{\mathrm{ref}}$ (which is already given by a diffusion model) and the last iterate $f_k$.
> More specifically, the DPO objective $F$ can be represented as the expectation of $\log \sigma(-f(x_1) + f(x_2))$, where $x_1, \; x_2$ are the *win* and *lose* data points respectively which can be easily generated through the diffusion model corresponding to $p_\mathrm{ref}$ (and $\sigma$ is the sigmoid function). Additionally, the computation of KTO objective and its functional derivatives are also feasible in a similar manner.
>
> Just for clarification, we present an implementation of the Dual Averaging method as follows:
>
> 1. Initialize $\hat{f}_0$ by neural networks;
> 2. Get i.i.d. data points $x_1,\dots,x_n$ from the diffusion model corresponding to the reference model $p_\mathrm{ref}$;
> 3. Iteratively estimate $f$;
>     1. Empirically calculate $\frac{\delta F}{\delta q_k}$ for each $x_i$ with the density ratio $\exp(-\hat{f}_k)$ given in the previous steps;
>     2. Construct $\hat{f}\_{k+1}$ by estimating the average of $\frac{\delta F}{\delta q_k}$ by minimizing the mean squared error on the tuples $(x_i, \frac{\delta F}{\delta q_k}(x_i))_{i=1}^n$.
>
> Note again that, during the algorithm, we don't need to calculate $D_\mathrm{KL}(q\|p_\mathrm{ref})$ and $q \propto \exp(-f)p_\mathrm{ref}$.

---

> ### Author Response · Authors · 2024-11-20
> **Official Comment (2/2)**
>
> **Role and utility of phase 2 (sampling with correction term)**
>
> I appreciate your input, but Dual Averaging optimizes obtains only the output density ratio of samples at the final denoising step.
> The role of Phase 2 is to construct the denoising path with the correction terms, looking ahead the optimized final output density.
> The correction term at each denoising step is uniquely determined by the final output density ratio with Doob's $h$-transform technique.
>
> In addition, we agree that the proposed correction term requires multiple reverse process, however, this kind of techniques like Phase 2 has been empirically used in image generation tasks [R1], Bayesian computation [R2] and filtering [R3].
>
> Particularly, as a more practical option, the idea of approximating the correction term using neural ODE solvers for faster implementation has also been proposed in [R3].
>
> **Why do we need Phase 2?**
>
> It should be noticed that knowing the density does not imply that we can sample data from the density.
> You may consider that we can apply the Langevin dynamics to generate data from $q_K$, however, it is almost  intractable in real complicated data sets such as images due to their high dimensionality and multimodality.
> Instead, our strategy is to make the most use of the sampling efficiency of the diffusion model.
> It has been empirically and theoretically known that the sampling efficiency of the diffusion model is independent of the isoperimetric inequality (such as log-Sobolev inequality) of the target distribution, which is especially favorable for high dimensional multi-modal data.
> Our algorithm aims to construct an aligned diffusion model just by adding a correction term in Phase 2 in which we use the $h$-transform technique.
> In other words, Phase 2 is required to convert the density ratio to an (efficient) sampling algorithm.
>
>
> **$L_p$-smoothness of the score $\nabla \log p_t$ and isoperimetric conditions**
>
> Your point is important; indeed, it is the case that $L_p$-smoothness of the score $\nabla \log p_t$ implies log-Sobolev in equality with some constant $C > 0$, assuming that $p_\mathrm{ref}$ satisfies log-Sobolev inequality.
> However, the log-Sobolev inequality becomes very loose when the dimension $d$ goes extremely high. The constant $C$ becomes exponentially small with $d$, as we mentioned in the line 178 to 180 in the first submission; *which is highly complex and has multi-modality, leads to
> failure or weak satisfaction of LSI. As a result, the convergence of MFLD is significantly slowed down.*
> Therefore, $L_p$-smoothness of the score is weak enough to add it.
>
> ---
>
> Since your suggestions were very valuable, we have clarified these parts you pointed out.
>
> If all the issues you raised have been adequately resolved, we would humbly request you to consider revising the rating score upwards, should you find it reasonable.
>
> ---
>
> [R1] Masatoshi Uehara, Yulai Zhao, Kevin Black, Ehsan Hajiramezanali, Gabriele Scalia, Nathaniel Lee Diamant, Alex M Tseng, Tommaso Biancalani, Sergey Levine: Fine-Tuning of Continuous-Time Diffusion Models as Entropy-Regularized Control. arXiv, 2024.
>
> [R2] Jeremy Heng and Valentin De Bortoli and Arnaud Doucet: Diffusion Schrödinger Bridges for Bayesian Computation. Statistical Science, 2024.
>
> [R3] Nicolas Chopin, Andras Fulop, Jeremy Heng, Alexandre H. Thiery: Computational Doob $h$-transforms for Online Filtering of Discretely Observed Diffusions. ICML, 2023.

---

> > ### Comment · Reviewer_141C · 2024-11-27
> > **Reply to rebuttal**
> >
> > I sincerely appreciate the authors' efforts and have raised my rating to 5. However, I still have some questions:
> >
> > Assumption on the Correction Term:
> > You mentioned that using multiple reverse processes to estimate the correction term has been implemented by others, such as in [1]. However, such fine-tuning methods severely suffer from catastrophic forgetting, which implies that the correction term cannot be guaranteed. Given this, I don't think Assumption 3.4—central to your theorem and distinct from prior extensive literature—is reasonable.
> >
> > Purpose of Phase 2:
> > As you mentioned, diffusion can address high-dimensional sampling effectively. However, I still believe that using diffusion (a score-based method) when the density is already known is unnecessary and redundant.
> >
> > [1] Masatoshi Uehara, Yulai Zhao, Kevin Black, Ehsan Hajiramezanali, Gabriele Scalia, Nathaniel Lee Diamant, Alex M Tseng, Tommaso Biancalani, Sergey Levine: Fine-Tuning of Continuous-Time Diffusion Models as Entropy-Regularized Control. arXiv, 2024.

---

> ### Comment · Area_Chair_4jV8 · 2024-11-26
> **Please reply to authors**
>
> Reviewer 141C,
>
> Thank you very much for your work on reviewing this submission.
>
> Could you read the author's reply, check if they have addressed your comments, and acknowledge it?
>
> Further, please make sure, if necessary, to explain what your new stance is based on this information.

---

> ### Author Response · Authors · 2024-11-28
> **Reply to Reviewer 141C**
>
> Thank you for your thoughtful and insightful questions. Below, we address them in detail:
>
> - **Assumption on the Correction Term**
>
>     - You are correct that we assumed the correction term at each step of the denoising process can be estimated with the error $\epsilon_{\rho,l}^2$ as stated in Theorem 3. However, we also demonstrated in Theorem 4 that $\epsilon_{\rho,l}^2$ grows linearly with $d$, which is relatively acceptable in high-dimensional settings.
>
> - **Catastrophic forgetting**
>
>     - First, our approach leverages entropy-regularized fine-tuning, which mitigates catastrophic forgetting in the aligned model by incorporating KL-regularization into the loss function, as detailed in [R1, R2, R3, R4]. This KL-regularization enables us to preserve the "naturalness" of the pre-trained model by forcing the aligned model to have the same support as te pre-trained diffusion model, which prevents *reward collapse* and catastrophic forgetting.
>     - Second, the correction terms are produced by a separate model, ensuring that their inclusion helps safeguard the pre-trained model and reduces the risk of catastrophic forgetting compared with such methods as classifier-free guidance and RL-finetuning which directly fixes the pre-trained model, as demonstrated in [R2].
>
>
> - **Purpose of Phase 2**
> -
>     - We would like to emphasize that in our setting, the densities $p_\mathrm{ref}$ and $q$ themselves remain *unknown*, instead, only the estimate of the density ratio $q / p_\mathrm{ref}$ is available. Hence, we cannot exploit the density information directly.
>     - Additionally, the sampling from the Gibbs distribution $\exp(-f)$ is quite challenging when it does not or weakly satisfy isoperimetric condition even when $f$ is accessible. In fact, this motivates the development of new sampling methods (e.g., [R5]) that utilize diffusion models without relying on ispperimetry. We note that our problem also falls in the similar situation.
>
> ---
>
> [R1] Masatoshi Uehara et al. (2024). Fine-Tuning of Continuous-Time Diffusion Models as Entropy-Regularized Control. ArXiv preprint. https://arxiv.org/abs/2402.15194
>
> [R2] Xiner Li et al. (2024). Derivative-Free Guidance in Continuous and Discrete Diffusion Models with Soft Value-Based Decoding. ArXiv preprint. https://arxiv.org/abs/2408.08252
>
> [R3] Wenpin Tang. (2024). Fine-tuning of diffusion models via stochastic control: entropy regularization and beyond. ArXiv preprint. https://arxiv.org/abs/2403.06279
>
> [R4] Hanyang Zhao et al. (2024). Scores as Actions: a framework of fine-tuning diﬀusion models
> by continuous-time reinforcement learning. ArXiv preprint. https://arxiv.org/abs/2409.08400
>
> [R5] Huang, X., Zou, D., Dong, H., Ma, Y. &amp; Zhang, T.. (2024). Faster Sampling without Isoperimetry via Diffusion-based Monte Carlo. Proceedings of Thirty Seventh Conference on Learning Theory, in Proceedings of Machine Learning Research 247:2438-2493 Available from https://proceedings.mlr.press/v247/huang24a.html.

---

### Official Review · Reviewer_aaJd · 2024-11-04

**Soundness:** 3
**Presentation:** 3
**Contribution:** 3
**Rating:** 8
**Confidence:** 2

**Summary:**

The authors propose a distributional optimization algorithm that builds on the dual averaging (DA) algorithm combined with Doob's
ℎ-transform. This framework encompasses several methods, including Reinforcement Learning(RL), Direct Preference Optimization (DPO), and Kahneman-Tversky Optimization (KTO). The paper includes an error analysis and provides numerical implementations of Gaussian mixtures and image generation tasks.

**Strengths:**

1. The paper provides a general framework for distributional optimization, integrating a sophisticated optimization algorithm (dual averaging) with Doob's ℎ-transform to capture the score function effectively.

2. The authors present a convergence rate analysis based on suitable assumptions, enhancing the theoretical robustness of the approach.

3. Experiments are conducted on both synthetic data (Gaussian mixtures) and real-life CT images, demonstrating the practical applicability of the proposed method.

**Weaknesses:**

The method introduces an additional entropic regularization term involving a hyperparameter, $\beta'$. In the theoretical analysis, beta prime is assumed to be larger than $\beta$, yet in the experiments, $\beta'$ is set equal to $\beta$ in all cases. There is a lack of discussion on the role of these parameters and their impact on the results.

**Questions:**

Typically, the score estimation error is assumed in the $L^2$ sense, but Assumption 3.3 assumes it in the $L^\infty$ sense. Could this assumption be improved?

---

> ### Author Response · Authors · 2024-11-20
>
> We thank the reviewer for the thoughtful comment and constructive feedback. We address the technical concerns below.
>
> ---
>
> **The role of hyperparameters**
>
> Thanks for paying attention to an important point.
> The hyperparameter $\beta'$ plays a similar role to an inverse learning rate and should be set so that the solution does not diverge in the theoretical analysis.
> More concretely, when the objective $F$ is smooth, we have shown that the objective function with additional regularization imposed by the hyperparameter $\beta'$
>
> $\tilde{L}\_k(q) = F(q) + \beta D_\mathrm{KL}(q\|p_\mathrm{ref}) + \frac{\beta'}{k} D_\mathrm{KL}(q\|p_\mathrm{ref})$
>
> monotonically decreases during our DA algorithm under the constraint of $\beta'$.
> This result also implies that the hyperparameter $\beta'$ controls the speed of the convergence of the regularized objective $F(q) + \beta D_\mathrm{KL}(q\|p_\mathrm{ref})$. In other word, the smaller $\beta'$ yields a more *aggressive* update while there would not be a theoretical guarantee when $\beta'$ violates the constraint.
>
> In our experiment, we chose the hyperparameter settings in a naive manner to simplify the experiments. However, we have observed how the DPO loss during the Dual Averaging behaves when $\beta'\, (>\beta)$ varies with Gaussian Mixture Model. The additional experimental results with $(\beta,\beta')=(0.04,0.04\times i)~(i=1,\dots,5)$ showed that the smaller $\beta'$, the faster the convergence is, confirming that it indeed plays a role similar to that of an inverse  learning rate.
>
>
> **The score estimation error**
>
> Thanks for pointing out an important point.
> Indeed, the $L^\infty$-error constraint is not necessary, and it can be replaced by the $L^2$-error constraint because we are using only the $L^2$-error in the proof.
> We have fixed this point in the revised manuscript.
>
> ---
>
> Should all of your concerns be satisfactorily addressed, we would greatly appreciate it if you might consider improving the rating score, if you deem it appropriate.

---

> > ### Comment · Reviewer_aaJd · 2024-11-27
> >
> > I appreciate the authors' informative response, which has addressed my questions thoroughly. As a result, I have increased my score.

---

### Official Review · Reviewer_hXn1 · 2024-11-09

**Soundness:** 3
**Presentation:** 3
**Contribution:** 3
**Rating:** 8
**Confidence:** 2

**Summary:**

Authors study the problem of sampling from distribution $\hat q$ given as the minimizer of a general optimization problem in the space of distributions given by $F(q) + KL(q||p_{ref})$, where $p_{ref}$ is the base distribution (e.g. obtained by pretraining) and $F$ is a functional in the space of distributions. $F$ can be defined as the DPO objective given user preferences of the samples from the distribution. A simple toy example of the DPO based F, which authors use for sampling from a single gaussian when the reference measure $p_{ref}$ is mixture of gaussians,  is when the DPO preference function between two samples picks the one that is closer to the correct gaussian mean.

Authors introduce an algorithm for this task based on applying the Dual averaging method in the space of distributions, and learning the gradient of $F$ at each iteration by neural networks. At the end of DA, they get access to a neural net that approximates the log density and the score of $\hat q$ with respect to $p_{ref}$. Then, they use a trick based on Doob’s $h$-transform relation to translate this information to the score function of  $\hat q$, which they use to obtain fresh samples from $\hat q$ using the backward process of Ornstein-Uhlenbeck.

**Strengths:**

The overal approach linking optimization in the space of distributions to diffusion models is new to me and looking interesting. Authors seem to derive bound for smoothness coefficients of score function of $\hat q$ based on smoothness of the diffusion for the reference measure. Their approach in using Doob's h-transform for estimating the score functions of the corrected distribution is also very interesting. The presentation of the beginning part of the paper looks clean and authors clearly state their high level goal.

**Weaknesses:**

Overall choice of the Dual averaging for this task seems unmotivated, I don’t see why we can’t use simpler algorithms like GD in the space of distributions, and learning the gradients with neural nets?

Other concerns:
One concern that I have is where authors use Doob’s h-transform to estimate the score of $\hat q$ as \nabla \log\E[e^{-f(X_t)} | X_T], by estimating the derivative of probability density instead of log density, in line 1252, with samples. Given that estimating the density is ill-conditioned in general, can authors elaborate on the exact guarantee they get for this approach?

The presentation of the proofs can be more clear, right now the logical order between Lemmas, etc is not clear. I would suggest authors add more intuition before stating the mathematical formulations in each Lemma, in how that relates to the bigger picture.

The derivative notation in line 143 is not clear.
224: one and zero should be written in English rather than numbers

**Questions:**

What is $\nabla_x \nabla_x^\top$ in line 1471
What is the meaning of subindices $|_{x=\bar X_t}$ ? The notation is very confusing
What is the benefit of using the Dual averaging method compared to gradient descent in the Wasserstein space? I assume you can also implement that by learning the gradient of the functional by neural nets?

Can authors clarify why they use the proof of Liu et al for their proof of convergence of the Dual averaging method in the non-convex case, instead of the more classical older papers on Dual averaging?

In the case of gaussian mixture where the authors define the objective of DPO  with the goal of sampling from one of the single gaussians, is it the case that it can mathematically be shown that the minimizer of the DPO objective is exactly a single gaussian?

In the case of gaussian mixture, Is it possible to prove guarantees for learning the sequence $\bar q_k$ rigorously with neural nets or some polynomial regression?

---

> ### Author Response · Authors · 2024-11-20
>
> We thank the reviewer for the thoughtful comment and constructive feedback. We address the technical concerns below.
>
> ---
>
> **The motivation of Dual Averaging**
>
> Indeed, we can derive a Gradient Descent (GD) version of DDO, however, we also see that the GD approach does not directly give the density ration $q^{(K)}/p_{\mathrm{ref}}$ required for the $h$-transform calculation.
> Indeed, the update of the GD method can be formulated as the following type of an optimization problem:
>
> $q^{(k)} \leftarrow \arg\min_q \mathbb{E}_q[g] + \beta'^{-1} \mathrm{KL}(q || q^{(k-1)}).$
>
> This update gives only the density ratio between the current solution $q^{(k)}$ and the last update $q^{(k-1)}$ (i.e., $q^{(k)}/q^{(k-1)}$) instead of the direct ratio $q^{(k)}/p_{\mathrm{ref}}$ that can be obtained by the DA approach.
> We employed the later one because the GD approach should take the product $\prod_{k=1}^K q^{(k)}/q^{(k-1)}$ to obtain the final outcome $q^{(K)}/p_{\mathrm{ref}}$ which is complicated, computationally demanding and could cause an accumulated error.
>
>
> **Estimation of the correction term**
>
> Note that we are essentially estimating the Radon-Nikodym derivative  between $q^{(K)}$ and $p_{\mathrm{ref}}$ which can exist even when the density is degenerated.
> Actually, under some boundedness condition on the first order variation, the density ratio can be bounded and hence we can estimate the $h$-transform even when the density is ill-posed.
>
> Moreover, the time-discretization error in the reverse process can be well regulated as in our Theorems 3 and 4 by paying attention to the short-time Gaussian approximation at each time step.
> This discretization error vanishes as the time step size goes to zero and the score estimation becomes accurate.
>
> **Presentation of proof**
>
> Thanks for your suggestion.
> Please notice that there is a technical overview of the argument at the beginning of the proof section (C.1). However, following your comment, we have added a concise explanation of each lemma before each lemma is presented. Especially, we considerably enlarged Section A to give the proof overview for the convergence of our DA method.
>
> **Notation**
>
> The definition of the functional derivative (also known as first order variation) is quite standard. We have added a sentence to make the definition clearer.
> We also added a table of notations at the beginning of the appendix, which we believe helps the readers a lot.
>
> To address your other questions regarding the notation, $\nabla_x \nabla_x^\top$ refers to Hessian matrix, so the $i,j$-th component of the $d\times d$ matrix $\nabla_x \nabla_x^\top f(x)$ is $\frac{\partial}{\partial x_i}\frac{\partial}{\partial x_j} f(x)$. $\nabla_x \nabla_x^\top$ is also abbreviated as $\nabla_x^2$, so we have standardized them to the latter.
> In addition, the sub-indices $\mathbb{E}[\dots | \bar{X}\_t = x]|_{x=\bar{X}_t}$ refers to substituting the random variable $\bar{X}_t$ into $x$. It might have been simpler to write $\mathbb{E}[\dots|\bar{X}_t]$, so we have changed this notation to this.
>
>
> **Proof citations for the Dual Averaging method**
>
> The convergence rate of the DA method for non-convex objective has not been so much interested in the literature, and thus as far as we know, there is not a direct reference for that (although an accelerated DA method with a line searched step-size is considered in [R1]). For example, the original paper by Nesterov (2009) dealt only the convex setting. Thus, we cited Liu et al. (2023) for the most relevant reference.
>
> [R1] Yurii Nesterov, Alexander Gasnikov, Sergey Guminov, Pavel Dvurechensky: Primal-dual accelerated gradient methods with small-dimensional relaxation oracle. 2018.
>
> **Minimizer of the DPO objective**
>
> The minimizer of the DPO objective would not be a strictly single Gaussian, because we also have the regularization coefficient $\gamma$ in the DPO objective.
> However, if the preference label is generated _exactly_ from the Bradley-Terry model, then the minimizer of DPO with infinitely many preference labels (pairs of winning and losing data) and zero regularization parameter will exactly become the single Gaussian.
>
> **Guarantees for learning the sequence $q^{(k)}$**
>
> For a non-zero $\beta$, the optimal density ratio is bounded and smooth for a non-zero KL-regularization term. In that situation, our theory implies that our method with a sufficiently large neural network can estimate the optimal density ratio with small error.
> As for an estimation error, we need to go through a finite sample generalization error analysis but it is not in the scope of this paper. We would like to defer it to the future work.
>
> ---
>
> If all of your concerns have been fully addressed, we would sincerely appreciate it if you could kindly consider raising the rating score, should you find it appropriate.

---

### Author Response · Authors · 2024-11-21
**General Comment**

First of all, we would like to express our sincere gratitude for your thoughtful and constructive feedback on our work. Your comments and suggestions have been invaluable in helping us identify areas for improvement and clarify our contributions.

We have carefully reviewed all your comments and would like to address a few key points here:

1. **Motivation of our algorithm**:
    - The reviewers expressed concerns about why Dual Averaging is effective in the current distribution optimization task.
    - By employing Dual Averaging, we only need the neural network to directly learn the density ratio $q^{(K)}/p_\mathrm{ref}$ of the output distributions at the final step of the denoising process in the diffusion models, while a naive gradient descent type method does not. This approach enables us to achieve an aligned output density.
    - Using this density ratio, the Doob's $h$-transform uniquely determines the correction terms for each time step in the aligned diffusion model.

2. **Computational efficiency of our algorithm**

    - The reviewers raised questions about the computational cost associated with the Doob's $h$-transform technique.
    - Our simplest solution (i.e., estimating the correction term at each time step using Monte Carlo) has a time complexity of $\mathcal{O}(L^2)$, where $L$ represents the number of time steps in the denoising process. When we set $N$ as the number of particles needed to estimate one correction term, to compute the correction term for each sample simultaneously, $\mathcal{O}(N)$ memory space is required.
    - We emphasize that Doob's $h$-transform techniques used in our phase 2 have been empirically applied in image generation tasks [R1], Bayesian computation [R2], and online filtering [R3]  with efficient sampling schemes.


3. **Comparision between the prior work**:
   - The reviewers pointed out the need for a comparison with existing methods [R4]. In response, we conducted additional numerical experiments and confirmed that our algorithm achieves a smaller loss while remaining within acceptable computational resource limits during Dual Averaging phase.


4. **The role of the hyperparameter**:
   - The reviewers raised questions about the role of the hyperparameter $\beta'$ in our algorithm. In our approach, $\beta'$ control the speed of optimization: The objective function with additional regularization imposed by the hyperparameter $\beta'$, which is $L(\hat{q}^{(k)}) + \beta'/k D_\mathrm{KL}(\hat{q}^{(k)}\|p_\mathrm{ref})$, monotonically decreases during Dual Averaging. Through experiments, we observed that smaller values lead to slower optimization, and we have included these results in the manuscript.

5. **Presentation of our contributions**
    - The reviewers suggested that the introduction and theorem proofs should be made more accessible to readers who may not be familiar with the relatively niche field of distribution optimization. We have incorporated this advice into the manuscript and then the manucript was substantially improved.
    - Additionally, we received numerous suggestions to improve the clarity of notation and enhance the precision of expressions.

We deeply appreciate the time and effort you have invested in reviewing our submission. Your feedback has helped us significantly refine and strengthen our work. Thank you again for your valuable insights and for contributing to the quality of our research.


Sincerely,
Authors of Submission4323.

---

[R1] Masatoshi Uehara, Yulai Zhao, Kevin Black, Ehsan Hajiramezanali, Gabriele Scalia, Nathaniel Lee Diamant, Alex M Tseng, Tommaso Biancalani, Sergey Levine: Fine-Tuning of Continuous-Time Diffusion Models as Entropy-Regularized Control. arXiv, 2024.

[R2] Jeremy Heng and Valentin De Bortoli and Arnaud Doucet: Diffusion Schrödinger Bridges for Bayesian Computation. Statistical Science, 2024.

[R3] Nicolas Chopin, Andras Fulop, Jeremy Heng, Alexandre H. Thiery: Computational Doob $h$-transforms for Online Filtering of Discretely Observed Diffusions. ICML, 2023.

[R4] Bram Wallace, Meihua Dang, Rafael Rafailov, Linqi Zhou, Aaron Lou, Senthil Purushwalkam, Stefano Ermon, Caiming Xiong, Shafiq Joty, Nikhil Naik (2024). Diffusion model alignment using direct preference optimization. In Proceedings of the IEEE/CVF Conference on Computer Vision and Pattern Recognition (pp. 8228-8238).

---

> ### Comment · Area_Chair_4jV8 · 2024-11-26
> **Question about related work**
>
> Thank you very much for all your replies, and for the added work that you have done on this submission. I will take of all these into account for my meta-review.
>
> Would it be possible for you to comment on the relationship of your work with (Marion et al., 2024)? It seems related to your project.
>
> Marion et al: Implicit Diffusion: Efficient Optimization through Stochastic Sampling - https://arxiv.org/abs/2402.05468

---

> > ### Author Response · Authors · 2024-11-28
> > **Response to Area Chair**
> >
> > Thank you for your valuable suggestion.
> >
> > We have carefully read the paper [R1] and acknowledge that it unifies Langevin dynamics and the fine-tuning of diffusion models as a general distributional optimization problem over the probability space. We found this study to be both intriguing and highly informative.
> >
> > However, we observed the following limitations in their theoretical analysis:
> >
> > - In Sections 4.1 and 4.2, the analysis is inherently constrained by the requirements of density availability and the logarithmic Sobolev inequality, which are fundamental to Langevin dynamics.
> > - In Section 4.3, the analysis of diffusion models is limited to the reinforcement learning approach and the single Gaussian assumption. Notably, it appears that optimization guarantees for the DPO objective are not addressed in their work.
> >
> > We have incorporated the information about the paper you kindly suggested into our work.
> >
> > [R1] Marion et al: Implicit Diffusion: Efficient Optimization through Stochastic Sampling - https://arxiv.org/abs/2402.05468

---

> > > ### Comment · Area_Chair_4jV8 · 2024-11-28
> > >
> > > Thanks a lot for your quick reply.

---

### Meta-Review · Area_Chair_4jV8 · 2024-12-20

**Metareview:**

This paper proposes a new algorithm for distribution optimization, specifically for aligning diffusion models, based on dual averaging and Doob's h-transform. The approach offers a general framework applicable to various alignment tasks and demonstrates promising empirical results. Reviewers found this paper interesting.

While the paper presents a novel and potentially valuable contribution to distribution optimization, it could benefit from clearer explanations for a broader audience, more detailed comparisons with existing methods, and further investigation of the computational costs and the role of hyperparameters. In particular, for the camera-ready, some of the citations suggested by the reviewers on RL for distribution optimization should be mentionned in Example 1 (e.g. Marion et al. (2024))

**Additional Comments On Reviewer Discussion:**

The reviewers contributed to the discussion with authors, and there were no major points to settle.

---

### Decision · Program_Chairs · 2025-01-22

Accept (Poster)